# SREBP modulates the NADP⁺/NADPH cycle to control night sleep in *Drosophila*

Vittoria Mariano [1,2], Alexandros K. Kanellopoulos[1], Giuseppe Aiello [1], Adrian C. Lo[1], Eric Legius [2], Tilmann Achsel[1] & Claudia Bagni [1,3] ✉

Sleep behavior is conserved throughout evolution, and sleep disturbances are a frequent comorbidity of neuropsychiatric disorders. However, the molecular basis underlying sleep dysfunctions in neurological diseases remains elusive. Using a model for neurodevelopmental disorders (NDDs), the *Drosophila Cytoplasmic FMR1 interacting protein* haploinsufficiency (*Cyfip85.1/+*), we identify a mechanism modulating sleep homeostasis. We show that increased activity of the sterol regulatory element-binding protein (SREBP) in *Cyfip85.1/+* flies induces an increase in the transcription of wakefulness-associated genes, such as the malic enzyme (*Men*), causing a disturbance in the daily NADP⁺/NADPH ratio oscillations and reducing sleep pressure at the night-time onset. Reduction in SREBP or Men activity in *Cyfip85.1/+* flies enhances the NADP⁺/NADPH ratio and rescues the sleep deficits, indicating that SREBP and Men are causative for the sleep deficits in *Cyfip* heterozygous flies. This work suggests modulation of the SREBP metabolic axis as a new avenue worth exploring for its therapeutic potential in sleep disorders.

Sleep supports physiological well-being, and it is evolutionarily conserved from worms to humans[1]. Sleep loss leads to severe consequences such as deficits in psychomotor vigilance, attention, cognitive functions, and cellular metabolism[2].

However, the understanding of the biological functions of sleep remains a challenge[3]. Sleep is regulated by a fine-tuned interplay between the homeostatic drive and the circadian rhythm[4,5]. Key genes regulating circadian rhythm have been identified[6] and are conserved from flies to humans[7,8]. In addition, it is known that the wakefulness status leads to the accumulation of wake-dependent homeostatic factors in the brain that promote sleep, while the sleep status clears or abrogates these factors to allow the waking state[9]. Among these factors, neurotransmitters, synaptic proteins, metabolic substrates, and immune modulators have been identified[9]. However, the molecular mechanisms that mediate sleep pressure, i.e., the homeostatic system, remain elusive. Thus, identifying molecular pathways that regulate sleep need is crucial to understanding the different functions of sleep[10].

Recent evidence shows that sleep and cell metabolism are two interconnected processes modulating each other; periods of prolonged wakefulness and insufficient sleep profoundly impact the metabolome and transcriptome profiles[11–13]. Different studies have suggested that sleep drives the decline in synaptic activity to consolidate or strengthen particular synapses and circuitry[14], prevents cellular damage[15], and favors metabolic restoration and clearance within the brain[16,17]. Recent work has shown that the ratio of NADP⁺ to NADPH within sleep-inducing neurons plays a role in sleep pressure in flies[18], and that genes involved in lipid metabolism are linked to fly sleep homeostasis[19,20]. Therefore, sleep may act specifically as a reparative and anabolic process at the cellular level[17,21,22].

While sleep deficits are considered comorbid symptoms associated with neuropsychiatric disorders, growing evidence supports a bidirectional effect. Lack of sleep exacerbates aggression, impulsivity, repetitive behaviors, hyperactivity, anxiety, depression, and withdrawal in psychiatric patients[23–25]. On the other side, behavioral impairment seems to worsen sleep homeostasis[26]. Abnormal sensory processing, over-sensitivity to environmental stimuli, and increased arousal may concur to the insurgence of delayed sleep onset problems[27].

[1]Department of Fundamental Neurosciences, University of Lausanne, Lausanne 1005, Switzerland. [2]Department of Human Genetics, KU Leuven, Leuven 3000, Belgium. [3]Department of Biomedicine and Prevention, University of Rome "Tor Vergata", Rome 00133, Italy. ✉e-mail: Claudia.Bagni@unil.ch

Copy number variations (CNVs) encompassing the *CYFIP1* gene are associated with increased risk for several neuropsychiatric disorders such as schizophrenia (SCZ), autism spectrum disorders (ASDs), attention deficit hyperactivity disorder (ADHD), reduced cognitive dysfunction, and motor dysfunctions[28–34]. Notably, sleep disturbances such as insomnia or sleep problems have been reported in individuals with BP1-BP2 microdeletion/duplication harboring the *CYFIP1* gene[35–37].

Animal models for *Cyfip1* haploinsufficiency recapitulate ASD and SCZ-related behavioral deficits and neurodevelopmental alterations[38–43]. We have recently shown that haploinsufficiency in the *Drosophila* homolog of the human *CYFIP1* (and *CYFIP2*) leads to associative learning deficits and dysregulation of mitochondria homeostasis and energy metabolism impinging on GABAergic neurotransmission and ultimately affecting social behaviors[30,43].

*Drosophila melanogaster* is a powerful model for dissecting the genetics and biological pathways underlying sleep behavior[44–47]. Sleep in *Drosophila* has most of the hallmarks of mammalian sleep and is associated with a higher sensory arousal threshold and a characteristic posture. In addition, the circadian and homeostatic regulation of the sleep-wake cycle is also conserved[44,48]. Genetic screenings and gene profiling studies in *Drosophila* allowed the identification of genes and pathways affecting wakefulness and sleep, highlighting the importance of cellular metabolism, protein homeostasis, synaptic excitability, and immune response[49–53].

Here, we show that *Cyfip* heterozygous (*Cyfip*[85.1/+]) flies have decreased sleep need at night-time onset and increased SREBP activity. We observe that the transcriptomic profile of *Cyfip* mutant flies is enriched in genes regulating the wakefulness-associated pathways, including lipid metabolism genes such as malic enzyme (*Men*) that is regulated by the sterol element-binding protein SREBP[54,55]. Men is known to reduce NADP$^+$ to NADPH, and we show that Men overproduction disrupts the NADP$^+$/NADPH ratio before sleep time. Pharmacological and genetic dampening of SREBP activity renormalizes malic enzyme activity and the NADP$^+$/NADPH ratio, alleviating the sleep disturbances in *Cyfip* heterozygous flies. Taken together, our data describe a molecular and mechanistic axis impaired in neurodevelopmental conditions associated with sleep disturbances.

## Results

### CYFIP regulates night-time sleep

The sleep pattern of *Cyfip* heterozygous flies (*Cyfip*[85.1/+])[56] and control flies was assessed over 24 h (Fig. 1a) using the *Drosophila* activity monitoring system (Supplementary Fig. 1a). The average sleep amount at night was significantly reduced in *Cyfip*[85.1/+] compared to control flies (Fig. 1a, b), while no change in the daytime sleep was observed (Supplementary Fig. 1b). Reduced night-time sleep was mainly due to shortened sleeping bouts (Supplementary Fig. 1c). To measure sleep quality and strength[57], we analyzed the consolidation index (CI), the sleep latency, and the wake after sleep onset (WASO) index. *Cyfip*[85.1/+] flies revealed a sleep fragmentation phenotype and low sleep quality (Fig. 1c) with an increased number of sleep bouts (Supplementary Fig. 1d). The sleep latency and WASO (Supplementary Fig. 1e, f) were increased in the *Cyfip*[85.1/+] mutants, suggesting an impairment in initiating and maintaining sleep.

Next, we probed the ability of flies to fall asleep after disturbing their sleep by external stimuli, following a previously described assay[58,59]. Flies were exposed to a light pulse of 5 min provided at ZT16, ZT18, and ZT20. The number of flies that were awakened by the light stimulus did not differ between *Cyfip*[85.1/+] and control conditions, because we chose a stimulus sufficiently strong to efficiently disturb the overall population (Supplementary Fig. 1g). Notably, *Cyfip* mutants showed a reduction by approximately 50% of baseline sleep over the three time points, thus remaining awake for a longer time (Fig. 1d), suggesting a state of increased arousal. The affected night-time sleep

in the *Cyfip* heterozygous flies was not due to starvation, because food intake was not different between the two genotypes during daytime (Supplementary Fig. 1h) nor 2 h before night onset (Supplementary Fig. 1i).

Although *Cyfip* expression during brain development has previously been described[39,56], its pattern of expression in the adult brain has been less investigated. We created an HA-tagged fly (CYFIP[N-HA]) using the CRISPR-Cas9 technology[60] (Supplementary Fig. 1j, k) and observed an HA expression in the entire brain and at synapses, as shown by the colocalization with the synaptic protein Bruchpilot (Supplementary Fig. 1l).

To test whether the decrease in night sleep was caused by CYFIP depletion in the brain, we knocked-down *Cyfip* using two different *UAS-Cyfip-RNAi* lines (IR[1] and IR[2]) driven by neuronal or glial promoters. We used the neuronal *Ras2Gal4* driver, which is expressed largely in the overall neuronal population of adult fly CNS[61], although not exclusively during larval development[62,63], coupled with *Tub-Gal80*[ts] to specifically avoid lethal effects of *Cyfip* knockdown during larval development[56]. *Cyfip* RNAi driven by the *Ras2Gal4;TubGal80*[ts] driver showed reduced *Cyfip* mRNA levels in adult fly heads (Supplementary Fig. 2a) and caused a night-time sleep reduction, phenocopying both the sleep pattern and sleep deficits of *Cyfip* mutant flies (Supplementary Fig. 2b–g). CYFIP abrogation in glia cells using *Repo-Gal4* and *NrvGal4* drivers did not affect night-time sleep (Supplementary Fig. 2h), suggesting that the impaired *Cyfip* expression in neurons is mainly responsible for the observed sleep dysregulation. Since the *Ras2Gal4* driver is not exclusively neuronal[62,63], CYFIP might be required also in non-neuronal tissue to regulate night-time sleep behavior.

We have previously shown that *Cyfip* reduction in GABAergic circuits causes social behavior abnormalities, an effect that is mediated by the mitochondrial transporter Aralar[43]. To investigate if Aralar or the involvement of the GABAergic system influence the sleep phenotype, we decreased *Cyfip* expression in GABAergic neurons and decreased excessive *Aralar* in the *Cyfip*[85.1/+] background. Under these experimental conditions, night-time sleep was not affected (Supplementary Fig. 3a, b), and the *Cyfip*[85.1]/*Aralar*[MI07SS2] flies maintained the sleep deficits (Supplementary Fig. 3b), suggesting that social and sleep behaviors are regulated by *Cyfip* through two distinct pathways. Finally, abrogating CYFIP in a cluster of sleep-regulating neurons in the dorsal fan-shaped body (dFBs/*ExFl2*) marked by the *R23E10Gal4* driver[64] was sufficient to reduce the amount of night sleep (Supplementary Fig. 3c–d). However, the involvement of other sleep-regulating circuits contributing to the sleep phenotype cannot be excluded.

The sleep pattern results from a coordinated action of circadian and homeostatic mechanisms. To assess whether *Cyfip*[85.1/+] flies have circadian rhythm deficits, we tested flies in complete darkness (free running activity) and analyzed the periodicity. The two genetic conditions did not show any difference in circadian rhythm (Supplementary Fig. 3e–f). In addition, no sleep defects were observed upon *Cyfip* RNAi expression in clock neurons or in pars intercerebralis or in ellipsoid body (Supplementary Fig. 3g). Therefore, we examined whether reduced sleep in *Cyfip* mutants might arise from defective homeostatic regulation. Flies were mechanically sleep deprived (SD) at night for 12 h and sleep recovery was analyzed the following day. *Cyfip*[85.1/+] flies showed decreased daytime sleep and recovered less from sleep deprivation (Fig. 1e). Similar results were obtained when *Cyfip*[85.1/+] were sleep deprived for 6 h and sleep recovery was measured in the following 6 h (Δ sleep) (Supplementary Fig. 3h). Notably, *Cyfip* mutants exhibited reduced sleep rebound compared to control flies. Together, these findings suggest that *Cyfip* mutant flies exhibit reduced night-time sleep, suggesting an alteration in the homeostatic control of sleep due to deficits in sleep initiation and maintenance. Furthermore, *Cyfip* heterozygous flies

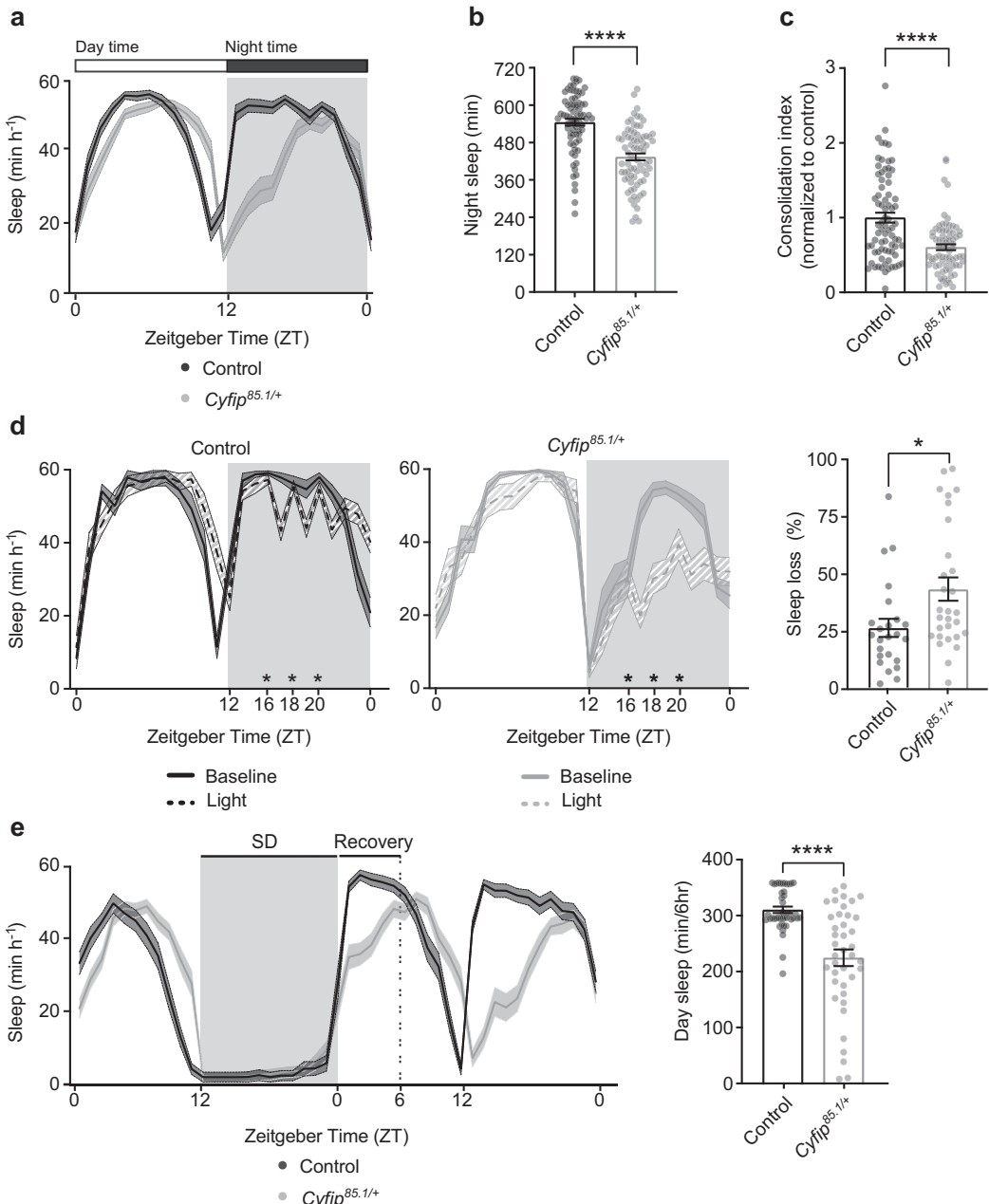

**Fig. 1 | CYFIP regulates the amount and quality of night-time sleep.**
**a** Representative sleep profile of control (black, *n* = 38) and *Cyfip*[85.1/+] (gray, *n* = 39) flies. Mean ± S.E.M. **b, c** Quantification of night sleep amount (**b**) and Consolidation Index (CI) at night (**c**) in control (*n* = 75) and *Cyfip*[85.1/+] (*n* = 75) flies. Two-tailed unpaired Student's *t* test, ****p < 0.0001. Mean ± S.E.M. **d** Waking response after sleep perturbation at ZT16, ZT18 and ZT20 (marked by "*") in control (*n* = 24) and

*Cyfip*[85.1/+] (*n* = 29) flies. Sleep loss is measured as the ratio between sleep after stimuli and the baseline sleep. Two-tailed unpaired Student's *t* test, *p = 0.015. Mean ± S.E.M. **e** Sleep deprivation (12 h) and sleep profile of control (*n* = 42) and *Cyfip*[85.1/+] (*n* = 40) flies. Two-tailed unpaired Student's *t* test, ****p < 0.0001. Mean ± S.E.M. Source data are provided as a Source Data file.

show reduced sleep rebound after sleep deprivation, underlying deficits in sleep homeostatic mechanisms.

**Cyfip mutant flies have an imbalance in the NADP⁺/NADPH ratio**

Recently, it was shown that the sleep-promoting dorsal fan-shaped body neurons are sensitive to the balance of the intracellular $NADP^+$/NADPH pool[18]. The activity of voltage-gated potassium channels of the *Shaker-Hyperkinetic* family ($K_v$ channels), important for neuronal excitability, is also inhibited by binding of the metabolite NADPH and therefore by low $NADP^+$/NADPH ratios[65]. *Shaker* activity promotes the sleep state[18], implying that low $NADP^+$/NADPH ratios decrease sleep

pressure. Given the observed alteration in homeostatic sleep control in *Cyfip*[85.1/+] mutant flies, we hypothesized that this might reflect an imbalance in brain $NADP^+$/NADPH ratio oscillations in relevant time periods such as sleep initiation. We next measured the $NADP^+$/NADPH ratio over the 24-h cycle in control and *Cyfip*[85.1/+] flies using hydrophilic interaction chromatography coupled to targeted mass spectrometry analysis (HILIC-MS/MS). Importantly, we observed that in control flies the $NADP^+$/NADPH ratio follows the behavioral state of the fly, i.e., rising at periods when the flies are awake and show increased locomotion activity (ZT0 and ZT12) and descending at ZT16 when sleep need is gradually reduced (Fig. 2a). In contrast, *Cyfip* mutant flies do

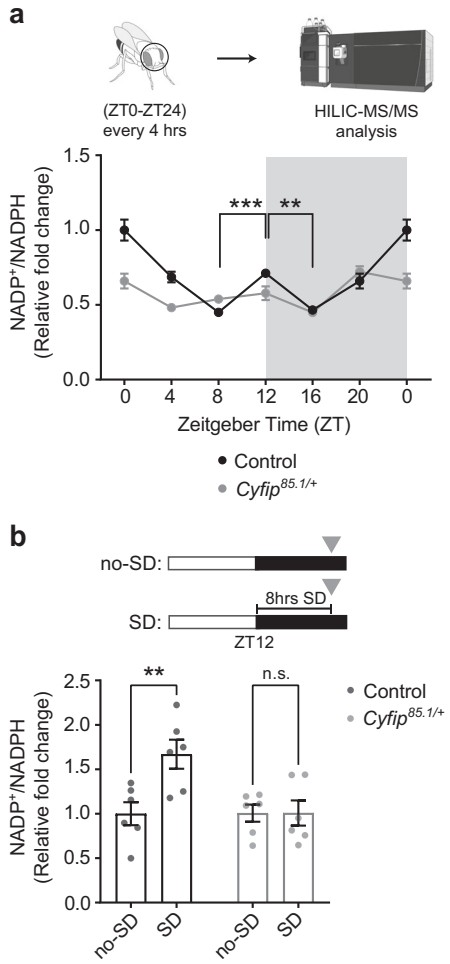

**Fig. 2 | *Cyfip* mutant flies have an imbalance in the NADP⁺/NADPH ratio. a** Upper inset, schematic of the experimental design; lower inset, quantification of the NADP⁺/NADPH ratio by HILIC-MS/MS in control and *Cyfip*$^{85.1/+}$ fly heads. Control $n = 5$, *Cyfip*$^{85.1/+}$ $n = 5$ over 24 h. $n$ = independent samples, pool of 50 fly heads, for each time point. Two-way ANOVA followed by Tukey's multiple comparisons test; time point effect $p < 0.0001$, genotype effect $p < 0.0001$, interaction effect $p < 0.0001$, $F_{(6, 56)} = 9.061$, control ZT8 vs. ZT12 ***$p = 0.0009$; control ZT12 vs. ZT16 **$p = 0.0022$; *Cyfip*$^{85.1/+}$ ZT8 vs. *Cyfip*$^{85.1/+}$ ZT12 $p = 0.99$; *Cyfip*$^{85.1/+}$ ZT12 vs. *Cyfip*$^{85.1/+}$ ZT16 $p = 0.34$; *Cyfip*$^{85.1/+}$ ZT8 vs. *Cyfip*$^{85.1/+}$ ZT16 $p = 0.74$. Mean ± S.E.M. (see also Supplementary Data 6). Created with BioRender.com. **b** Upper inset, schematic of the experimental design. no-SD: undisturbed sleep, SD: sleep deprived. Lower inset, NADP⁺/NADPH ratio in control and *Cyfip*$^{85.1/+}$ fly heads in no-SD (control, $n = 6$, and *Cyfip*$^{85.1/+}$, $n = 6$) and SD conditions (control, $n = 6$, and *Cyfip*$^{85.1/+}$, $n = 6$). $n$ = independent samples, pool of 23 fly heads. Two-way ANOVA followed by Sidak's multiple comparisons test, sleep condition effect $p = 0.0217$, genotype effect $p = 0.0243$, interaction effect $p = 0.022$, $F_{(1, 20)} = 6.165$, control no-SD vs. SD **$p = 0.0043$; *Cyfip*$^{85.1/+}$ no-SD vs. SD $p > 0.999$, n.s. = not significant. Mean ± S.E.M. Source data are provided as a Source Data file.

not show significant changes in the NADP⁺/NADPH ratio (between ZT8 and ZT16) or at the evening peak of locomotion activity (ZT12) (Fig. 2a).

To assess if the imbalance in NADP⁺/NADPH ratio could reflect a deficit in the homeostatic sleep mechanism and in the sleep pressure accumulation, we monitored the NADP⁺/NADPH in the whole brain, in control and *Cyfip*$^{85.1/+}$ flies during undisturbed sleep (no-SD) and after 8 h of sleep deprivation (SD), as described[13] (Fig. 2b, upper inset). After sleep deprivation, control flies show an increase in the NADP⁺/NADPH ratio, possibly consistent with an increase in sleep need, while no changes in the NADP⁺/NADPH ratio were observed in the *Cyfip*$^{85.1/+}$ flies (Fig. 2b, lower inset). These data suggest that the biochemical reactions regulating NADP balance are dysregulated in the *Cyfip*$^{85.1/+}$ brain at sleep

onset, thus inhibiting the accumulation of NADP⁺, hence decreasing the need for sleep necessary for initiating the sleep behavior.

## CYFIP regulates energy metabolism

To characterize the molecular mechanism underlying sleep need and amount, we performed a comparative RNA-Seq in *Cyfip* mutants and control fly heads at ZT16 (Fig. 3a), the period when *Cyfip* mutants showed impaired sleep (Fig. 1a). Principal component (PC) analysis revealed a segregation of the samples according to the genotype (PC1 representing 55.4% of the data variance, Fig. 3b). 1303 genes were differentially expressed, with 730 being upregulated and 573 downregulated in the *Cyfip* mutants (Fig. 3c, d and Supplementary Data 1). KEGG pathway enrichment analysis on the dysregulated genes (Fig. 3e and Supplementary Data 2) highlighted energy metabolism, lipid metabolism, amino acid metabolism, metabolism of cofactors and vitamins, and carbohydrate metabolism pathways being significantly dysregulated, possibly linking metabolic status and sleep homeostasis.

To investigate whether the dysregulated genes in *Cyfip*$^{85.1/+}$ might affect sleep and wakefulness, we compared the identified upregulated and downregulated genes with a *Drosophila* database for genes associated with wakefulness and sleep state[13]. The upregulated genes in *Cyfip*$^{85.1/+}$ significantly overlapped with wakefulness-associated genes (27 out of 121, hypergeometric test $p = 4.515^{-08}$), as did 4 downregulated genes (Fig. 3f and Supplementary Data 3). The overlapping genes are related to oxidative stress and antioxidants (i.e., *GstE1*, and *Cytochrome P450 subunits*), cellular metabolism, and immune response activity (i.e., *CG16756*). Interestingly, around 25% of these genes (7 out of 27) are related to lipid metabolism. Among them, genes such as the fatty acid synthetase 1 (*FASN1*) and Acetyl Coenzyme A synthase (*AcCoAS*) are more involved in the fatty acid synthesis, while the malic enzyme (*Men*) supplies molecules for lipid synthesis (i.e., NADPH). In *Drosophila*, the malic enzyme (*Men*) is one of the main contributors to the NADPH pool for fatty acid biosynthesis[66,67], together with the glucose-6-phosphate dehydrogenase (*Zw*) and the phosphogluconate dehydrogenase (*Pgd*) of the pentose phosphate pathway (PPP) and the cytosolic isocitrate dehydrogenase (IDH). We did not observe differential expression of IDH and the genes involved in the PPP, *Zw*, and *Pgd*, but we observed an over-expression of *Men* levels (Supplementary Data 1).

Together, these results show that the wakeful behavior exhibited by *Cyfip*$^{85.1/+}$ during the night-time (ZT16) is associated with a transcriptome profile linked to wakefulness, pointing to a specific dysregulation of genes related to lipid homeostasis and to a possible role in the regulation of the NADP⁺/NADPH ratio.

## SREBP activity regulates the amount of night-time sleep

The expression of genes related to lipid homeostasis (i.e., *Men, FASN1*, acetyl-CoA carboxylase (*ACC*)) is under the control of the sterol regulatory element-binding proteins (SREBPs). This regulation is conserved between flies and mammals[55,68]. To address whether dysregulation in SREBP expression and/or activity might be causative of the dysregulation of lipid genes, and in particular of *Men*, observed in *Cyfip* mutants, we compared the RNA-Seq dataset with transcriptomic datasets performed in human, murine, and *Drosophila* models overexpressing SREBP[69-71]. We identified 130 overlapping genes (Fig. 4a and Supplementary Data 4) and protein-protein interaction (PPI) network analysis revealed a major involvement in lipid metabolism, neuronal signaling, and protein synthesis (Supplementary Fig. 4a). Subsequently, we compared the SREBP-overexpression dataset to genes associated with wakefulness and sleep[13] and found 21 common genes (Supplementary Data 5). Among those genes, 10 were identified as wakefulness-associated genes and dysregulated in *Cyfip*$^{85.1/+}$ flies, underlying the relevance of CYFIP and SREBP in such a regulation (hypergeometric test $p = 3.467^{-21}$, Fig. 4b and Supplementary Table 1). Of those, *AcCoAS*,

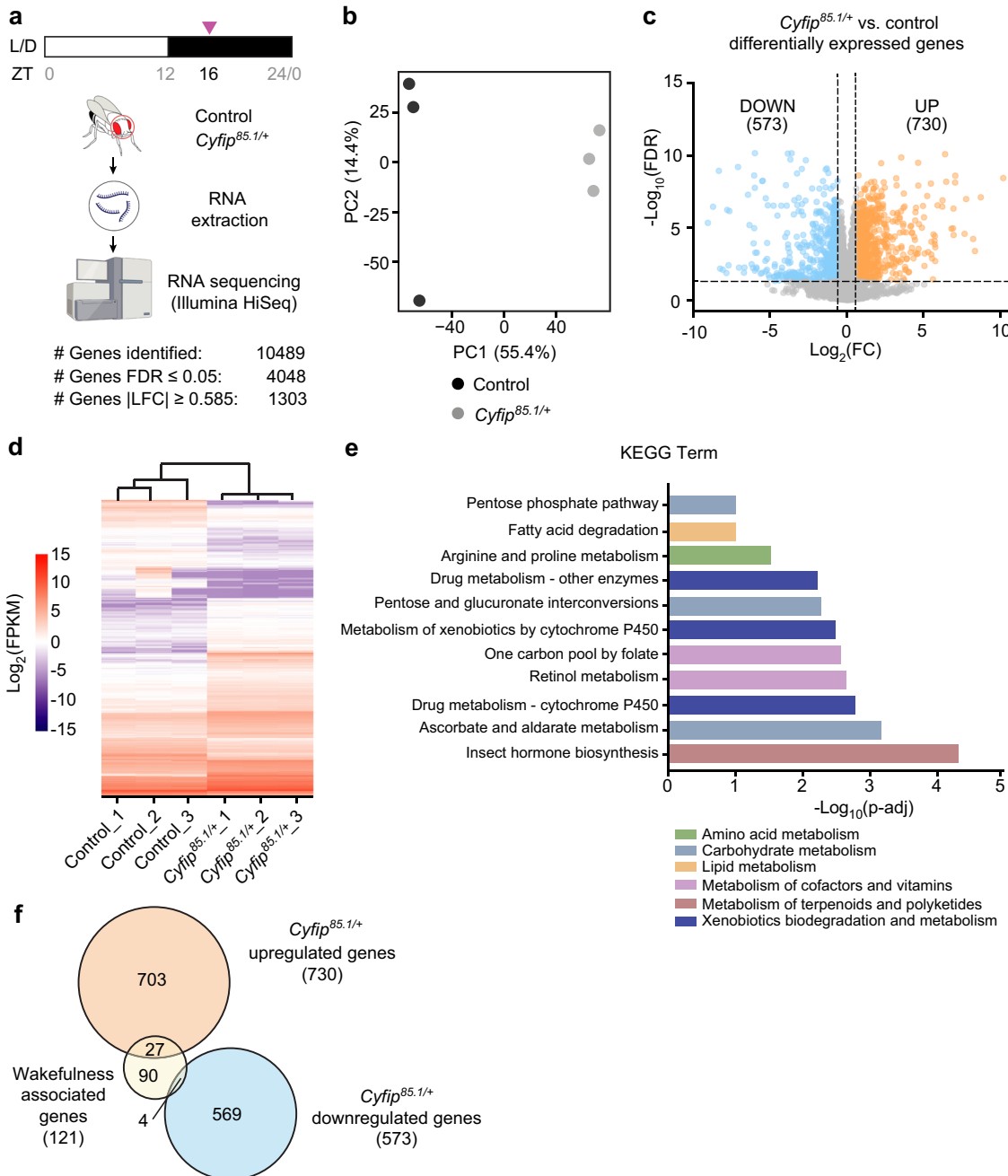

**Fig. 3 | The *Cyfip* mutant transcriptome revealed a dysregulation of energy metabolism. a** Schematic of the experimental design. Listed is the number of genes that were identified and passed consecutive filters. Created with BioRender.com. **b** Principal component analysis of RNA-seq data. Samples are color-coded according to the genotype. *n* = 3, where *n* = independent samples, pool of 20 fly heads. **c** Volcano plot showing the global transcriptional changes in control vs *Cyfip*[85.1/+] determined by RNA-seq. Moderated *t* test followed by the Benjamini-Hochberg method test for false discovery rate (FDR). Each circle represents one RNA, and colored circles represent differentially expressed genes (DEGs) significantly upregulated (FDR ≤ 0.05 and LFC ≥ 0.585 (in orange)) or significantly downregulated (FDR ≤ 0.05 and LFC ≤ −0.585 (in blue)). **d** Heatmap representing the DEGs significantly dysregulated in the different samples. **e** KEGG pathway enrichment analysis performed on the dysregulated mRNAs in *Cyfip*[85.1/+] mutants. Only the top interesting pathways for sleep and circadian rhythm behavior are represented in the scheme (see also Supplementary Data 2). Over-representation analysis (ORA) followed by Benjamini–Hochberg for false discovery rate. Parent KEGG terms are indicated in color. **f** Upregulated and downregulated genes (FDR ≤ 0.05, |LFC| ≥ 0.585) in *Cyfip*[85.1/+] flies have been compared with the database for *Drosophila* wakefulness genes[13]. *p* = 4.514[−08] assessed by hypergeometric test, overlap between wakefulness-promoting genes and upregulated genes in *Cyfip*[85.1/+] flies. Source data are provided in Supplementary Data files.

*CG33110, Dbi, FASN1* and *Men* are directly involved in lipid metabolism. The overexpression of *AcCoAS, FASN1*, and *Men*, in addition to other dysregulated wakefulness-associated genes, was confirmed by RT-qPCR (Fig. 4c). These mRNAs are also found in databases containing gene expression profiles upon SREBP modulation[69–71], further reinforcing their dependence on SREBP levels.

Next, we assessed whether SREBP levels and/or activity were altered in *Cyfip*[85.1/+] flies. Notably, *Cyfip*[85.1/+] mutant brain exhibited increased levels of precursor (SREBP-p) and active cleaved SREBP (SREBP-c)[72,73] compared to controls (Fig. 4d) between ZT12 and ZT16. The increased SREBP activity in *Cyfip*[85.1/+] was further confirmed using a reporter for SREBP activation (Gal4-SREBP::GFP)[74] (Fig. 4e, f). Together

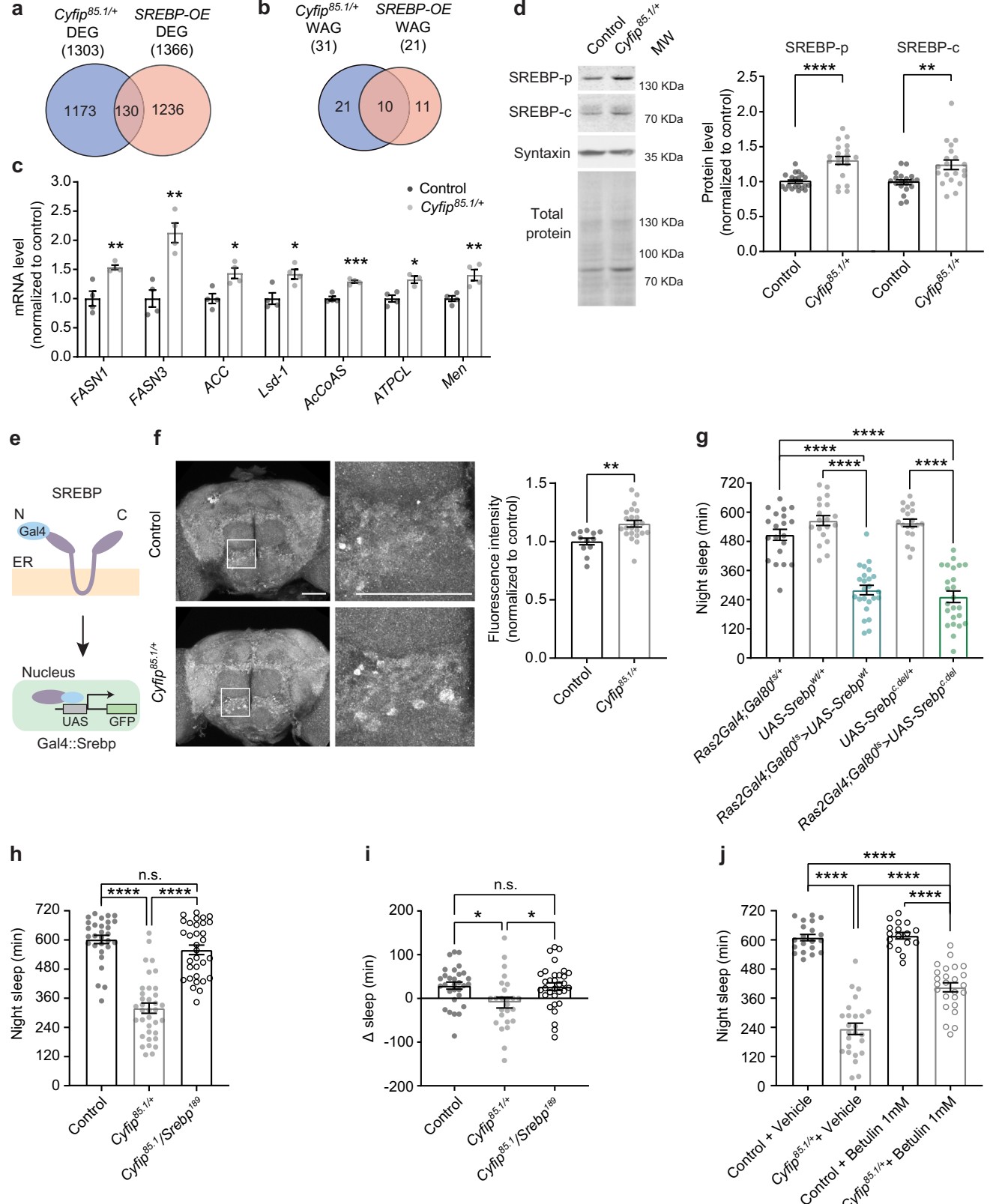

these results indicate that SREBP is overexpressed and overactive in *Cyfip[85.1/+]* mutant brain and suggest that SREBP might play a key role in sleep regulation, a function so far completely unexplored.

We therefore directly assessed the consequences of increased SREBP activity on the *Drosophila* night-time sleep behavior. Neuronal overexpression (driven by *Ras2Gal4*[62]) of the full-length SREBP

(*Srebp[wt]*) phenocopied the night-time sleep deficits observed in the *Cyfip[85.1/+]*, as well as upon *Cyfip* RNAi (Fig. 4g and Supplementary Fig. 4b). Furthermore, upon constitutively active nuclear form of SREBP (*Srebp[C.del]*) overexpression, the amount of night sleep was decreased (Fig. 4g and Supplementary Fig. 4b). Increased activity of SREBP had also an effect on the daytime sleep and the WASO index,

**Fig. 4 | Increased lipid synthesis in *Cyfip^{85.1/+}* is due to the upregulation of SREBP activity.** Venn diagram showing (**a**) the overlap between the DEGs in *Cyfip^{85.1/+}* and DEGs in SREBP-OE[69–71]. $p = 0.98$, (**b**) the overlap between the wakefulness-associated genes (WAGs) differentially expressed in *Cyfip^{85.1/+}* and in *SREBP-OE*. $p = 3.467^{-21}$, (**a–b**) one-sided hypergeometric test. **c** RT-qPCR in control and *Cyfip^{85.1/+}* flies at ZT16 normalized to *rpl32* and *rpl13* mRNA. *FASN1* **$p = 0.007$*, *FASN3* **$p = 0.0023$*, *ACC* *$p = 0.0123$*, *Lsd-1* *$p = 0.0167$*, *AcCoAS* ***$p = 0.0006$*, *Men* **$p = 0.0089$*, control and *Cyfip^{85.1/+}* ($n = 4$). *ATPCL* *$p = 0.0127$*, control ($n = 4$) and *Cyfip^{85.1/+}* ($n = 3$), $n$ = pool of 15 fly heads. Mean ± S.E.M. **d** Representative blots and quantification of precursor and active SREBP, in controls and *Cyfip^{85.1/+}*, between ZT12-ZT16, normalized over syntaxin and coomassie. SREBP-p, ****$p < 0.0001$ (control and *Cyfip^{85.1/+}*, $n = 20$), SREBP-c, **$p = 0.0029$. Control ($n = 19$) and *Cyfip^{85.1/+}* ($n = 20$), pool of 15 fly heads. Mean ± S.E.M. **e** Schematics of the *Gal4-SREBP::GFP* reporter and its activation. **f** GFP relative intensity, in control ($n = 12$) and *Cyfip^{85.1/+}* ($n = 23$) brains. **$p = 0.0017$. Mean ± S.E.M. Scale bar = 50 μm. Maximum intensity Z-projections and magnification of the region marked by the square. **g** Night sleep in *Ras2Gal4;TubGal80^{ts/+}* ($n = 20$), *UAS-Srebp^{wt/+}* ($n = 19$), *UAS-Srebp^{c.del/+}* ($n = 19$), *Ras2Gal4;TubGal80^{ts} > UAS-Srebp^{wt}* ($n = 24$) and *Ras2Gal4;TubGal80^{ts} > UAS-Srebp^{c.del}* ($n = 24$) flies. One-way ANOVA and Sidak's multiple comparisons test, ****$p < 0.0001$. Mean ± S.E.M. **h** Night sleep in controls ($n = 29$), *Cyfip^{85.1/+}* ($n = 36$) and *Cyfip^{85.1}/Srebp^{189}* ($n = 32$) flies. Kruskal–Wallis and Dunn's multiple comparisons test, ****$p < 0.0001$. Mean ± S.E.M. **i** Sleep rebound after 6 h of SD in controls ($n = 30$), *Cyfip^{85.1/+}* ($n = 25$) and *Cyfip^{85.1}/Srebp^{189}* ($n = 32$). One-way ANOVA, Sidak's multiple comparisons test; control vs. *Cyfip^{85.1/+}* *$p = 0.019$; *Cyfip^{85.1/+}* vs. *Cyfip^{85.1}/Srebp^{189}* *$p = 0.027$. Mean ± S.E.M. **j** Night sleep in control ($n = 20$) and *Cyfip^{85.1/+}* ($n = 25$) vehicle vs. control ($n = 17$) and *Cyfip^{85.1/+}* ($n = 25$) betulin treated flies. Two-way ANOVA and Tukey's test, genotype-treatment interaction $p < 0.0001$, $F_{(1, 83)} = 18.25$; ****$p < 0.0001$. Mean ± S.E.M. Two-tailed unpaired Student's $t$ test in (**c**, **d** and **f**). **h**, **i** n.s. = not significant. Source data are provided as a Source Data file.

namely the sleep quality (Supplementary Fig. 4c, d). Thus, increased levels and/or activity of SREBP in neurons affect sleep behavior in flies, leading to the exacerbation of an insomnia-like phenotype. Notably, these data link SREBP to sleep behavior.

To explore whether the increase in SREBP mediates the sleep disturbances in *Cyfip* mutant flies, we evaluated sleep in *Cyfip^{85.1}* flies crossed with *Srebp* loss-of-function mutants (*Srebp^{189/+}*[75]). *Cyfip^{85.1}/ Srebp^{189}* flies showed a restoration of the amount of night-time sleep (Fig. 4h and Supplementary Fig. 4e), improved sleep quality (Supplementary Fig. 4f, g) and sleep rebound after sleep deprivation (Fig. 4i) while no effect on day sleep was observed. Furthermore, pharmacological reduction of SREBP activity using betulin (1 mM for 48 h), an inhibitor of SREBP activation[76], also increased the night-time sleep in *Cyfip* mutants compared to untreated flies (Fig. 4j and Supplementary Fig. 5a, b). Note that vehicle (DMSO in blue food) exacerbates sleep deficits (Supplementary Fig. 5b). Importantly, such a treatment with betulin did not have any genotype-specific effect on food consumption (Supplementary Fig. 5c). The pharmacological and genetic dampening of SREBP activity in *Cyfip^{85.1}* flies reveals that SREBP modulation is relevant for proper night-time sleep amount and quality, and that increases in SREBP levels underlie the sleep disturbances seen in *Cyfip^{85.1}* flies.

### The SREBP−Malic enzyme axis affects sleep

Thus far, we have demonstrated a role for SREBP in sleep homeostasis in flies, and that NADP+/NADPH ratio oscillations are altered in *Cyfip^{85.1/+}* flies. HILIC-MS/MS revealed that the *Cyfip^{85.1}/Srebp^{189}* double mutant partially restored the NADP+/NADPH ratio at night-time onset (ZT12) (Fig. 5a). As SREBP controls *Men* levels[77,78] and *Men* is a key contributor to the NADPH pool[66,67], we hypothesized that the SREBP effects on sleep in *Cyfip^{85.1/+}* might occur via *Men*-mediated regulation of the NADP+/NADPH. Wild-type flies undergoing sleep deprivation showed an increased *Men* mRNA expression (Supplementary Fig. 5d), in agreement with previous observations[13]. No difference in *Cyfip* or *Srebp* mRNA expression was observed (Supplementary Fig. 5e, f). Of note genetic reduction of *Men* in *Cyfip^{85.1/+}* flies (*Cyfip^{85.1}/Men^{BG02790}*) was able to enhance the NADP+/NADPH ratio at control levels at night-time onset (Fig. 5a). Consistently, we observed increased Men activity in *Cyfip^{85.1/+}* brains compared to controls at ZT10, before the night-time onset and Men activity was increased to control levels in the *Cyfip^{85.1}/ Srebp^{189}* mutants (Fig. 5b). Finally, when Men activity is genetically reduced in the *Cyfip^{85.1/+}* flies, loss of night-time sleep is ameliorated (Fig. 5c and Supplementary Fig. 5g).

Interestingly, *Men^{BG02790/+}* flies showed an increase in night sleep amount and in the NADP+/NADPH ratio at night-time onset (Fig. 5a, c and Supplementary Fig. 5g), demonstrating the property of Men to affect NADP levels and sleep behavior.

Notably, despite the increase in *Men* transcript upon sleep deprivation in control flies, we did not observe changes in Men activity

(Supplementary Fig. 5h). On the contrary Men activity was increased, upon sleep deprivation, in *Cyfip* heterozygous flies (Supplementary Fig. 5i). This is in line with the previously observed deficits in the NADP+/NADPH ratio after sleep deprivation in *Cyfip^{85.1/+}* flies (Fig. 2b) and further underlines that the pathological upregulation of Men drives changes in the NADP+/NADPH ratio and sleep deficits of *Cyfip* heterozygous flies.

Altogether, these results demonstrate that the observed sleep deficits in the *Cyfip^{85.1/+}* flies are mediated by the SREBP-Men-NADPH axis and show that SREBP activity and NADPH balance set the level of sleep amount at sleep onset.

## Discussion

Using a *Drosophila* model for the human *CYFIP1* haploinsufficiency (*Cyfip^{85.1/+}* mutant flies) a hotspot genetic variant increasing the risk for SCZ and a candidate risk factor for ASD, we describe a not yet identified molecular mechanism for sleep disturbances.

Our findings uncover four key aspects in the regulation of sleep: (1) SREBP levels are altered in *Cyfip* mutant flies; (2) SREBP activity on wakefulness genes such as malic enzyme regulates night-time sleep onset and amount; (3) the NADP+/NADPH are regulated by the SREBP/Men axis; (4) specific modulations of this metabolic pathway ameliorate the observed sleep deficits (Figs. 4, 5) suggesting this pathway could be explored for the development of sleep disorder treatments.

Specifically, our data show that increased activity of the sterol regulatory element-binding protein (SREBP) due to *Cyfip* heterozygosity (Fig. 4) induces an imbalance of the NADP+/NADPH ratio in the brain specifically at night, mediated by the overactivity of the malic enzyme (Fig. 5), resulting in deficits in sleep onset and amount (Fig. 1 and model in Fig. 5d).

Notably, night-time and daytime sleep features differ in flies[79]. Brain activity, measured as local field potential (LFP) and responsiveness to stimuli during night compared to daytime sleep, suggest different levels of homeostatic pressure on these processes throughout the 24 h[80,81]. While these differences have been reported, the function of daytime and night-time sleep in *Drosophila* remains unclear, although some hypotheses point to memory consolidation and synaptic homeostasis[80,81]. Therefore, the identification of factors, such as CYFIP, regulating differentially daytime and night-time sleep helps to shed light on the characteristics of these temporally distanced sleep. Only few genetic factors and mechanisms have been so far identified as regulators of daytime and night-time sleep, for instance, hormones, immune response, oxidative stress, and lipid binding molecules[79]. Here we show that at night-time *Cyfip^{85.1/+}* flies revealed a dysregulation in transcripts regulating carbohydrate metabolism, monooxygenases (*i.e.*, cytochrome P450), retinol, and lipid and energy metabolism among others, suggesting a convergence towards metabolic processes (Fig. 3). Of note, in a previous study, using unbiased proteomics in *Cyfip^{85.1/+}* fly brain, we detected an upregulation of the

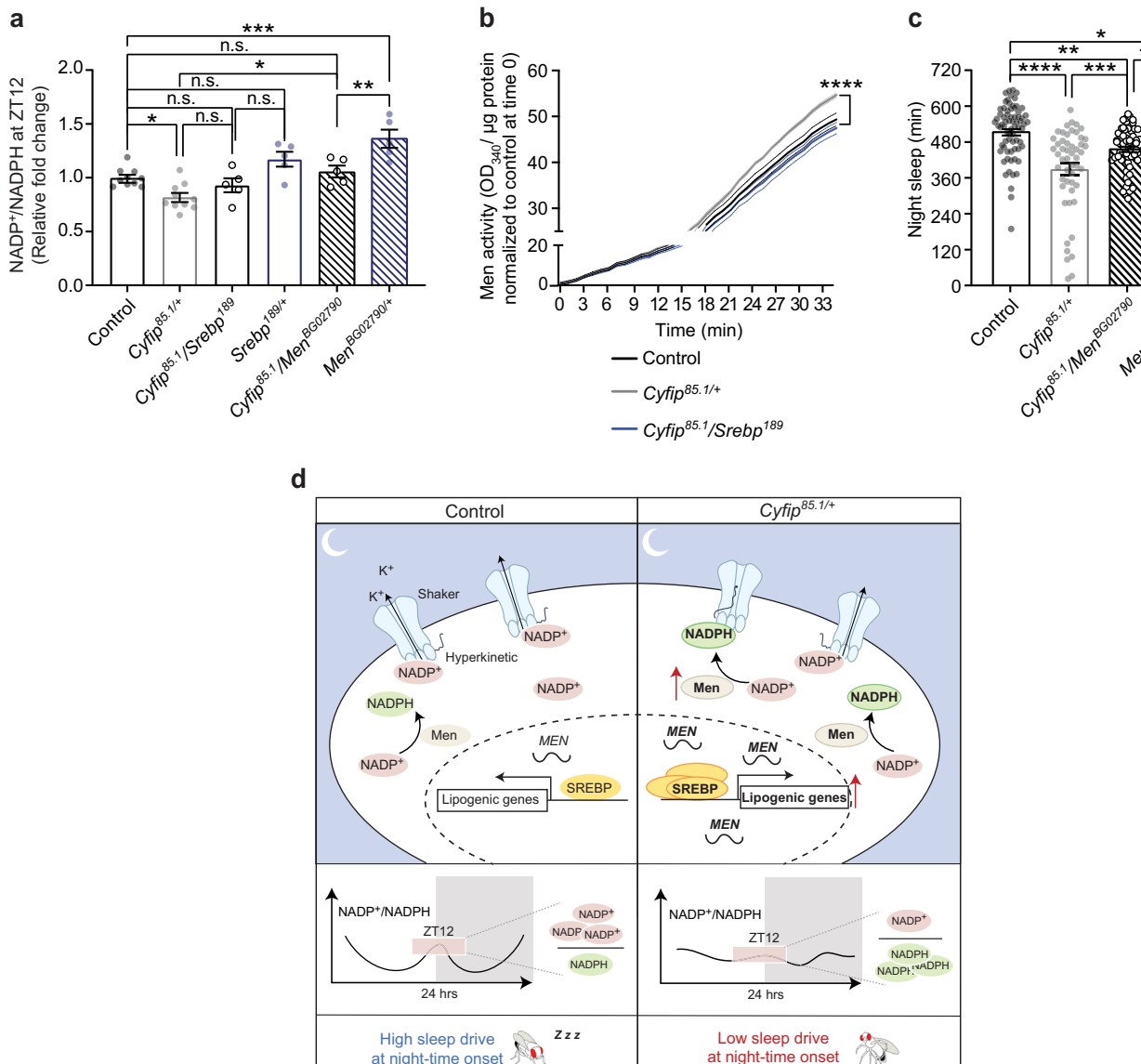

**Fig. 5 | Men enzyme activity pairs SREBP to sleep. a** NADP⁺/NADPH ratio at ZT12 by HILIC-MS/MS in control ($n = 10$), $Cyfip^{85.1/+}$ ($n = 10$), $Cyfip^{85.1}/Srebp^{189}$ ($n = 5$), $Srebp^{189/+}$ ($n = 5$), $Cyfip^{85.1}/Men^{BG02790}$ ($n = 5$), $Men^{BG02790/+}$ ($n = 5$). $n$ = independent samples, pool of 45 fly heads. One-way ANOVA test and Sidak's multiple comparisons test, control vs. $Cyfip^{85.1/+}$ *$p = 0.0359$; $Cyfip^{85.1/+}$ vs. $Cyfip^{85.1}/Men^{BG0279}$ *$p = 0.0188$; $Cyfip^{85.1}/Men^{BG02790}$ vs. $Men^{BG02790}$ **$p = 0.0075$; control vs. $Men^{BG02790+}$ ***$p = 0.0001$. Mean ± S.E.M. **b** Men activity at ZT10, in control ($n = 5$), $Cyfip^{85.1/+}$ ($n = 6$) and $Cyfip^{85.1}/Srebp^{189}$ ($n = 6$). $n$ = independent samples, pool of 10 fly heads. Two-way repeated measures ANOVA and Tukey's multiple comparisons test, time effect

****$p < 0.0001$, genotype effect *$p = 0.0137$, interaction ****$p < 0.0001$, $F_{(138, 966)} = 11.14$; control vs. $Cyfip^{85.1/+}$ *$p = 0.0304$; $Cyfip^{85.1/+}$ vs. $Cyfip^{85.1}/Srebp^{189}$ **$p = 0.0026$; control vs. $Cyfip^{85.1}/Srebp^{189}$ $p = 0.57$, n.s. = not significant. Mean ± S.E.M. **c** Quantification of night-time sleep in control ($n = 71$), $Cyfip^{85.1/+}$ ($n = 51$), $Cyfip^{85.1}/Men^{BG02790}$ ($n = 60$) and $Men^{BG02790/+}$ ($n = 35$) flies. One-way ANOVA and Sidak's multiple comparisons test, control vs. $Men^{BG02790/+}$ *$p = 0.046$; $Cyfip^{85.1/+}$ vs. $Cyfip^{85.1}/Men^{BG0279}$ ***$p = 0.0007$; control vs. $Cyfip^{85.1}/Men^{BG0279}$ **$p = 0.0031$; control vs. $Cyfip^{85.1/+}$ and $Cyfip^{85.1}/Men^{BG0279}$ vs. $Men^{BG02790/+}$ ****$p < 0.0001$. Mean ± S.E.M. Source data are provided as a Source Data file. **d** Model.

ATP-Citrate Lyase, mitochondrial Men and Fatty-Acid Binding Protein (FABP)[43], strengthening a role for CYFIP in metabolic regulation.

Sleep restriction and sleep deprivation affect lipid gene expression and lipid profiles in murine and humans, supporting the idea that lipid synthesis and remodeling are relevant during sleep to restore the metabolic pool used during wakefulness supporting the organism physiology and neuronal activity[82–86]. We propose that increased expression of genes involved in energy metabolism and lipid synthesis in the *Cyfip* mutant reduces the homeostatic pressure at the night onset affecting the homeostatic sleep regulation (Fig. 5d).

SREBP is a central regulator of lipid biosynthesis and remodeling in many cell types[54]. SREBP functions have been studied primarily in lipogenic tissues (i.e., liver and adipose tissue)[87], but have remained

largely unexplored in the brain. Because the blood-brain barrier limits the transport of lipids in the brain, de novo synthesis of lipids in brain tissue is essential, and such a mechanism might require a specific regulatory process during wakefulness and sleep. Lipid homeostasis is essential for brain[88], neuronal outgrowth[73,89,90], myelination[91], signaling, and energy supply[92,93]. Only recently, SREBP was shown to regulate lipid membrane biosynthesis, neuronal progenitor development[94], synaptic genes expression, vesicle pool recycling, dendrite growth and development[73,90] and membrane function[69].

Here we show that modulation of SREBP is relevant to set the proper night-time sleep (Fig. 4) and that SREBP activity influences sleep onset through the modulation of the NADP⁺/NADPH level. It was recently discovered that the binding of NADP⁺ or NADPH to the Shaker

Kv channel complex and its β-subunit Hyperkinetic can inactivate or activate Kv currents regulating neuronal excitability[95]. Thus, the NADP$^+$/NADPH ratio in specific sleep-inducing circuits (i.e., ExFl2) reflects neurons' sleep pressure; when the NADPH cofactor is bound to Shaker, its activity is inhibited and wakefulness promoted, while the sleep switch is promoted by NADP$^+$ binding[18]. Our data show that over 24 h the NADP$^+$/NADPH ratio in the whole brain increases at moments of intense loco-motion/wakefulness and/or after sleep deprivation and decreases when sleep behavior is more solid with the reduction of sleep need. NADP fluctuations are therefore relevant to promote the night-time sleep onset, through its effect on sleep-promoting neuronal circuits. The accumulation of the NADP$^+$ over NADPH during sleep deprivation in wild-type flies, suggests its involvement in the homeostatic sleep regulation (Fig. 2). We cannot exclude that NADP$^+$/NADPH changes might be related to increased locomotor activity: periods of motor/wakefulness increase sleep need. However, in support of our model, a mutation that impacts the NADP$^+$/NADPH ratio alters sleep behavior (Fig. 5).

Here, we propose that the driving force of the sleep deficits in Cyfip mutants is the dysregulation of the NADP$^+$/NADPH ratio (Fig. 2). Consistently, in Cyfip heterozygous flies, SREBP-driven over-expression and over-activity of the malic enzyme induce a reduced accumulation of NADP$^+$ at ZT12, possibly maintaining the sleep-regulating circuit OFF (Kv channels inhibited) and explaining the prolonged wakefulness at night-time onset (Fig. 5d).

Notably, a key player in regulating the NADP$^+$/NADPH balance is the malic enzyme, which catalyzes the conversion of cytosolic malate to pyruvate, reducing NADP$^+$ at NADPH. Men expression and activity is pathologically upregulated in Cyfip$^{85.1/+}$ flies, resulting in the impaired NADP$^+$/NADPH and sleep. Like Men, dysregulation of other enzymes involved in NADP$^+$/NADPH homeostasis might contribute to sleep disorders.

Sleep-wake disorders can have a negative impact on cognition, social function, mood, and quality of life in people and exacerbate the behavioral deficits in people suffering from psychiatric disorders[26]. Mutations in the human CYFIP1 gene have been linked to neurobehavioral disturbances, epilepsy, and psychiatric problems, emerging as risk factor for ASD and SCZ. Cyfip mutant flies exhibited sleep onset problems and low sleep rebound, a sign of deficits in homeostatic sleep regulation (Fig. 1), a feature also recently reported in children with ASD[96]. In addition, Cyfip heterozygous flies exhibit impaired associative learning[30], a hallmark of sleep loss[97–100], suggesting that sleep disorders and/or chronic sleep deprivation might contribute and exacerbate the cognitive defects reported in Cyfip mutants.

We demonstrate that SREBP over-activation mediates the sleep deficits in the Cyfip heterozygous flies. Interestingly, the KO mouse model for SREBP1c displays SCZ-like behaviors and GABAergic hypofunction[101], also commonly found in ASD. In addition, genome-wide association studies[102] classified the human SREBF1 gene (homolog of the Drosophila Srebp) as a risk factor for SCZ, and single nucleotide variants in the DNA binding sites recognized by SREBF1 are associated with ASD[103]. Finally, the human SREBF1 gene has also been associated with excessive daytime sleepiness (EDS), a symptom of chronic insufficient sleep[104]. SREBP-Men inducing NADP$^+$/NADPH alterations might therefore represent a common mechanism conserved in flies and mammals that contribute to a wide range of neurodevelopmental and neuropsychiatric disorders, including ASD and SCZ. Remarkably, alterations in NADP$^+$/NADPH levels are observed in plasma of children with ASD[105].

## Methods
### Fly stocks and genetics
Flies were maintained on standard cornmeal fly food at 25 °C, 60–80% relative humidity, in a 12-h light/dark cycle. Flies were trained to 12 h light/dark condition since the larval stage. Males 5–7 days after eclosion were used for all the experiments. The fly line used as control is a wild-type Canton-S $w^{1118}$ (BDSC 3605). The following stocks were previously described and kindly provided: Cyfip$^{85.1}$ and UAS-Cyfip-IR[1] by A. Giangrande (IGBMC, France); Ras2Gal4;TubGal80$^{ts}$, NrvGal4 and Gad1Gal4 by E.M.C. Skoulakis (BSRC, Greece); R23E10Gal4 by G. Miesenböck (CNCB, United Kingdom). RepoGal4 by L. Neukomm (Unil, Switzerland); ClkGal4_4.1 M, Mai179Gal4 and CryGal4_39 by F. Rouyer (CNRS, France); APLGal4 (VTO43924-GAL4) by A. Fiala (Univ. Göttingen, Germany). The following fly lines were obtained from the Bloomington Stock Center (BDSC): Srebp$^{189}$ (39396), GAL4-dSREBPg.K-B31:UAS-GFP ('SREBP activation reporter line', 39612), UAS-Srebp$^{WT}$ (8236), UAS-Srebp$^{c.del}$ (constitutively active Srebp, 8244), Men$^{BG02790}$ (12824), Aralar$^{MI07SS2}$ (43727), PdfGal4 (6900), c819Gal4 (30849), c767Gal4 (30848) and Herm3xP3-ECFP,alphatub-piggy-BacK10_M6 (32070). The UAS-Cyfip-IR[2] (37908) line was obtained from the Vienna Drosophila Resource Center. All transgenic lines were either generated in the $w^{1118}$ background or backcrossed 6 times into the $w^{1118}$ background. Temperature-induced experiments were performed as follows: UAS-Cyfip-IR flies expressing Ras2Gal4;TubGal80$^{ts}$ and respective controls were raised at 18 °C throughout development to avoid the lethal developmental effect described for CYFIP abrogation. The TubGal80$^{ts}$ was induced at 29 °C, 3–5 days after eclosion, for 3 days, allowing a strong Gal4 induction. From day 4 onwards flies were assessed for sleep behavior at 29 °C or used for RNA extraction. To test the expected lack of expression of the UAS-Cyfip-IR construct in flies expressing TubGal80$^{ts}$ at 18 °C, Ras2Gal4;TubGal80$^{ts}$ > UAS-Cyfip-IR flies and respective controls were raised, maintained, and tested at 18 °C. For the behavioral experiments involving UAS-Srebp$^{wt}$ and UAS-Srebp$^{c.del}$, Ras2Gal4;TubGal80$^{ts}$ > UAS-Srebp$^{wt}$ or UAS-Srebp$^{c.del}$ and respective control flies were raised at 25 °C throughout development, maintained and tested 5 to 7 days after eclosion at 25 °C.

### CRISPR/Cas9 generation of the CYFIP-HA tag fly strain
Scarless genome editing approach based on the PiggyBac transposon (PBac)[106] was used to generate the Drosophila CYFIP$^{N-HA}$ tag.

pBS-DsRed-attp-Cyfip N term-HA wSL cloning (donor plasmid): the 2614 bp fragment spanning the Cyfip gene (CG4931) start codon was PCR amplified using genomic DNA from nos-Cas9 (BDSC 78781) as a template and specific primers (5′-GAGCTCGAGCACGCCTCCGGTGATAGTAC −3′ and 5′-ACATCTAGACGGCATCAGAGGGATTCTCC-3′). The amplified fragment was cloned into the pBS-SK vector using the XhoI/XbaI sites. The resulting plasmid was used to amplify the left and right homology arms by PCR with the following primers (5′-GAGCTCGAGCACGCCTCCGGTGATAGTAC-3′ and 5′-CAATATGATTATCTTTCTAGGGTTAAATTCTAACAAGGCGGGGC-3′; 5′-GAGAAGATTACGCTAGCCGACG-3′ and 5′-AGAACTAGTCGGCATCAGAGGGATTCTCC-3′). The HA tag assembled from the primers containing the HA sequence: (5′-GCAGACTATCTTTCTAGGGTTAACCATTTCGCGCTGCTGGAGCATGATAAGCAGCCCAGCATGTACCC-3′ and 5′-CGTCGGCTAGCGTAATCTTCTCgGTGGCGTAATCGGGCACATCGTAGGGGTACATGCTGGGCTGCTTA-3′). Note that a silent mutation (g) was introduced in the protospacer adjacent motif (PAM) sequence to prevent cutting by the Cas9. The 3xP3-DsRed marker cassette flanked by TTAA target sequences was amplified from the pHD-sfGFP-ScarlessDsRed plasmid (DGRC stock 1365, kindly provided by B. McCabe, EPFL) with primers (5′-TTAACCCTAGAAAGATAATCATATTG-3′ and 5′-TTAACCCTAGAAA-GATAGTCTGCG-3′). The generated fragments (3xP3-DsRed, left and right homology arms, and HA tag) were assembled using the Gibson assembly cloning standard protocols (New England Biolabs) and cloned into the pBS-SK vector, generating the donor plasmid.

pCFD3-dU6-Cyfip gRNA cloning (gRNA plasmid): a sgRNA, 5′-TGATAAGCAGCCCAGCATG-3′, was designed using a on-line tool (https://www.crisprscan.org), synthesized as double-stranded oligonucleotide (5′-GTCGTGATAAGCAGCCCAGCATGA-3′ and 5′-AAACTCATGCTGGGCTGCTTATCA-3′), and cloned into the pCFD3-dU6:3gRNA vector (Addgene 49411)[107]. Donor and gRNA plasmids were injected into

*nos-Cas9* (II Chr) (BDSC 78781) embryos (*BestGene Inc*. transgenic service). CRISPR homology DNA repair (HDR) transformants were screened based on the DsRed expression. HDR-positive stocks were then crossed with the PBac-transposase expressing fly stock (BDSC 32070) to induce the DsRed cassette excision. The HA insertion at the N-terminus of the *Cyfip* gene was confirmed amplifying (by PCR) the region surrounding the HA insertion in *CYFIP^{N-HA/+}* flies (Supplementary Fig. 1j). WB analysis detected the HA tag above 130 kDa, in agreement with the molecular weight of the recombinant HA-CYFIP (Supplementary Fig. 1k). The *CYFIP^{N-HA}* tag flies were backcrossed with the *w^{1118}* strain.

## Sleep-Wake activity assays

Sleep assays were performed as previously described[57]. 5-7-day-old male flies raised in light/dark (LD)-entrained cultures were individually placed in 65 × 5 mm glass tubes containing 5% sucrose and 2% agar. Experiments were performed in an incubator with 60% humidity. Light was turned on at Zeitgeber Time 0 (ZT0) and off at ZT12. Fly activity was recorded for at least 3 consecutive days using the *Drosophila* Activity Monitoring (DAM) system (Trikinetics, Inc.). Activity records were collected in 1 min bins and analyzed using a custom-written R script (code available at: https://github.com/adrianclo/dam3). Sleep was defined as 5 consecutive minutes of inactivity. The sleep parameters were calculated as follows: Night-time sleep, the amount of sleep between ZT12-ZT24; Day-time sleep, the amount of sleep between ZT0 and ZT12; Night-time sleep bout number and length, number, and length of sleep events between ZT12 and ZT24; Latency to sleep is defined as the amount of time to the first sleep episode following light off (ZT12); Wake After Sleep Onset (WASO) is defined as the amount of time the fly is active following the first sleep episode between ZT12 and ZT24; Consolidation Index (CI) results from summing the squares of all the sleep-bout lengths in minutes divided by the total amount of sleep between ZT12 and ZT24[57]. For sleep deprivation experiments, baseline sleep was recorded for 3 nights and then the flies were subjected to a sleep deprivation protocol using the Vortexer Mounting Plate (Trikinetics, Inc.) on the Multi-Tube Vortexer (VWR-2500). Sleep deprivation was carried out with the following sets: shaking for 2 s, every 60 s with a randomized protocol for 6 h (ZT12-ZT18) or 12 h. For each fly the Δ sleep was calculated as the amount of night-time sleep between ZT18 and ZT24 after deprivation (rebound sleep) minus amount of night sleep between ZT18 and ZT24 of the day before the deprivation (baseline sleep) for each fly. For 12 h of sleep deprivation, only flies exhibiting > 70% sleep loss throughout the night were used for the analysis. Arousal was tested similarly to[58,59]. In brief, flies were kept for 4 days at 12-h light/dark cycle. On the fourth night, a light pulse of 5 min was delivered at ZT16, ZT18, and ZT20. To analyze the effects of sleep perturbation by light stimuli at night, we calculated the sleep loss, at ZT16, ZT18 and ZT20, as follows: 100% − (night-time sleep after arousal/baseline night-time sleep) expressed as %. Flies with at least 2% of sleep loss were included in the analysis. Analysis of the awakenings was performed as described[108]: "not-responding" are flies with no activity 5 min before and 5 min after the stimulus and "responding" are flies with no activity 5 min before the stimulus and activity 5 min after the stimulus. "Awake flies", those that were moving in the 5 min preceding the stimulus, were excluded from the analysis. To evaluate rhythmicity, flies trained in 12:12 LD conditions were recorded for 4–5 days in 12:12 LD and then for 10 days in DD conditions. Actograms and periodicity were analyzed using the FaasX software (Rouyer & Boudinot, CNRF) (v. 1.21)[109]. Double-plotted actograms of average group activity were generated using the following settings: Light Cycle: LD_2400; Filter: Off; Hash Density: 08; Light on: ZT0 (08:30); Light off ZT12 (20:30); Fly survival: through the entire experiment. The periodicity was calculated using the chi-square periodogram method on at least 5 days in DD conditions, using the following settings: power > 120 and width > 1.5 h.

## Immunohistochemistry

Brain dissection of male flies was carried out as previously described[110]. Brains were dissected on ice-cold in 1X PBS for 30 min maximum, fixed in 4% formaldehyde for 20 min, and washed for 30 min with PBST (1X PBS and 0.3% Triton-X). Blocking was performed with 10% normal goat serum (Sigma, cat. G9023) for 1 h. Tissues were stained overnight with the following antibodies: anti-GFP (Invitrogen, cat. A-11122, RRID: AB_221569, 1:1000), anti-BRP (DHSB, clone nc82, RRID: AB_2314866, 1:1000), and anti-HA 11 epitope tag (BioLegend, clone 16B12, cat. 901501, RRID: AB_2801249, 1:1000). The following secondary antibodies conjugated with Alexa Fluor (Thermo Fisher Scientific) were used at 1:1000 for 2 h at RT: anti-rabbit IgG 488 (cat. A-11034, RRID: AB_2576217), anti-mouse IgG 488 (cat. A-11029, RRID: AB_2534088), anti-mouse IgG 546 (cat. A-11030, RRID: AB_2534089) and anti-mouse IgG 647 (cat. A-21236, RRID: AB_2535805). After washes, the brains were mounted on Mowiol 4−88 mounting medium and imaged with a Leica SP8 confocal microscope using a 40x NA 1.25 water immersion objective. Images were taken using the same confocal and laser intensity settings. For the experiments with the SREBP activation reporter line, brain dissections were performed at ZT12 and fluorescent intensity was analyzed using ImageJ software (NIH). Fluorescence intensity was analyzed based on the sum of the different Z-stack intensity projections quantifying a ROI of the same dimension in 3 brain areas with strong GFP expression (superior lateral protocerebrum; suboesophageal ganglion and the ventrolateral protocerebrum). To account for background intensity, a ROI of the same dimension outside the brain tissue was subtracted.

## Western blotting

Total protein extracts were prepared from fly heads and homogenized in RIPA buffer (150 mM NaCl, 50 mM Tris HCl, 1% Triton-X, 1% NaDoc, 1 mM EDTA, 1:100 Protease Inhibitor Cocktail from Sigma, 1:10 Phospho-Stop from Roche). Protein extracts were quantified using the Pierce™ BCA Protein Assay Kit (ThermoFisher, cat. 23225), separated by 10% SDS-Bis-Acrylamide gels and analyzed by Western blotting. The 10 kDa to 180 kDa PAGEruler pre stained Protein Ladder (Thermo-Fisher, cat. 26616) was used. The following antibodies were used: anti-SREBP (BD Biosciences, cat. 557036, RRID: AB_384985, 1:1000), anti-Syntaxin (DHSB, clone 8c3, RRID: AB_528484, 1:2000) and anti-HA 11 epitope tag (BioLegend, clone 16B12, RRID: AB_2801249, 1:1000) overnight at 4 °C. Membranes were incubated with DyLight (Thermo-Scientific) anti-mouse secondary antibodies 680- (cat. 35519, RRID: AB_1965956) and DyLight 800- (cat. SA5-10176, RRID: AB_2556756) at 1:5000 for 1 h and acquired using the Odyssey Infrared Imaging System (Li-COR Bioscience). Total protein staining (Li-COR Bioscience) or Coomassie staining was used for quantification. Signal intensity of the bands was quantified using ImageQuant (GE Healthcare) and each protein of interest was normalized for syntaxin and total protein content.

## RNA isolation and quantitative real time-PCR

Fly heads were quickly collected at the specific time point (ZT16) or during undisturbed sleep and after 8 h of sleep deprivation (no-SD and SD). Total RNA was extracted using Trizol Reagent (Gibco) according to the manufacturer's instructions. cDNA was prepared using the Superscript III first strand synthesis kit (Invitrogen, cat. 18080093) and random primers (Promega). RT-qPCR was performed on the iCycler iQ Real-Time PCR Detection System (Bio-Rad Laboratories) using SYBR Green-based detection (Roche, cat. 04887352001). Transcript levels were determined from threshold cycle values based on a standard curve derived from serial cDNA dilutions. Relative mRNA expression levels were normalized to *Rpl13* or *Rpl32* using the comparative ΔΔCT method. The primers used are listed in Supplementary Table 2.

## cDNA library generation and RNA-sequencing

Fly heads were quickly collected at ZT16 using liquid nitrogen. Total RNA was isolated using Trizol Reagent (Gibco) according to the manufacturer's instructions to extract total RNA. RNA quality was evaluated on a Fragment Analyzer (Agilent Technologies, Inc., Santa Clara, CA 95051, USA) and the RQN were between 8.5 and 9.7. RNA-seq libraries were prepared using 500 ng of total RNA and Illumina TruSeq Stranded mRNA reagents (Illumina; San Diego, California, USA) on a Sciclone liquid handling robot (PerkinElmer; Waltham, Massachusetts, USA) using a PerkinElmer-developed automated script. Cluster generation was performed with the resulting libraries using the Illumina HiSeq SR Cluster Kit v4 reagents and sequenced on the Illumina HiSeq 2500 using HiSeq SBS Kit v4 reagents. Sequencing data were demultiplexed using the bcl2fastq Conversion Software (v. 2.20, Illumina; San Diego, California, USA).

## RNA-sequencing data processing and analysis

Purity-filtered reads were adapter- and quality trimmed with Cutadapt (v. 1.8)[111]. Reads matching to ribosomal RNA sequences were removed with fastq_screen (v. 0.9.3). Remaining reads were further filtered for low complexity with reaper (v. 15-065)[112]. Reads were aligned against *Drosophila melanogaster* BDGP6.86 genome using STAR (v. 2.5.2b)[112] and the estimation of the isoforms abundance was computed using RSEM (v. 1.2.31)[113]. The number of read counts per gene locus was summarized with htseq-count (v. 0.6.1)[114] using *Drosophila melanogaster* BDGP6.86 gene annotation. Quality of the RNA-Seq data alignment was assessed using RSeQC (v. 2.3.7)[115]. Statistical analysis was performed for genes in R (R v. 3.3.2). Genes with low counts were filtered according to the rule of 1 count per million (cpm) in at least 1 sample. Library sizes were scaled using TMM normalization (EdgeR package v. 3.14.0)[116] and log-transformed with the limma voom function (Limma package version 3.28.21)[117]. Differential expression was computed with limma[118] by fitting all samples into a linear model. Next the comparison between *Cyfip*[85.1/+] vs control was performed. Moderated t-test was used for each contrast. The adjusted p-value was computed by the Benjamini-Hochberg method, controlling for false discovery rate (FDR or adj. *p*-value). Genes with an adjusted *p*-value ≤ 0.05 (FDR ≤ 0.05) and log2 fold change greater than 0.585 or smaller than −0.585 (|LFC| ≥ 0.585, 50% fold increase/33% decrease) were considered differentially expressed (DEGs). KEGG pathways analysis for the DEGs was performed using GeneTrail2 3.0 (https://genetrail2.bioinf.uni-sb.de)[119]. Over-representation analysis (ORA) was applied with the Benjamini-Hochberg false discovery rate test. Hypergeometric test was performed using R (https://stat.ethz.ch/R-manual/R-devel/library/stats/html/Hypergeometric.html).

## Protein-protein interaction (PPI) network analysis

Network analysis on the gene overlap between SREBP overexpression DEG databases and dysregulated *Cyfip*[85.1/+] genes at ZT16 was performed using the online bioinformatic tool STRING interactome (https://string-db.org) for PPI networks with a high confidence score cutoff and experimental evidence required. The PPI networks were graphed and visualized using Cytoscape 3 (RRID:SCR_003032).

## Metabolite extraction and protein quantification

*Drosophila* heads (45–50) were extracted and homogenized adding 150 μL of MeOH:H$_2$O (4:1), in the Precellys 24 sample Homogenizer (2 × 20 s at 10000 rpm, Bertin Technologies, Rockville, MD, US) with ceramic beads. The bead beater (Cryolys) was air-cooled down at a flow rate of 110 L/min at 6 bar. After centrifugation, the resulting supernatant was collected and transferred into LC vials for analysis. Protein pellets were evaporated and lysed in 20 mM Tris-HCl (pH 7.5), 4 M guanidine hydrochloride, 150 mM NaCl, 1 mM Na$_2$EDTA, 1 mM EGTA, 1% Triton, 2.5 mM sodium pyrophosphate, 1 mM beta-glycerophosphate, 1 mM Na$_3$VO$_4$, 1 μg/ml leupeptin using the Cryolys

Precellys 24 sample Homogenizer (2 × 20 s at 10000 rpm, Bertin Technologies, Rockville, MD, US) with ceramic beads. BCA Protein Assay Kit was used to measure total protein concentration.

## Hydrophilic interaction liquid chromatography coupled to tandem mass spectrometry (HILIC-MS/MS) analysis

Extracted samples were analyzed by HILIC - MS/MS[120,121] in negative ionization mode using a 6495 triple quadrupole system (QqQ) interfaced with 1290 UHPLC system (Agilent Technologies). Chromatographic separation was carried out in a SeQuant ZIC-pHILIC (100 mm, 2.1 mm I.D. and 5 μm particle size, Merck, Damstadt, Germany) column. The mobile phase was composed of $A$ = 20 mM ammonium acetate and 20 mM NH$_4$OH in water at pH 9.7 and B = 100% ACN. The linear gradient elution from 90% (0–1.5 min) to 50% B (8–11 min) down to 45% B (12–15 min). Finally, the initial chromatographic conditions were established as a post-run during 9 min for column re-equilibration. The flow rate was 300 μL/min, column temperature at 30 °C and the sample injection volume was 2 μl. ESI source conditions were set as follows: dry gas temperature 290 °C and flow 14 L/min, sheath gas temperature 350 °C, nebulizer 45 psi, and flow 12 L/min, nozzle voltage 0 V, and capillary voltage −2000 V. Dynamic Multiple Reaction Monitoring (dMRM) was used as acquisition mode with a total cycle time of 600 ms. Optimized collision energies for each metabolite were applied.

Pooled QC samples (representative of the entire sample set) were analyzed periodically (every 6 samples) throughout the overall analytical run in order to assess the quality of the data, correct the signal intensity drift (attenuation in most cases, that is inherent to LC-MS technique and MS detector due to sample interaction with the instrument over time) and remove the peaks with poor reproducibility (CV > 30%)[122,123]. In addition, a series of diluted quality controls (dQC) were prepared by dilution with methanol: 100% QC, 50%QC, 25%QC, 12.5%QC and 6.25%QC. Then, metabolites were selected also considering the linear response on the diluted QC series.

## Data processing

Raw LC-MS/MS data were processed using the Agilent Quantitative analysis software (v. B.07.00, MassHunter Agilent technologies). Relative quantification of the metabolites was based on EIC (Extracted Ion Chromatogram) areas for the monitored MRM transitions. The obtained tables (containing peak areas of detected metabolites across all samples) were exported to R software and signal intensity drift correction and noise filtering (using CV (QC features) > 30%, if necessary) were compiled within the MRM PROBS software[124].

## *Drosophila* betulin administration

Betulin (Sigma, Cat. B9757) was dissolved in DMSO. The solution was added to the Formula 4–24® Instant *Drosophila* Medium (blue food) in water to a final concentration of 1 mM. Male flies 5-7 days old were placed in plastic vials containing the blue food with betulin or vehicle and fed *ad libitum* for 48 h. Vehicle treatment consisted of solvent (DMSO) added to blue food.

## Food intake assay

A colorimetric estimation of food intake was performed as previously described[43]. To monitor food intake during the period ZT0-ZT12, groups of 5–7 days old male flies were transferred to food containing 2.5% (w/v) non-degradable dye (Erioglaucine blue disodium salt; Sigma-Aldrich, cat. 861146), 2% agar and 5% sucrose every 2 h from ZT0 to ZT12, left to feed ad libitum for 2 h then collected for food intake assessment. For betulin treatment, flies were transferred for 48 h on food containing 2.5% (w/v) erioglaucine blue, 2% agar and 5% sucrose added with water, DMSO, or 1 mM betulin dissolved in DMSO. After feeding period, individual flies were homogenized in 65 μl of chilled PBS 1X and centrifuged at 10.000 g for 1 min. Finally, 50 μl of

supernatant was used for absorbance recording at 620 nm using a spectrophotometer.

### NADP⁺/NADPH quantification

$NADP^+/NADPH$ ratio was measured from brain lysates using the $NADP^+$/NADPH Quantification Colorimetric Kit (Abcam, cat. ab65349) following the manufacturer's instructions. Colorimetric measurements were performed at 450 nm using a Hidex Sense 96-wells plate reader. 23 fly heads per sample were used.

### Malic enzyme assay

Enzymatic activity of the malic enzyme was performed as previously described[67]. Briefly, 10 heads per sample from 5 day old male flies, at ZT10 or after 8 h of sleep deprivation (no-SD vs. SD), were homogenized in freshly prepared grinding buffer (100 mM Tris-HCl, 0.15 mM $Na_2NADP$, pH 7.4) and centrifuged at 15000 $g$ 4 °C for 5 min to pellet the debris. 10 μl of extract was used for total protein quantification. Activity assay was performed in a 96-well plate using 10 μl of extract and 100 μl of freshly prepared buffer (100 mM Tris-HCl, 0.34 mM $Na_2NADP$, 50 mM $MnCl_2$, 50 mM malate, pH 7.4). Absorbance was measured every 30 s for 20 min at OD 340, 25 °C using a Hidex Sense Spectrophotometer (Labgene). Samples were analyzed in technical duplicates. The absorbance was normalized to the total protein content measured by the Pierce BCA Protein Assay Kit.

### Statistics

Statistical analyses were carried out using Prism (v. 8.0) and/or R. The statistical tests are listed in the respective figure legends and in Supplementary Data 6. The normal distribution of the data was assessed using the Shapiro-Wilk test prior to statistical analyses. For comparison between two independent groups, unpaired two-tailed Student's $t$ test or the Mann−Whitney test for non-parametric data were used. When comparing more than two independent groups, one-way analysis of variance (ANOVA) was used followed by Tukey's or Holm Sidak's multiple comparisons test or the non-parametric Kruskal−Wallis test followed by Dunn's post hoc test correction for multiple comparisons. Two-way analysis of variance (ANOVA) followed by Tukey's multiple comparisons test for non-repeated measurement data, two-way analysis of variance (ANOVA) with repeated measurement followed by Holm Sidak's or Tukey's post hoc test was used to examine two independent variables. Fisher's exact test and hypergeometric distribution analyses were assessed using R. For all analyses, $P$-values < 0.05 were considered significant and annotated as follows: $*p < 0.05$, $**p < 0.01$, $***p < 0.001$, $****p < 0.0001$. Results are presented as mean ± standard error of the mean (S.E.M.). Exact $P$-values for $*p < 0.05$, $**p < 0.01$ and $***p < 0.001$, $n$ of independent samples, or individual flies, or pool of flies (see figure legends), and descriptive statistics are reported in Supplementary Data 6.

### Reporting summary

Further information on research design is available in the Nature Portfolio Reporting Summary linked to this article.

## Data availability

All relevant data are available in the manuscript, in Supplementary Information files and/or are provided as Source Data. The RNA-Seq datasets generated in this study have been deposited in the GEO (NCBI) database under accession code GSE220524. Further information and requests for resources and reagents should be directed to and will be fulfilled by the Lead Contact, Claudia Bagni (claudia.bagni@unil.ch). There are restrictions to the availability of *Cyfip* mutant flies due to a signed MTA with Angela Giangrande (IGBMC, France). Source data are provided with this paper.

## Code availability

Codes used for sleep analysis are available at https://github.com/adrianclo/dam3.

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

## Acknowledgements

The authors thank A. Giangrande, G.Tavosanis, E.M.C. Skoulakis, G. Miesenböck, L. Neukomm, F. Rouyer, A. Fiala, and B. McCabe for fly stocks and reagents. We acknowledge the Developmental Hybridoma Studies Bank (DHSB) for antibodies, the Bloomington *Drosophila* Stock Center, the Vienna *Drosophila* Resource Center, BestGene Inc. injection service, Flybase and the DRSC/TRiP Functional Genomic Resources for information and resources. We are grateful to the Lausanne Genomic Technologies Facility (J. Weber and S. Calderon) for RNA-Sequencing, analysis and support. We are grateful to J. Ivanisevic, H. Gallart-Ayala and T. Teav at the Metabolomics Unit, University of Lausanne, for support, advice, and analysis of the metabolites. We thank A. Giangrande and all the members of the Bagni laboratory for discussions and suggestions. Special thanks to M. Midroit, M. Reinero, and G. Pedini. We thank A. Crevoisier for excellent administrative assistance and K. Dickson for helpful comments and editing of the manuscript. Illustrations were created using BioRender.com. This study was supported by SNFS 310030-182651, KUL Funds Opening the Future (OTF - Belgium), Novartis Foundation for medical-biological research (Switzerland), NCCR Synapsy 51NF40-158776 (Switzerland) to C.B., Autism Speaks Meixner Translational Postdoctoral Fellowship (Grant No 9728), Pierre Mercier Foundation and Autism Research Institute to A.K.K.

## Author contributions

V.M. and C.B. conceived the study and wrote the manuscript with inputs from all the authors; T.A. conceived some of the experiments and analyzed the data; V.M and A.K.K. designed the experiments; V.M. performed experiments and data analysis; G.A. performed some of the experiments and data analysis; A.C.L. designed the custom R script for sleep/wake behavior analysis and performed data analysis. E.L. provided scientific inputs during the entire development of the project. All co-authors provided feedback on the manuscript.

## Competing interests

The authors declare no competing interests.
