## [Peer Review File · Nature Communications]

SREBP modulates the NADP⁺/NADPH cycle to control night sleep in *Drosophila*REVIEWER COMMENTS

Reviewer #1 (Remarks to the Author):

Mariano et al, present an interesting article proposing a mechanism modulating sleep homeostasis, specifically night-time sleep. They show that upon activation of the sterol regulatory element-binding protein (SREBP) in a *Drosophila* model of neurodevelopment disorders, *Cyfp*, there is a transcriptional upregulation of genes known to be involved in lipid homeostasis. They focused on a couple of genes related to this, including malic enzyme (*Men*). An increase in *Men* activity is known to regulate the cycling of NADP⁺/NADPH levels, which have been linked in other work (from the Miesenbock lab) to sleep pressure. They demonstrate that alterations in this ratio, driven by lipid metabolism, are involved in regulating sleep pressure. In this way they have provided another link between the sleep state and the role metabolic changes have in controlling sleep processes.

This is a solid piece of work that contributes important new knowledge to this rapidly developing field. Additionally, there is a growing realisation that not all sleep is equivalent in flies, and the findings presented here point to a role for metabolic pathways in night-time sleep (as opposed to daytime sleep). The paper is overall well-written and easy to follow, and the data are largely consistent. There are some issues however that the authors should consider, as well as more minor suggestions that might need to be attended to.

Major concerns

The authors make a major case about molecular pathways regulating night sleep specifically (starting with their title), but after presenting some daytime sleep comparisons early on, they neglect to show daytime sleep results for many of their genetic and pharmacological manipulations (e.g., Fig 4e-h, extended data Figure 2). These data exists, so the authors should show it. The prediction would be no effects on day sleep, anywhere. If this is not uniformly the case in their experiments, they should speculate why not. More broadly, more discussion might be needed to help readers understand why daytime sleep would not be subject metabolic regulatory processes. The authors begin the paper by bringing up this interesting day/night sleep dichotomy, but after having established it in one mutant do little to return to this problem, which should be relevant to all of their conditions.

The link between *Men* expression and the NADP/NADPH pool may need to be further substantiated. The authors show that the ratios of NADP⁺/NADPH are altered in the *Cyfp* haploinsufficient flies and that these ratios are known to affect sleep pressure. Whilst the authors have shown genetic interactions between *Srebp* > *Cyfp* and *Men* > *Cyfp* they don't show the acute control of the NADP⁺/NADPH by *Men* in either of these backgrounds.

Regarding the Betulin drug provision experiments, the method of administration was monitored via consumption of the drug mixed into blue food. However, there is no quantification of how much food was digested, or if the drug was found in the (relevant areas of the) brain.

Figure 2b (the 24hr profile for NADP⁺/NADPH ratios, in mutants vs control) is key to this study. Unfortunately, the it is also one of the less believable graphs. The difference between the mutant and controls depends on one point, where significance seems marginal. That the 16hr timepoint happens to match the timepoint showing decreased nighttime sleep in the mutant could be viewed as coincidental. I wonder if there are additional approaches that could be taken to validate this result? Additionally, regarding the immunohistochemical experiments, were brain dissections done at a specific time of the day? If SREBP levels change throughout the course of 24 hours (peak expression at ZT12-ZT16) all dissections would need to be completed at a specific set time to account for circadian variation. Whether this was accounted for is unclear.

Starvation leads to sleep loss as well; this is well documented in the field, and is obviously related to metabolism homeostatic processes as well. Have the authors excluded the possibility that they are actually describing one aspect of that well-described phenomenon? In other words, all cells might

experience NADP⁺/NADPH ratio changes after extended wakefulness, and these might be reversed upon starvation conditions. Is it possible this has less to do with sleep than more general brain processes linked to hunger, with the mutant animals investigated here being simply more hungry at the start of the night (hence, not sleeping)? It's not clear that this has been ruled out or even considered.

Minor concerns and suggestions

Figure 1a - a Schematic would be helpful to the reader to follow experimental setup.

Figure 1b - Total max of nighttime sleep should be 720 minutes

Figure 3 and generally seq data - Further explain the rationale behind using a cutoff fold change of +/- 1.3 as the standard method is to use is Log₂FC of 0.58 = LogFold of 1.5. Whilst the authors do state that this was due to a FC of 1.3 of Cyfip gene this may need to be detailed more?

Figure 3b could be more informative, e.g. showing some raw seq reads.

Figure 4d - The authors have used fluorescent intensity and whilst confocal settings are kept consistent, how have they accounted for background intensity?

Extended figure 2d/e - When comparing the data of 2d and 2e some disparities are seen. Cyfip knockdown in RepoGal4 looks (*) more significant than in R23e10Gal4. Some differences in panel d seem as large as in e, yet none are significant in d. This seems suspect. Could this be due to the statistical testing as the former also includes the Nrv-Gal4 dataset, so simply a problem of multiple comparisons? Secondly The UAS-Cyfip-IR controls differ between the datasets. Finally, there is no quantification of these knockdowns.

Extended Figure 2F - Which UAS-Cyfip-IR line was used?

Methods: Sleep-Wake activity assays - light as an arousal stimulus is unusual, as it interferes with the circadian clock. This might need some discussion, to address concerns that circadian factors aren't at play here. It would also be interesting to know how many flies were not responding to this stimulus.

Fig. 5: in the concluding model (panel d) the authors seem to imply a daytime sleep effect, as well as a night time effect (increased NADP⁺/NADPH ratio leads to low sleep drive during both day and night). Yet, this is not what is proposed in the title and abstract and discussion. The authors might need to rethink their effects on daytime sleep, and this included showing all the daytime sleep data so that readers can see the extent to which day and night sleep correlate in these manipulations. This is critical, because without the word 'night' in the title, the story packs somewhat less of a punch.

Reviewer #2 (Remarks to the Author):

Mariano et al. use a previously established Drosophila model for neurodevelopmental disease to investigate the molecular mechanisms underlying sleep dysregulation in these disorders. They study flies that have a 50% reduction in Cyfip, a protein implicated in various neuropsychiatric disorders. Their data suggest that Cyfip inhibits SREBP which in turn upregulates malic enzyme. Thus, a reduction in Cyfip is proposed to enhance SREBP and malic enzyme levels, which then increase the NADPH/NADP ratio, resulting in a putative decrease in the metabolic drive for sleep. In general, the work is solid, especially the links between Cyfip, SREBP, and malic enzyme, and the functional connection between lipid metabolism and sleep regulation is interesting and potentially important. However, there are questions about whether the observed behavioral phenotypes are specifically related to sleep homeostasis, which could limit the novelty of the findings.

1. The sleep phenotypes observed may not reflect defects in sleep homeostasis. For example, in Figures 1D-1E, the data show that Cyfip hets are hyperaroused, which could account for the reduced baseline sleep and decreased recovery sleep after deprivation. Do levels or activity of Cyfip, malic enzyme, or SREBP vary according to sleep need?

2. It is not clear that Cyfip hets lack an increase of NADP/NADPH around dusk. The data appear to show an increase, but just earlier, from ZT8 to ZT12. Also, if the NADP/NADPH ratio reflects sleep need, then one would expect the peak to be ZT12, not ZT16, since animals would be sleeping from

ZT12-16, which should lower sleep need.

3. In *Cyfp* hets and in a number of other manipulations, there is a suggestion of a rightward shift of both daytime and nighttime sleep. It would be helpful if the authors performed locomotor analyses in constant darkness to assess whether there are underlying circadian defects.

4. Fig S2-- To support the notion that reduction of *Cyfp* levels affect sleep, the authors use 2 different RNAi lines crossed to *ras2-Gal4*. It is unclear why the authors chose here and in Figure 4 to use *ras2-Gal4*, as published literature suggests that it labels both non-neuronal and neuronal cells. These experiments should be repeated with *nsyb-Gal4* or *elav-Gal4*. Also, the authors should show the sleep profile traces for the experiments in Figure S2—does knockdown of *Cyfp* show a similar decrease in early night sleep seen in *Cyfp/+* flies?

5. Fig S2—they suggest that the effects of *Cyfp* knockdown on sleep can be localized to the *ExF12* neurons. To strengthen this potential mechanism, they should show knockdown of *SREBP* and malic enzyme in the *ExF12* neurons as well.

Minor comments:

1. Page 3, line 2, provide citation for sleep being conserved from worms to humans.

2. Figure 1E, it is unusual to examine recovery sleep after sleep deprivation during the night, when sleep is already high. The authors do examine recovery sleep during the daytime after 12 hr of sleep deprivation in Figure S2. Recommend moving the panel from Figure S2 to the main figures, and Figure 1E to the supplemental figures.

3. The authors should specify how many times the transgenic lines were backcrossed into their control background strain.

4. Can the authors comment on the marked expression of the *SREBP* reporter in ellipsoid body ring?

5. Genotypes are sometimes ambiguous. For example in Figure 5, *Cyfp/SREBP* animals—are these homozygous or heterozygous for the *SREBP* allele?

6. For *tubGal80ts* experiments, the authors state in the methods that they subject the flies to 29 deg for 3 days. Which days did they use for their baseline sleep recording?

7. Figure S2E—"R23E18" should be "R23E10"

7. The discussion is somewhat meandering. Recommend tightening this section.

Reviewer #3 (Remarks to the Author):

By characterising the sleep profile of *Cyfp* haploinsufficiency mutants, Mariano and coauthors identified a novel and original mechanism underlying the regulation of sleep homeostasis in flies. Using a series of complementary and elegant experiments, the authors convincingly find that the activity of the master regulator of lipogenesis *SREBP* is upregulated in *Cyfp* mutants, causing the upregulation of lipogenic genes, and, in turn, nighttime sleep defects via an imbalance of the *NADP⁺/NADPH* ratio. These findings are highly relevant to the field. I recommend publication in *Nature Communications* provided that the authors address the major and minor points raised below.

Major points:

1) Figure 2b. It looks like the *NADP⁺/NADPH* ratio oscillates in a circadian-dependent manner following the sleep pattern of the flies and not in a sleep need-dependent manner (as correctly stated in the abstract but not in the main text). Otherwise, why would the *NADP⁺/NADPH* start increasing when the flies are asleep, i.e. when sleep need is being dissipated, and not when sleep need is highest, i.e. at the end the 12-hour day period (ZT8-12)? Even though it is an interesting hypothesis, I do not think that the authors can causally link the changes in *NADP⁺/NADPH* ratio to the regulation of nighttime sleep and in particular sleep homeostasis. To address the latter, they would need to show that the *NADP⁺/NADPH* ratio is differentially modulated after sleep deprivation (by compared sleep-deprived versus rested flies) in WT versus mutant flies. In addition, the authors should carefully

rephrase the interpretation of the NADP⁺/NADPH ratio results.

2) Using a series of complementary and elegant experiments, the authors convincingly show that the master regulator of lipogenesis SREBP is unregulated in *Cyfp* mutants. What about the defects in sleep homeostasis shown in Extended Data Figure 2g and Figure 1e? Can they be rescued upon decreasing the expression of SREBP? Moreover, does neuronal SREBP expression block synaptic homeostasis? This should be addressed in the paper.

3) The images shown in Figure 4d are very difficult to quantify and to interpret due to the low signal-to-noise ratio. Looking at the pictures, it seems that the fluorescence signal is higher in WT than mutant 'ring neurons' for example (it is visible from the pictures that SREBP is active in a subpopulation of ring neurons). Yet, the 'background' (or neuropil or SREBP?) signal is much higher in mutant than WT brains and quantifying this signal without correcting for background or without attributing it to neuronal expression might lead to wrong conclusions. One way to circumvent this problem would simply be to stain for GFP (instead of doing live imaging) and additionally to counterstain the neuropil with anti-Bruchpilot for example. This would not only amplify the signal and improve the signal-to-noise ratio but it would also allow to do ratiometric imaging and to correct, e.g., for background noise or antibody penetration. Without these changes, the results shown in Figure 4d are not conclusive.

4) In Kanellopoulos et al, the same lab found a higher mitochondrial activity in the *Cyfp* mutant brain. This led to a redistribution of the GABA pool from synaptic vesicles into the mitochondria, thereby dampening GABAergic synaptic transmission, which, in turn, had severe effects on the social behaviour and locomotor activity of mutant flies. In the same paper, the authors also showed that the social deficits but not the locomotor activity could be rescued by decreasing mitochondrial activity or increasing GABA levels. I am surprised that the authors did not follow up on this finding in the context of sleep. It would be interesting to know whether similar manipulations as used in Kanellopoulos et al can also rescue the effects on nighttime sleep and sleep homeostasis observed in *Cyfp* mutants. If not, it might be very intriguing that the same protein might affect different types of behaviour (social behaviour and sleep) via entirely different metabolic mechanisms and possibly different neuronal substrates.

5) In Kanellopoulos et al, the same lab mapped the effect of *Cyfp* mutant protein onto specific GABAergic neurons. It would be interesting to know whether, e.g., knocking down *Cyfp* in these subsets of GABAergic neurons has an effect on sleep?

6) Page 8, line 12 & page 15, line 6: based on the sleep deprivation experiments and the NADP⁺/NADPH results, the authors propose that the mutants show a lower need for sleep. Yet, how do they reconcile this claim with the fact that the mutants show defects in associative learning (Woo et al (2019, *Biol Psychiatry*), a hallmark of cognitive defects due to sleep deprivation? Couldn't it also be that the sleep homeostasis machinery is defect, resulting in a reduction of homeostatic sleep?

Minor points:

1) Typo in Extended Data Figure 2e: third group from the left should be 'R23E10' instead of R23E18'.

2) In Extended Data Figure 2f, some, e.g., 'Mai179Gal4', but not all of the Gal4 names are written in italic characters. The same applies to other figures throughout the nomenclature. Could the authors homogenize the nomenclature?

3) Page 12, Line 24/25: does this imply that the malic enzyme is only over-active at ZT16 or regulated in a circadian-dependent manner? Based on the previously published proteomics results it doesn't seem to be the case. It would be informative if the authors would comment on that.

4) Page 14, line 20: 'Thus, the NADP⁺/NADPH ratio reflects neurons' sleep pressure..' This was only shown for dFB neurons in Kempf et al (2019). The authors should specify that.

5) Page 13, line 21/22: 'supporting the idea that lipid synthesis and remodeling is important during sleep, ..'. If so, then why do Cyfip mutants sleep less despite an upregulation of genes involved in lipid synthesis? What do the authors speculate?

Reviewer #4 (Remarks to the Author):

In this manuscript, Mariano et al provide an interesting and thorough description of a sleep phenotype associated with the Cyfip gene and its epistatic pathway. The authors show that cyfip mutant heterozygous flies have a short-sleep nocturnal phenotype and possible homeostatic misregulation and they propose that upregulation of SREBP and Men are responsible for this phenotype. The work features an impressive amount of work, generally well-controlled and well presented. I personally do not like the overly translational approach in the manuscript, insisting over and over again on possible links with human disease that may or may not exist; I think the manuscript ultimately suffers from adopting this approach and the findings end up being secondary but I recognise this may be the authors' style and they are entitled to use whichever style they prefer.

Overall, I think the work is solid and self-standing and I do not have any requests for further experiments. I only have some minor suggestions/requests mostly regarding data presentation and wording of the conclusions:

1. The term haploinsufficient refers to genes, not to animals. The use of "haploinsufficient flies" (e.g. line 3 pg5) is technically inappropriate. The flies are heterozygous for the gene mutation whilst the gene is haploinsufficient.

2. An EEG/MRI abnormality is not a "sleep disturbance" (line 12 pg 4). It may in some cases correlate with one but a change in EEG pattern is not a disturbance per se.

3. In the figures, please add more information regarding the number of animals used. Figure 1a-c, for instance, says the n is 75 without specifying the n for each condition. Is it 75 per condition or overall?

4. Replace all bar plots (such as the ones in fig 1bc,d) with something statistically more appropriate showing the actual distribution of data (as you do for instance in 1e).

5. Some plots have dots as values but no indication of what they represent. For instance, I assume the dots in the rightmost panel of 1e or in 4e represent outliers while the dots in 2a or 4c are actual data points? It would be good to stick to more consistent data visualisation.

6. Do you have any information regarding the expression pattern of Cyfip in the adult brain? Schenck et al characterised its expression during development but what about adulthood? The results in Fig. 4d suggest its expression (or at least its action on SREBP) is limited to the ellipsoid bodies and that is confirmed by the 23E10 KD experiment but do you know if the gene is expressed anywhere else? I am asking because of the results in fig.2 and fig 5b which do not seem really compatible with a very localised expression of the gene. If you do have data about cyfip expression please show it. If you do not have those data and do not want to obtain them please add some caveats/discussion regarding the results in fig. 2 and 5. I am just quite surprised that a change in enzymatic activities in only a few neurons can be picked up doing biochemical assays in the whole brain.

7. Please provide more details regarding the statistical analysis used to study the overlap of genes shown in fig. 3 and discussed on pg 9. A hypergeometric distribution is mentioned in the figure legend without further details. Was the same analysis done for the data in figure 4a? with what significance?

8. The epistatic experiments shown in fig. 5 are very nice but the only evidence that Men is involved with this process is coming from the one experiment shown in fig. 5c and 5b. While suggestive, I am not convinced this is enough evidence for such a definitive claim. I recommend using a more cautious approach in your conclusions.

9. On the same line: please amend the title to be more accurate in its description. "drives" and "control" are two very strong words for something that affects only a relatively small part of the phenotype.

Point by Point

Mariano et al. NCOMMS-21-46266-T

“SREBP drives the NADP⁺/NADPH cycle to control night sleep”.

We thank the reviewers for the constructive and helpful comments that allowed to strengthen the main messages of our paper. We have taken the precious advice into consideration, followed all their suggestions and provide new data in the revised version.

The revised manuscript contains **29 new panels, 7 figures for the reviewers only and 13 revised panels**. The new datasets conclusively show how the SREBP-MEN axis regulates NADP⁺/NADPH levels affecting sleep behaviour.

Below a list of the new and revised figures and answers to the reviewers.

New datasets with a major impact on our conclusions:

- **New Fig. 2:** NADP fluctuations set the level of night-time sleep and are regulated by homeostatic mechanisms. The assessment of the NADP⁺/NADPH ratio over the 24 hrs in control and *Cyfp* mutant flies using targeted metabolomic analysis strengthened our initial hypothesis and findings that NADP ratio fluctuations set the level of sleep amount at sleep onset. *We show for the first time in physiological conditions that the overall NADP⁺/NADPH in brain is regulated by sleep homeostasis and that this mechanism is affected in Cyfp heterozygous flies.*

- **New Fig. 5:** SREBP - Men axis regulates NADP⁺/NADPH levels: We show that abrogation of Men in the *Cyfp*^{85.1/+} background can fully rescue the NADP ratio deficits at ZT12 (panel a) as well the night-time sleep phenotype (panel c) further validating the link between NADP⁺/NADPH and Men. *We conclude that Men modulation affects NADP ratio.*

- **New Suppl. Fig. 2e, Suppl. Fig. 3d, Suppl. Fig. 4c, Suppl. Fig. 4g, Suppl. Fig. 5b and 5e.** We have analysed previously performed experiments and included the new analyses on the amount of daily sleep. *We strengthen the specificity of the metabolic SREBP-Men axis in the regulation of night-time sleep onset and need.*

Revised Figures:

Revised Fig. 1e: The previous Extended Data 2g is now part of this figure.

Revised Fig. 3: New dataset analysis in panels 3c and 3d. Revised panels 3e and 3f and relative tables (Suppl. Data 1, 2, 3, 4 and 5 and Suppl. Table 1).

Revised Fig. 4: New dataset analysis in panel 4a. New experiment in panel 4f.

Revised Fig. 5: Revised panels 5d.

Revised Suppl. Fig. 1: New experiment in panels 1h, 1i, 1j, 1k and 1l. New dataset analysis in panel 1g. New scheme in panel 1a.

Revised Suppl. Fig. 2: New experiment in panel 2a, new dataset analysis in panels 2c, 2e. Revised panels 2b and 2h.

Revised Suppl. Fig. 3: New experiments in panels 3a, 3b, 3e and 3f. New dataset analysis in panel 3d. Revised panels 3c, 3g, 3h.

Revised Suppl. Fig. 4: New dataset analysis in panels 4c and 4g. Revised panel 4a.

Revised Suppl. Fig. 5: New experiment in panels 5c and 5d. New dataset analysis in panels 5b and 5e.

Revised Suppl. Data 1, 2, 3 and 4: New dataset analysis.

Revised Suppl. Table 1: Previous Suppl. Table 5 is now Table 1

In addition, we provide

New Suppl. Data 5 and 6: dataset analysis on the RNA-seq and detailed statistics.

Reviewers' Comments:

Reviewer #1 (Remarks to the Author):

Mariano et al, present an interesting article proposing a mechanism modulating sleep homeostasis, specifically night-time sleep. They show that upon activation of the sterol regulatory element-binding protein (SREBP) in a *Drosophila* model of neurodevelopment disorders, *Cyfp*, there is a transcriptional upregulation of genes known to be involved in lipid homeostasis. They focused on a couple of genes related to this, including malic enzyme (*Men*). An increase in *Men* activity is known to regulate the cycling of NADP⁺/NADPH levels, which have been linked in other work (from the Miesenbock lab) to sleep pressure. They demonstrate that alterations in this ratio, driven by lipid metabolism, are involved in regulating sleep pressure. In this way they have provided another link between the sleep state and the role metabolic changes have in controlling sleep processes.

This is a solid piece of work that contributes important new knowledge to this rapidly developing field. Additionally, there is a growing realisation that not all sleep is equivalent in flies, and the findings presented here point to a role for metabolic pathways in night-time sleep (as opposed to daytime sleep). The paper is overall well-written and easy to follow, and the data are largely consistent. There are some issues however that the authors should consider, as well as more minor suggestions that might need to be attended to.

We appreciate the reviewer's positive comments on our work and performed the suggested experiments that helped to improve our manuscript. We hope that he/she is pleased with the new datasets included in the revised version of the manuscript.

Major points:

1) The authors make a major case about molecular pathways regulating night sleep specifically (starting with their title), but after presenting some daytime sleep comparisons early on, they neglect to show daytime sleep results for many of their genetic and pharmacological manipulations (e.g., Fig 4e-h, extended data Figure 2). These data exists, so the authors should show it. The prediction would be no effects on day sleep, anywhere. If this is not uniformly the case in their experiments, they should speculate why not. More broadly, more discussion might be needed to help readers understand why daytime sleep would not be subject metabolic regulatory processes. The authors begin the paper by bringing up this interesting day/night sleep dichotomy, but after having established it in one mutant do little to return to this problem, which should be relevant to all of their conditions.

We thank the reviewer for suggesting to further dissect the day/night sleep dichotomy. In the revised version of the paper, in addition to the analysis of the night-time sleep amount, we have determined the daytime sleep (Suppl. Fig.2e, Suppl. Fig. 3d, Suppl. Fig. 4c, Suppl. Fig. 4g, Suppl. Fig. 5b and e). We observe very little effect, if any, on the daytime sleep, strengthening the hypothesis that the metabolic deficits in *Cyfp* mutant flies have a specific effect on the night-time sleep onset.

Additionally, we added the following paragraph in the discussion:

"In flies, night-time differs from daytime sleep⁸⁰. The arousal threshold is higher at night compared to daytime, suggesting that night-time sleep might be associated with sleep homeostasis⁸⁰. Remarkably, only a few genetic factors and mechanisms have been so far identified as regulators of daytime and night-time sleep^{81"}.

Please find below the new data included in the revised manuscript.

Supplementary Fig. 2e, Day sleep amount in *Ras2Gal4;TubGal80^{ts/+}* (n = 48), *UAS-Cyfp-IR^{1/+}* (n = 15), *Ras2Gal4;TubGal80^{ts}>UAS-Cyfp-IR¹* (n = 18), *UAS-Cyfp-IR^{2/+}* (n = 21) and *Ras2Gal4;TubGal80^{ts}>UAS-Cyfp-IR²* (n = 31). One-way ANOVA, Sidak's multiple comparison test, **p* < 0.05, ***p* < 0.01, ****p* < 0.001 and *****p* < 0.0001, n.s. = not significant. Mean ± S.E.M. (see also Supplementary Data 6).

Supplementary Fig. 3d, Day sleep amount in *R23E10Gal4/+* (n = 34), *UAS-Cyfp-IR¹* (n = 37), *UAS-Cyfp-IR²* (n = 36), *R23E10Gal4>UAS-Cyfp-IR¹* (n = 39), *R23E10Gal4>UAS-Cyfp-IR²* (n = 44). One-way ANOVA, n.s. = not significant, mean ± S.E.M.

Supplementary Fig. 4c, Day sleep in *Ras2Gal4;TubGal80^{ts/+}* (n = 20), *UAS-Srebp^{wt/+}* (n = 19), *Ras2Gal4;TubGal80^{ts}>UAS-Srebp^{wt}* (n = 24), *UAS-Srebp^{c.del/+}* (n = 19) and *Ras2Gal4;TubGal80^{ts}>UAS-Srebp^{c.del}* (n = 24). One-way ANOVA followed by Sidak's multiple comparison test **p* = 0.0209, *****p* < 0.0001. (see also Supplementary Data 6). Mean ± S.E.M.

Supplementary Fig. 4g, Day sleep amount in control (n = 29), *Cyfip*^{85.1/+} (n = 36) and *Cyfip*^{85.1}/*Srebp*¹⁸⁹ (n = 32) flies. One-way ANOVA followed by Sidak's multiple comparisons test, n.s. = not significant. Mean ± S.E.M.

Supplementary Fig. 5b, Day sleep amount in control (n = 20) and *Cyfip*^{85.1/+} (n = 25) vehicle vs. control (n = 17) and *Cyfip*^{85.1/+} (n = 25) treated. Two-way ANOVA followed by Tukey's multiple comparison test, genotype effect, $p < 0.0001$, $F_{(1, 83)} = 13.57$. ** $p = 0.0012$, control vs *Cyfip*^{85.1/+}. Mean ± S.E.M.

Supplementary Fig. 5e, Day sleep amount in control (n = 71), *Cyfip*^{85.1/+} (n = 51), *Cyfip*^{85.1}/*Men*^{BG02790} (n = 60) and *Men*^{BG02790/+} (n = 36) flies. One-way ANOVA followed by Sidak's multiple comparisons test, * $p = 0.039$, control vs. *Men*^{BG02790/+}. Mean ± S.E.M.

2) The link between Men expression and the NADP/NADPH pool may need to be further substantiated. The authors show that the ratios of NADP⁺/NADPH are altered in the *Cyfip* haploinsufficient flies and that these ratios are known to affect sleep pressure. Whilst the authors have shown genetic interactions between *Srebp* > *Cyfip* and *Men* > *Cyfip* they don't show the acute control of the NADP⁺/NADPH by *Men* in either of these backgrounds.

We agree with reviewer's comment that is important to further validate the link between *Men* and NADP⁺/NADPH. We therefore performed an independent, more specific, and more sensitive assay: targeted mass spectrometry (Hydrophilic Interaction Liquid Chromatography coupled to tandem mass spectrometry, HILIC-MS/MS) to detect with the NADP ratio in *Men* and *Srebp* mutant flies.

The mass spec analysis in the *Cyfip*^{85.1}/*Srebp*¹⁸⁹ fly heads shows a partial rescue (increase) of the NADP⁺/NADPH ratio. The analysis in the *Srebp*^{189/+} fly heads shows an increase (tendency) of the NADP⁺/NADPH ratio ($p = 0.054$) indicating an effect of *Srebp* on this molecular pathway. We decided

to replace the former Fig. 5a obtained using the kit Colorimetric NADP⁺/NADPH kit from Abcam with the new dataset generated with the HILIC-MS/MS.

Importantly, Men mutants have an opposite phenotype of *Cyfp*^{85.1/+} flies, with an increase in the NADP⁺/NADPH ratio. In addition, reduction of Men in the *Cyfp*^{85.1/+} background can fully rescue the NADP ratio confirming its regulation by Men (new Fig. 5a).

Figure 5a, NADP⁺/NADPH ratio at ZT12 by HILIC-MS/MS in control (n = 10), *Cyfp*^{85.1/+} (n = 10), *Cyfp*^{85.1}/*Srebp*¹⁸⁹ (n = 5), *Srebp*^{189/+} (n = 5), *Cyfp*^{85.1}/*Men*^{BG02790} (n = 5), *Men*^{BG02790/+} (n = 5). One-way ANOVA test followed by Sidak's multiple comparisons test, **p* = 0.0359, control vs. *Cyfp*^{85.1/+}; **p* = 0.0188, *Cyfp*^{85.1/+} vs. *Cyfp*^{85.1}/*Men*^{BG02790}; ***p* = 0.0075, *Cyfp*^{85.1}/*Men*^{BG02790} vs. *Men*^{BG02790/+}; ****p* = 0.0001, control vs. *Men*^{BG02790/+}. n = group of 45 fly heads/ sample. Mean ± S.E.M.

3) Regarding the Betulin drug provision experiments, the method of administration was monitored via consumption of the drug mixed into blue food. However, there is no quantification of how much food was digested or if the drug was found in the relevant areas of the brain.

We agree with the reviewer that is difficult to know exactly the amount of food consumption. However, we succeeded to monitor and compare the consumption of food containing betulin (1mM for 48hrs), vehicle (DMSO) or water (alone) in control and *Cyfp*^{85.1/+} flies. Food intake was assessed as previously described (Kanellopoulos et al., 2020 Cell) monitoring the absorbance at 620nm from each single fly abdomen. We could conclude that no preference/avoidance for the drug-containing food was observed between the two genotypes. This dataset has been added in the revised manuscript (new Supplementary Fig. 5c).

Supplementary Fig. 5c, Quantification of food ingestion in control (n = 10) and *Cyfp*^{85.1/+} (n = 10) flies treated with Betulin, DMSO or water. Two-way ANOVA n.s. = not significant. Mean ± S.E.M.

4) Figure 2b (the 24hr profile for NADP⁺/NADPH ratios, in mutants vs control) is key to this study. Unfortunately, the it is also one of the less believable graphs. The difference between the mutant and controls depends on one point, where significance seems marginal. That the 16hr timepoint happens to match the timepoint showing decreased nighttime sleep in the mutant could be viewed as coincidental. I wonder if there are additional approaches that could be taken to validate this result? We are very grateful to the reviewer for suggesting alternative approaches to detect the 24hr profile for NADP⁺/NADPH ratio in control and mutant flies. We performed targeted mass spectrometry analysis (HILIC - MS/MS) to measure the NADP ratio in fly heads every 4 hrs over 24 hrs. The mass spectrometry data is more sensitive and robust than the previously employed ELISA assay (see new Figs 2a and 5a). Importantly, we observe that in control flies the NADP⁺/NADPH follows the behavioural state of the fly, i.e., peaking at high wakefulness periods (ZT0 and ZT12) before sleep periods and descending at ZT16 when sleep need is reduced. In contrast, *Cyfip*^{85.1/+} flies do not show a significant change of the NADP⁺/NADPH ratio (between ZT8 and ZT16) after the peak of wakefulness (ZT12) (new Fig. 2a).

We propose that the NADP ratio oscillations in brain might reflect the sleep/wake behavioural state and that are relevant to induce the night-time sleep onset (model in Fig. 5d), consistently with the effect on sleep due to the increase of NADP⁺ optogenetically induced, observed by the prof. Miesenböck's lab in the dFB circuit. We decided to replace the former Fig. 2a obtained using the kit Colorimetric NADP⁺/NADPH kit from Abcam with the new dataset generated with the HILIC-MS/MS.

Fig. 2a, Upper inset, schematic of experimental design; lower inset, quantification of NADP⁺/NADPH ratio by HILIC-MS/MS in control and *Cyfip*^{85.1/+} fly heads. Control n = 5, *Cyfip*^{85.1/+} n = 5 over 24 hrs. Two-way ANOVA; time point effect $p < 0.0001$, genotype effect $p < 0.0001$, interaction effect $p < 0.0001$, $F_{(6, 56)} = 9.061$, with Tukey's multiple comparison test, *** $p = 0.0009$, control ZT8 vs. ZT12; * $p = 0.0022$, control ZT12 vs. ZT16; $p = 0.99$, *Cyfip*^{85.1/+} ZT8 vs. *Cyfip*^{85.1/+} ZT12; $p = 0.33$, *Cyfip*^{85.1/+} ZT12 vs. *Cyfip*^{85.1/+} ZT16; $p = 0.74$, *Cyfip*^{85.1/+} ZT8 vs. *Cyfip*^{85.1/+} ZT16. n = groups of 50 fly heads, for each time points. Mean \pm S.E.M. (See also Supplementary Data 6).

5) Additionally, regarding the immunohistochemical experiments, were brain dissections done at a specific time of the day? If SREBP levels change throughout the course of 24 hours (peak expression at ZT12-ZT16) all dissections would need to be completed at a specific set time to account for circadian variation. Whether this was accounted for is unclear.

We have now repeated the experiment and better specified in the methods section and legend that the IF in fly brain was performed at ZT12, considering therefore the circadian timing, new Fig. 4f.

6) Starvation leads to sleep loss as well; this is well documented in the field, and is obviously related to metabolism homeostatic processes as well. Have the authors excluded the possibility that they are actually describing one aspect of that well-described phenomenon? In other words, all cells might experience NADP⁺/NADPH ratio changes after extended wakefulness, and these might be reversed upon starvation conditions. Is it possible this has less to do with sleep than more general brain processes linked to hunger, with the mutant animals investigated here being simply more hungry at the start of the night(hence, not sleeping)? It's not clear that this has been ruled out or even considered.

We previously showed that *Cyfp*^{85.1/+} flies do not show differences in food consumption over 24 hours or part of the daytime period (ZT0-ZT6, Kanellopoulos et al., 2020). Following the reviewer suggestion, we investigated the feeding behaviour of *Cyfp* mutant flies assessing food consumption, see above comment (3), during the daytime phase (ZT0-ZT12) and before night-time (ZT10-ZT12). We did not observe differences of feeding behaviour between control and *Cyfp*^{85.1/+} flies. We have now added these datasets in the revised manuscript (new Supplementary Fig. 1h and i).

Supplementary Fig. 1h and i, h-i, Quantification of food consumption during the light-on period (ZT0-ZT12) in *Cyfp*^{85.1/+} (n=108) and control flies (n=108) (h) and just before light off (ZT10-ZT12) in *Cyfp*^{85.1/+} (n=10) and control flies (n=10) (i). Two-tailed unpaired Student's t-test, n.s. = not significant. Mean \pm S.E.M.

Minor concerns and suggestions:

1) Figure 1a - a Schematic would be helpful to the reader to follow experimental setup. Following the reviewer's suggestion, we have added a schematic of the *Drosophila* Activity Monitoring System (Trikinetics) experimental setup. (new Supplementary Fig. 1a).

a

Supplementary Fig. 1a, Schematic of the *Drosophila* Activity Monitoring (DAM) System.

2) Figure 1b – Total max of night-time sleep should be 720 minutes
We thank the reviewer for this suggestion. We have now revised the y-axis scale of the night-time and daytime sleep amount with 720 minutes.

3) Figure 3 and generally seq data – Further explain the rationale behind using a cutoff fold change of +/- 1.3 as the standard method is to use is Log₂FC of 0.58 = LogFold of 1.5. Whilst the authors do state that this was due to a FC of 1.3 of *Cyfp* gene this may need to be detailed more?

Figure 3b could be more informative, e.g. showing some raw seq reads. We are grateful for this suggestion. Initially we arbitrarily used as cut-off the difference in the *Cyfp* mRNA level between WT and *Cyfp*^{85.1/+}. Following the reviewer's suggestion, we reanalysed the RNA seq dataset with a more stringent cut-off of Log₂FC of ± 0.585 (50% fold increase/decrease). Importantly, all conclusions remain valid with the new analysis. We have updated and revised Fig. 3c, d, e and f, Fig. 4a and Suppl. Fig. 4a and relative tables. In addition, we added a volcano plot showing the up- and downregulated genes in *Cyfp*^{85.1/+} vs control flies and a heatmap (new Fig. 3c and d).

Fig. 3, **c**, Volcano plot showing the global transcriptional changes in control vs *Cyfip*^{85.1/+} determined by RNA-seq. Each circle represents one RNA and coloured circles represent DEGs significantly upregulated (FDR ≤ 0.05 and LFC ≥ 0.585 (in orange)) or significantly downregulated (FDR ≤ 0.05 and LFC ≥ -0.585 (in blue)). **d**, Heatmap representing the DEGs significantly dysregulated in the different samples.

4) Figure 4d – The authors have used fluorescent intensity and whilst confocal settings are kept consistent, how have they accounted for background intensity?

We thank the reviewer for this comment. Also following the other reviewers' advice, we improved the signal to noise ratio using an anti-GFP antibody (see methods) (new Fig. 4f). We quantified the same-dimension ROI on 3 brain areas showing a clearer GFP expression (superior lateral protocerebrum; suboesophageal ganglion and the ventrolateral protocerebrum). To account for background intensity, we subtracted an ROI of the same dimension outside the brain tissue, because any other area in the brain seems to express SREBP.

Fig. 4. **e**, Schematic representation of the *Gal4-SREBP::GFP* activation reporter and its activation. The transcription factor domain-encoding sequence was replaced by a *Gal4*-encoding sequence, rendering it specific for the UAS promoter sequence. **f**, Quantification of the

relative GFP labelling intensity at ZT12, in control (n = 12 brains) and *Cyfip*^{85.1/+} (n = 23 brains) flies. **p = 0,0017 by two-tailed unpaired Student's t-test. Mean ± S.E.M. Scale bar = 50 µm. In the representative figure, the max projections of the Z-stack and magnified view of the region marked by white square are presented.

5) Extended figure 2d/e – When comparing the data of 2d and 2e some disparities are seen. *Cyfip* knockdown in *RepoGal4* looks (*) more significant than in *R23E10Gal4*. Some differences in panel d seem as large as in e, yet none are significant in d. This seems suspect. Could this be due to the statistical testing as the former also includes the *Nrv-Gal4* dataset, so simply a problem of multiple comparisons? Secondly The UAS-*Cyfip*-IR controls differ between the datasets.

We thank the reviewer for raising this concern therefore we replicated the experiment one more time confirming the previous results: a reduction in night-time sleep in *R23E10Gal4* flies and no effect on night-time sleep in *RepoGal4* flies. We have revised the figure adding the additional values (revised Suppl. Fig. 3c and Suppl. Fig. 2h). Specifically, both *Cyfip* RNAi lines have a significant reduction in

sleep amount when crossed to the *R23E10Gal4* driver (Suppl. Fig. S3), while when crossed to the *RepoGal4* driver only one significant difference is detected, for RNAi line 1 compared against the no-driver control (there is no significant difference for RNAi line 2, nor for RNAi line 1 against the no-RNAi control). We conclude that knock-down of *Cyfp* in *R23E10*-positive cells reduces sleep. In contrast, *RepoGal4* driven *Cyfp* knock-down does not. Because there are several genotypes in the dataset, we applied Sidak's multiple comparisons test. The reported significance values refer to the corrected values for multiple comparison (Fig. 1, for the reviewer only). To have an independent statistical analysis we performed One Way ANOVA excluding the *NrvGal4* dataset and obtained the same results (Fig. 2, for the reviewer only).

Ordinary one-way ANOVA Multiple comparisons									
1	Number of families	1							
2	Number of comparisons per family	8							
3	Alpha	0.05							
4									
Sidak's multiple comparisons test									
	Mean Diff.	95.00% CI of diff.	Below threshold?	Summary	Adjusted P Value				
6	RepoGal4 > UAS-CYFIP RNAi 1 vs. RepoGal4/+	-31.82	-73.21 to 9.577	No	ns	0.2513	D-E		
7	RepoGal4 > UAS-CYFIP RNAi 2 vs. RepoGal4/+	13.80	-35.40 to 63.00	No	ns	0.9905	A-E		
8	UAS-CYFIP RNAi 1 vs. RepoGal4 > UAS-CYFIP RNAi 1	53.63	15.73 to 91.53	Yes	**	0.0010	C-D		
9	RepoGal4 > UAS-CYFIP RNAi 2 vs. UAS-CYFIP RNAi 2	-21.79	-67.93 to 24.34	No	ns	0.8239	A-B		
10	Nrv-Gal4/+ vs. Nrv-Gal4>CYFIP RNAi 1	37.44	-5.875 to 80.76	No	ns	0.1367	F-G		
11	Nrv-Gal4/+ vs. Nrv-Gal4>CYFIP RNAi 2	5.672	-37.65 to 48.99	No	ns	>0.9999	F-H		
12	UAS-CYFIP RNAi 1 vs. Nrv-Gal4>CYFIP RNAi 1	-34.84	-74.83 to 5.158	No	ns	0.1306	C-G		
13	UAS-CYFIP RNAi 2 vs. Nrv-Gal4>CYFIP RNAi 2	-52.83	-92.62 to -13.03	Yes	**	0.0025	B-H		
14									
Test details									
	Mean 1	Mean 2	Mean Diff.	SE of diff.	n1	n2	t	DF	
16	RepoGal4 > UAS-CYFIP RNAi 1 vs. RepoGal4/+	485.8	517.7	-31.82	15.06	39	30	2.113	260
17	RepoGal4 > UAS-CYFIP RNAi 2 vs. RepoGal4/+	531.5	517.7	13.80	17.90	20	30	0.7711	260
18	UAS-CYFIP RNAi 1 vs. RepoGal4 > UAS-CYFIP RNAi 1	539.5	485.8	53.63	13.79	42	39	3.890	260
19	RepoGal4 > UAS-CYFIP RNAi 2 vs. UAS-CYFIP RNAi 2	531.5	553.2	-21.79	16.78	20	43	1.299	260
20	Nrv-Gal4/+ vs. Nrv-Gal4>CYFIP RNAi 1	611.7	574.3	37.44	15.76	30	32	2.376	260
21	Nrv-Gal4/+ vs. Nrv-Gal4>CYFIP RNAi 2	611.7	606.1	5.672	15.76	30	32	0.3600	260
22	UAS-CYFIP RNAi 1 vs. Nrv-Gal4>CYFIP RNAi 1	539.5	574.3	-34.84	14.55	42	32	2.395	260
23	UAS-CYFIP RNAi 2 vs. Nrv-Gal4>CYFIP RNAi 2	553.2	606.1	-52.83	14.47	43	32	3.650	260

Fig 1 for the reviewer, Prism analysis One Way ANOVA followed by Sidak's multiple comparisons, **p < 0.01.

Ordinary one-way ANOVA Multiple comparisons									
1	Number of families	1							
2	Number of comparisons per family	4							
3	Alpha	0.05							
4									
Sidak's multiple comparisons test									
	Mean Diff.	95.00% CI of diff.	Below threshold?	Summary	Adjusted P Value				
6	RepoGal4 > UAS-CYFIP RNAi 2 vs. RepoGal4/+	13.80	-30.73 to 58.33	No	ns	0.8991	A-E		
7	RepoGal4 > UAS-CYFIP RNAi 1 vs. RepoGal4/+	-31.82	-69.28 to 5.644	No	ns	0.1289	D-E		
8	UAS-CYFIP RNAi 1 vs. RepoGal4 > UAS-CYFIP RNAi 1	53.63	19.33 to 87.93	Yes	***	0.0005	C-D		
9	RepoGal4 > UAS-CYFIP RNAi 2 vs. UAS-CYFIP RNAi 2	-21.79	-63.54 to 19.96	No	ns	0.5707	A-B		
10									
Test details									
	Mean 1	Mean 2	Mean Diff.	SE of diff.	n1	n2	t	DF	
12	RepoGal4 > UAS-CYFIP RNAi 2 vs. RepoGal4/+	531.5	517.7	13.80	17.69	20	30	0.7803	169
13	RepoGal4 > UAS-CYFIP RNAi 1 vs. RepoGal4/+	485.8	517.7	-31.82	14.88	39	30	2.138	169
14	UAS-CYFIP RNAi 1 vs. RepoGal4 > UAS-CYFIP RNAi 1	539.5	485.8	53.63	13.62	42	39	3.936	169
15	RepoGal4 > UAS-CYFIP RNAi 2 vs. UAS-CYFIP RNAi 2	531.5	553.2	-21.79	16.58	20	43	1.314	169

Fig. 2 for the reviewer, Prism analysis excluding the NrvGal4 genotypes. One Way ANOVA followed by Sidak's multiple comparisons.

5) Finally, there is no quantification of these knockdowns.

We thank the reviewer for his/her comment. We compared *Cyfp* mRNA levels by RT-qPCR in flies with pan-neuronal reduction of *Cyfp* expression (*Ras2Gal4;Gal80^{ts} > UAS-Cyfp-IR¹ or²*) to controls. We observed a reduction by at least 30% of *Cyfp* mRNA expression levels upon the two RNAi expression. We have now added this dataset in the revised version of the manuscript (new Suppl. Fig. 2a).

Supplementary Fig. 2a, **a**, Quantitative RT-PCR to assess the levels of *Cyfp* mRNA normalized to *rpl32*. *Ras2Gal4;TubGal80^{ts/+}* (n = 12), *UAS-Cyfp-IR^{1/+}* (n = 7), *Ras2Gal4;TubGal80^{ts} > UAS-Cyfp-IR¹* (n = 7), *UAS-Cyfp-IR^{2/+}* (n = 7) and *Ras2Gal4;TubGal80^{ts} > UAS-Cyfp-IR²* (n = 7). Ordinary one-way ANOVA followed by Sidak's multiple comparison test, **** $p < 0.0001$, ** $p = 0.0018$, *** $p = 0.001$, **** $p < 0.0001$. n = pool of 15 fly heads. Mean \pm S.E.M. (see also Supplementary Data 6).

6) Extended Figure 2F – Which UAS-Cyfp-IR line was used?

We have included the *UAS-Cyfp IR* line used, namely the line 1 generated in prof. Giangrande's laboratory (revised Suppl. Fig.3g).

Supplementary Fig. 3g, Night sleep amount upon *Cyfp* reduction in different clusters of clock neurons. Kruskal-Wallis test followed by Dunn's multiple comparisons test, * $p = 0.047$, n.s. = not significant. n = 18 – 20 (see also Supplementary Data 6). Mean \pm S.E.M.

7) Methods: Sleep-Wake activity assays – light as an arousal stimulus is unusual, as it interferes with the circadian clock. This might need some discussion, to address concerns that circadian factors aren't at play here.

We thank the reviewer for this comment. We believe there is no interference on the circadian rhythm by the light stimulus because for 3 days the light cycles were regular and only the last day 3 flashes of light were given, and the experiment ended (see methods). The light stimulus in *Drosophila* was previously used as arousal stimulus (for example Seugnet et al., 2009, PMID: 19494137; Ni et al., 2019, PMID: 30719975) and we were inspired by these publications showing that specific circuitries regulating arousal activated by light stimuli are independent from circadian pacemaker circuitries (Sheeba et al., 2008, PMID: 18771923). We have now mentioned these papers in the result section.

It would also be interesting to know how many flies were not responding to this stimulus.

To address this point, we have added a new panel that shows for each time point of the unexpected stimulus the number of flies responding to the stimulus (new Suppl. Fig. 1g).

g

	ZT16	ZT18	ZT20
	Responders/ total	Responders/ total	Responders/ total
Control	22/ 24 (91.7%)	19/ 24 (79.2%)	19/ 24 (79.2%)
Cyfp^{85.1/+}	18/ 30 (60.0%)	22/ 30 (83.3%)	22/ 30 (80.0%)

Supplementary Fig. 1g, Percentage of flies showing a reduction of sleep >2% compared to baseline sleep. Shown are the data for control and *Cyfp^{85.1/+}* flies, as well as three time points when the arousal stimulus was provided (ZT16, ZT18 and ZT20). Fisher's exact test, $p = 0.0118$, control vs *Cyfp^{85.1/+}* at ZT16, n.s. at ZT18 and ZT20.

8) Fig. 5: in the concluding model (panel d) the authors seem to imply a daytime sleep effect, as well as a night time effect (increased NADP+/NADPH ratio leads to low sleep drive during both day and night). Yet, this is not what is proposed in the title and abstract and discussion. The authors might need to rethink their effects on daytime sleep, and this included showing all the daytime sleep data so to that readers can see the extent to which day and night sleep correlate in these manipulations. This is critical, because without the word 'night' in the title, the story packs somewhat less of a punch. Following the reviewer's suggestion, we have now included the sleep data during the daytime for all the experiments performed in which we observed a night-time sleep phenotype further strengthening the importance of the metabolic pathway in the regulation of night-time sleep onset and sleep need.

In the revised version of the manuscript, we have modified the model highlighting the specificity of this mechanism at the night-time sleep onset.

New Fig. 5d, Model.

Reviewer #2 (Remarks to the Author):

Mariano et al. use a previously established *Drosophila* model for neurodevelopmental disease to investigate the molecular mechanisms underlying sleep dysregulation in these disorders. They study flies that have a 50% reduction in *Cyfp*, a protein implicated in various neuropsychiatric disorders. Their data suggest that *Cyfp* inhibits SREBP which in turn upregulates malic enzyme. Thus, a reduction in *Cyfp* is proposed to enhance SREBP and malic enzyme levels, which then increase the NADPH/NADP ratio, resulting in a putative decrease in the metabolic drive for sleep. In general, the work is solid, especially the links between *Cyfp*, SREBP, and malic enzyme, and the functional connection between lipid metabolism and sleep regulation is interesting and potentially important. However, there are questions about whether the observed behavioral phenotypes are specifically related to sleep homeostasis, which could limit the novelty of the findings.

We appreciate the reviewer's positive comments on our work and performed the suggested experiments that helped to improve our manuscript. We hope that he/she is pleased with the new datasets included in the revised version of the manuscript.

Major points:

1) The sleep phenotypes observed may not reflect defects in sleep homeostasis. For example, in Figures 1D-1E, the data show that *Cyfp* hets are hyperaroused, which could account for the reduced baseline sleep and decreased recovery sleep after deprivation.

We thank the reviewer for the comments. We do not think that the *Cyfp*^{85.1/+} are largely hyperaroused because after providing the light stimuli at night-time we did not observe more *Cyfp*^{85.1/+} responders compared to control flies, that could account for the increase in arousal and reduction of arousal threshold (new Suppl. Fig. 1g). Nevertheless, *Cyfp*^{85.1/+} flies have more troubles to fall asleep and to reach a baseline sleep level after light pulses compared to control flies (Fig. 1d).

g

	ZT16	ZT18	ZT20
	Responders/ total	Responders/ total	Responders/ total
Control	22/ 24 (91.7%)	19/ 24 (79.2%)	19/ 24 (79.2%)
Cyfp ^{85.1/+}	18/ 30 (60.0%)	22/ 30 (83.3%)	22/ 30 (80.0%)

Supplementary Fig. 1g, Percentage of flies showing a reduction of sleep >2% compared to baseline sleep. Shown are the data for control and *Cyfp*^{85.1/+} flies, as well as three time points when the arousal stimulus was provided (ZT16, ZT18 and ZT20). Fisher's exact test, $p = 0.0118$, control vs *Cyfp*^{85.1/+} at ZT16, n.s. at ZT18 and ZT20.

To further test if *Cyfp*^{85.1/+} flies had deficits in sleep homeostasis, we monitored the NADP⁺/NADPH in control and *Cyfp*^{85.1/+} flies during undisturbed sleep (noSD) or sleep-deprived (SD) as in Cirelli et al., 2005, PMID: 16001966. We observe that the NADP⁺/NADPH ratio is increased in control flies after sleep deprivation (new Fig. 2b), testifying the increase in sleep need reflected by the accumulation of NADP⁺ over NADPH while no change in the NADP⁺/NADPH ratio was detected in *Cyfp*^{85.1/+} flies.

b

Figure 2b, upper inset, schematic of the experimental design. noSD: undisturbed sleep, SD: sleep deprived. Lower inset, NADP⁺/NADPH ratio in flies in control and *Cyfp*^{85.1/+} fly heads in noSD (control, n = 6, and *Cyfp*^{85.1/+}, n = 6) SD conditions (control, n = 6, and *Cyfp*^{85.1/+}, n = 6). Two-way ANOVA, sleep condition effect $p = 0.0217$, genotype effect $p = 0.0243$, interaction effect $p = 0.022$, $F_{(1, 20)} = 6.165$, with Sidak's multiple comparison test $**p = 0.0043$, control noSD vs. SD, n.s. = not significant, *Cyfp*^{85.1/+} noSD vs. SD. n = pool of 23 fly heads. Mean \pm S.E.M.

Do levels or activity of *Cyfp*, malic enzyme, or SREBP vary according to sleep need?

Following the reviewers' comment we performed RT-qPCR to detect *Men* mRNA in wild-type during undisturbed sleep (noSD) or sleep-deprived (SD) as in Cirelli et al., 2005 PMID: 16001966. Our findings are also consistent with published literature showing that *Men* expression is associated with wakefulness and upregulated upon sleep deprivation in wild-type flies (Cirelli et al., 2005, PMID: 16001966) (new Fig. Suppl. 5d) and fly models of insomnia (Seugnet et al., 2009, PMID: 19494137).

Cyfp and *Srebp* mRNA levels did not change between noSD and SD conditions in wild-type flies (Fig. 3 for the reviewer only).

Fig. 3 for the reviewer, RT-qPCR to assess *Men* (A), *Cyfp* (B) and *Srebp* (C) gene expression level in control flies during undisturbed sleep (noSD, n = 4) or sleep-deprived (SD, n = 4). n = pool of 15 heads. Two-tailed Student's t-test, *p<0.05. Mean ± S.E.M.

2) It is not clear that *Cyfp* hets lack an increase of NADP/NADPH around dusk. The data appear to show an increase, but just earlier, from ZT8 to ZT12. Also, if the NADP/NADPH ratio reflects sleep need, then one would expect the peak to be ZT12, not ZT16, since animals would be sleeping from ZT12-16, which should lower sleep need.

We are very grateful to the reviewer for this comment. We have detected the NADP⁺/NADPH ratio in control and mutant flies using a more sensitive and robust method than the previously employed ELISA assay. Specifically, we performed targeted mass spectrometry analysis (HILIC - MS/MS) to measure the NADP ratio in fly heads every 4 hrs over 24 hrs (see new Figs 2a and 5a). Importantly, we observe that in control flies the NADP⁺/NADPH follows the behavioral state of the fly, i.e., peaking at high wakefulness periods (ZT0 and ZT12) before sleep periods and descending at ZT16 when sleep need is reduced. In contrast, *Cyfp*^{85.1/+} flies do not show a significant change of the NADP⁺/NADPH ratio (between ZT8 and ZT16) after the peak of wakefulness (ZT12) (new Fig. 2a). Our data are indeed consistent with the hypothesis of the reviewer. We decided to replace the former figure 2a obtained using the kit Colorimetric NADP⁺/NADPH kit from Abcam with the new dataset generated with the HILIC-MS/MS.

Fig. 2a, Upper inset, schematic of experimental design; lower inset, quantification of NADP⁺/NADPH ratio by HILIC-MS/MS in control and *Cyfp*^{85.1/+} fly heads. Control n = 5, *Cyfp*^{85.1/+} n = 5 over 24 hrs. Two-way ANOVA; time point effect $p < 0.0001$, genotype effect $p < 0.0001$, interaction effect $p < 0.0001$, $F_{(6, 56)} = 9.061$, with Tukey's multiple comparison test, *** $p = 0.0009$, control ZT8 vs. ZT12; * $p = 0.0022$, control ZT12 vs. ZT16; $p = 0.99$, *Cyfp*^{85.1/+} ZT8 vs. *Cyfp*^{85.1/+} ZT12; $p = 0.33$, *Cyfp*^{85.1/+} ZT12 vs. *Cyfp*^{85.1/+} ZT16; $p = 0.74$, *Cyfp*^{85.1/+} ZT8 vs. *Cyfp*^{85.1/+} ZT16. n = groups of 50 fly heads, for each time points. Mean ± S.E.M. (see also Supplementary Data 6).

3) In *Cyfp* hets and in a number of other manipulations, there is a suggestion of a rightward shift of both daytime and nighttime sleep. It would be helpful if the authors performed locomotor analyses in constant darkness to assess whether there are underlying circadian defects.

We agree with the reviewer's comment and add in the revised manuscript the analysis of the circadian rhythm of control and *Cyfp*^{85.1/+} flies. The actograms of the free-running locomotion in complete darkness and the periodicity (Tau value) of control and *Cyfp*^{85.1/+} flies did not reveal any circadian rhythm deficits in complete darkness and a typical periodicity (new Suppl. Fig. 3e and f). In addition, no sleep defects were observed upon *Cyfp* RNAi expression in clock neurons (new Suppl. Fig. 3g), further demonstrating that the circadian rhythm is not affected in the *Cyfp*^{85.1/+} flies.

Supplementary Fig. 3e-f: **e**, Double plotted actogram of control (n = 20) and *Cyfp*^{85.1/+} (n = 18) during a period of light-dark 12:12 followed by a free running rhythm in continuous darkness (DD). The start of the DD period is indicated by dashed red line. **f**, Rhythmicity (R),

expressed as percentage of rhythmic flies in the experiment and Period (Tau), in hrs of control (n = 20) and *Cyfp*^{85.1/+} flies (n = 18). Two-tailed unpaired Student's t-test, $p = 0.272$.

4) Fig S2-- To support the notion that reduction of *Cyfp* levels affect sleep, the authors use 2 different RNAi lines crossed to *ras2-Gal4*. It is unclear why the authors chose here and in Figure 4 to use *ras2-Gal4*, as published literature suggests that it labels both non-neuronal and neuronal cells. These experiments should be repeated with *nsyb-Gal4* or *elav-Gal4*.

Following the reviewer's comment we performed the same experiment using the *ElavGal4;TubGal80^{ts}* driver. We observed that flies expressing the driver *ElavGal4;TubGal80^{ts}* alone at 30°C show deficits in night-time sleep behaviour (Fig. 2A and B, and Fig. 4A-B for the reviewer only) consistent with previously published work (Tomita et al., 2011. PMID: 21917797). This line is therefore not suitable for our experiment.

Fig. 4 for the reviewer, Sleep profiles of *ElavGal4;TubGal80^{ts}* upon *UAS-Cyfp-IR¹* (A) and *UAS-Cyfp-IR²* (B) at 30°C. *ElavGal;TubGal80ts/+* control shows a marked nighttime sleep reduction. n = 20-32 flies.

When the *nSybGal4* driver is combined with the *UAS-Cyfp-IR^{1/2}* we did not see deficit in night-time sleep (Fig. 5, for the reviewer only). Of note, the GFP staining in *nSybGal4* and *Ras2Gal4;TubGal80^{ts}* does not show a complete overlapping neuronal staining (data not shown). We therefore believe that this difference in behaviour could be due to the different expression pattern between the *nSybGal4* driver and the *Ras2Gal4;TubGal80^{ts}* driver. As pointed out by the referee, while *nSybGal4* has a pattern exclusively neuronal the *Ras2Gal4;TubGal80^{ts}* was described to target also some non-neuronal cells in the developing larvae (salivary glands and gut. Walker et al., 2006, PMID: 17114577). However, in the adult fly it marks the overall neurons of the CNS and it is particularly enriched in neuronal circuitries associated to learning and memory, namely the MBs (Gouzi et al., 2011, PMID: 21949657) so we are confident that we are silencing *Cyfp* in neuronal cells in the adult.

Fig. 5 for the reviewer, Sleep profiles of the *nSybGal4* flies upon *UAS-Cyfp-IR¹* (A) and *UAS-Cyfp-IR²* (B) at 25°C. n = 20-32 flies.

Also, the authors should show the sleep profile traces for the experiments in Figure S2—does knockdown of *Cyfp* show a similar decrease in early night sleep seen in *Cyfp/+* flies?

We thank the reviewer for this comment, we added the graphs of the sleep profiles of *RasGal4;Gal80^{ts} > UAS-Cyfp-IR^{1 and 2}* and respective controls confirming that the abrogation of *Cyfp* in *Ras2+* neurons phenocopied the sleep deficits of *Cyfp^{85.1/+}* flies (new Suppl. Fig. 2c).

Supplementary Fig. 2c, Representative sleep profiles of *Ras2Gal4;TubGal80^{ts/+}* (n = 48), *UAS-Cyfp-IR^{1/+}* (n = 15), *Ras2Gal4;TubGal80^{ts} > UAS-Cyfp-IR¹* (n = 19) (upper panel) and *RasGal4;TubGal80^{ts/+}* (n = 48), *UAS-Cyfp-IR^{2/+}* (n = 21) and *Ras2Gal4;TubGal80^{ts} > UAS-Cyfp-IR²* (n = 31) (lower panel).

5) Fig S2—they suggest that the effects of *Cyfp* knockdown on sleep can be localized to the ExF12 neurons. To strengthen this potential mechanism, they should show knockdown of SREBP and malic enzyme in the ExF12 neurons as well.

Following the reviewer's comment, that we believe meant to overexpress SREBP (same condition we have in *Cyfp* mutants), we overexpressed the constitutively active form of *Srebp* (*Srebp^{c.del}*) using the *R23E10Gal4* driver. In this case, while we did not observe a significant reduction of sleep a

tendency towards impaired sleep. These preliminary data suggest that SREBP alone is not enough to affect sleep-regulation in ExFI2 neurons (data not shown). Because it was shown that there is another circuitry – upstream the ExFI2 neurons - is involved in sleep homeostasis, we hypothesise that multiple neurons might be involved (Liu et al., 2016, PMID: 27212237).

Men overexpression in this circuit is also a great idea, we have looked for this line extensively but unfortunately there are no described fly overexpressing *Men*. We thank the reviewer again for this suggestion and we hope to generate additional specific fly lines in the future.

Minor comments:

1) Page 3, line 2, provide citation for sleep being conserved from worms to humans.

We added a citation of a recent review discussing the universality of sleep as behavioural state and the recent developments supporting the conservation of neural dynamics of sleep states across phylogeny (Jaggard et al., 2021 PMID: 34583217).

2) Figure 1E, it is unusual to examine recovery sleep after sleep deprivation during the night, when sleep is already high. The authors do examine recovery sleep during the daytime after 12 hr of sleep deprivation in Figure S2. Recommend moving the panel from Figure S2 to the main figures, and Figure 1E to the supplemental figures.

Following this suggestion, we moved this experiment to the main figure (new Fig. 1e).

3) The authors should specify how many times the transgenic lines were backcrossed into their control background strain.

Transgenic fly lines were backcrossed for 6 generations in the WT background strain. We have added this information in the methods section.

4) Can the authors comment on the marked expression of the SREBP reporter in ellipsoid body ring?

We thank the reviewer for raising this concern that was also raised by the other reviewer. We indeed discovered that the stock we were using had a P{ActGFP}JMR1 transgene in both wild-type and *Cyfp*^{85.1/+}. generating GFP expression in ellipsoid bodies (Fig.6A, for the reviewer only). Therefore, we removed the P{ActGFP}JMR1 transgene present on the balancer (Fig. 6B, for the reviewer only). The new experiment confirmed the increase in GFP expression from the SREBP-responsive reporter in the *Cyfp*^{85.1/+} confirming the *Cyfp*-*Srebp* axis. We have replaced the figure (new Fig. 4d).

Fig. 6 for the reviewer, A-B, Confocal fly brain sections encompassing the ellipsoid body from *CyO*, P{ActGFP}JMR1, SREBP reporter flies (A), and SREBP reporter/+ (B). The P{ActGFP}JMR1 transgene has a marked expression of GFP in the ellipsoid body.

Fig. 4. **e**, Schematic representation of the *Gal4-SREBP::GFP* activation reporter and its activation. The transcription factor domain-encoding sequence was replaced by a *Gal4*-encoding sequence, rendering it specific for the UAS promoter sequence. **f**, Quantification of the relative GFP labelling intensity at ZT12, in control (n = 12 brains) and *Cyfip*^{85.1/+} (n = 23 brains) flies. **p

= 0,0017 by two-tailed unpaired Student's t-test. Mean ± S.E.M. Scale bar = 50 μm. In the representative figure, the max projections of the Z-stack and magnified view of the region marked by white square are presented.

5) Genotypes are sometimes ambiguous. For example in Figure 5, *Cyfip/SREBP* animals—are these homozygous or heterozygous for the *SREBP* allele?

We thank the reviewer for this comment, in the revised version of the manuscript we include a Supplementary table (new Suppl. Data 6) listing all the fly genotypes used in this work.

6) For *tubGal80ts* experiments, the authors state in the methods that they subject the flies to 29 deg for 3 days. Which days did they use for their baseline sleep recording?

We are sorry that this part of the text was not clear. We have better explained how the experiments with the temperature inducible *Gal80ts* lines were performed in the methods section as follows:

“Temperature-induced experiments were performed as follows: UAS-Cyfip-IR flies expressing Ras2Gal4;TubGal80^{ts} were raised at 18°C throughout their development to avoid the Cyfip developmentally lethal effect. The TubGal80^{ts} was induced at 29°C, 3-5 days after eclosion, for 3 days allowing the acute and strong induction of the Gal4. From day 4 onward flies were assessed for sleep behavior at 29°C or RNA extraction. To demonstrate that the UAS-Cyfip-IR is not expressed upon TubGal80^{ts} at 18°C, flies were raised, kept, and tested at 18°C. For the behavioural experiments involving UAS-Srebp^{wt} and ^{c.del}, Ras2Gal4;TubGal80^{ts} > UAS-Srebp^{wt} or ^{c.del} and respective control flies were raised at 25°C throughout development, kept and tested 5 to 7 days after eclosion at 25°C.”

7) Figure S2E—“R23E18” should be “R23E10”

Many thanks, we have corrected the typo.

8) The discussion is somewhat meandering. Recommend tightening this section.

The discussion has been revised, it is shorter and more focused.

Reviewer #3 (Remarks to the Author):

By characterising the sleep profile of *Cyfip* haploinsufficiency mutants, Mariano and coauthors identified a novel and original mechanism underlying the regulation of sleep homeostasis in flies. Using a series of complementary and elegant experiments, the authors convincingly find that the

activity of the master regulator of lipogenesis SREBP is upregulated in *Cyfp* mutants, causing the upregulation of lipogenic genes, and, in turn, nighttime sleep defects via an imbalance of the $NADP^+/NADPH$ ratio. These findings are highly relevant to the field. I recommend publication in *Nature Communications* provided that the authors address the major and minor points raised below.

We greatly appreciate the reviewer's positive comments on our work. In the revised version of the manuscript, we have followed his/her suggestions and performed the requested experiments accordingly. We believe that the new data have strengthened the impact of our findings.

Major points:

1) Figure 2b. It looks like the $NADP^+/NADPH$ ratio oscillates in a circadian-dependent manner following the sleep pattern of the flies and not in a sleep need-dependent manner (as correctly stated in the abstract but not in the main text). Otherwise, why would the $NADP^+/NADPH$ start increasing when the flies are asleep, i.e. when sleep need is being dissipated, and not when sleep need is highest, i.e. at the end the 12-hour day period (ZT8-12)? Even though it is an interesting hypothesis. We are very grateful to the reviewer for this comment. We have detected the $NADP^+/NADPH$ ratio in control and mutant flies using a more sensitive and robust than the previously employed ELISA assay. Specifically, we performed targeted mass spectrometry analysis (HILIC - MS/MS) to measure the NADP ratio in fly heads every 4 hrs over 24 hrs (see new Figs 2a and 5a). Importantly, we observe that in control flies the $NADP^+/NADPH$ follows the behavioural state of the fly, i.e., peaking at high wakefulness periods (ZT0 and ZT12) before sleep periods and descending at ZT16 when sleep need is reduced. In contrast, *Cyfp*^{85.1/+} flies do not show a significant change of the $NADP^+/NADPH$ ratio (between ZT8 and ZT16) after the peak of wakefulness (ZT12) (new Fig. 2a). Our findings are consistent with the hypothesis of the reviewer. We decided to replace the former figure 2a obtained using the kit Colorimetric $NADP^+/NADPH$ kit from Abcam with the new dataset generated with the HILIC-MS/MS.

Fig. 2a, Upper inset, schematic of experimental design; lower inset, quantification of $NADP^+/NADPH$ ratio by HILIC-MS/MS in control and *Cyfp*^{85.1/+} fly heads. Control $n = 5$, *Cyfp*^{85.1/+} $n = 5$ over 24 hrs. Two-way ANOVA; time point effect $p < 0.0001$, genotype effect $p < 0.0001$, interaction effect $p < 0.0001$, $F_{(6, 56)} = 9.061$, with Tukey's multiple comparison test, *** $p = 0.0009$, control ZT8 vs. ZT12; * $p = 0.0022$, control ZT12 vs. ZT16; $p = 0.99$, *Cyfp*^{85.1/+} ZT8 vs. *Cyfp*^{85.1/+} ZT12; $p = 0.33$, *Cyfp*^{85.1/+} ZT12 vs. *Cyfp*^{85.1/+} ZT16; $p = 0.74$, *Cyfp*^{85.1/+} ZT8 vs. *Cyfp*^{85.1/+} ZT16. $n =$ groups of 50 fly heads, for each time points. Mean \pm S.E.M. (see also Supplementary Data 6).

I do not think that the authors can causally link the changes in $NADP^+/NADPH$ ratio to the regulation of night-time sleep and in particular sleep homeostasis. To address the latter, they would need to show that the $NADP^+/NADPH$ ratio is differentially modulated after sleep deprivation (by compared sleep-deprived versus rested flies) in WT versus mutant flies. In addition, the authors should carefully rephrase the interpretation of the $NADP^+/NADPH$ ratio results.

To further test if *Cyfp*^{85.1/+} flies had deficits in sleep homeostasis, we monitored the $NADP^+/NADPH$ in control and *Cyfp*^{85.1/+} flies during undisturbed sleep (noSD) or sleep-deprived (SD) as in Cirelli et

al., 2005; PMID: 16001966. We observe that the NADP⁺/NADPH ratio is increased in control flies after sleep deprivation (new Fig. 2b), testifying the increase in sleep need reflected by the accumulation of NADP⁺ over NADPH. No changes in the NADP⁺/NADPH ratio is observed in the *Cyfp^{85.1/+}* flies.

Fig. 2b, upper inset, schematic of the experimental design. noSD: undisturbed sleep, SD: sleep deprived. Lower inset, NADP⁺/NADPH ratio in flies in control and *Cyfp^{85.1/+}* fly heads in noSD (control, n = 6, and *Cyfp^{85.1/+}*, n = 6) SD conditions (control, n = 6, and *Cyfp^{85.1/+}*, n = 6). Two-way ANOVA, sleep condition effect $p = 0.0217$, genotype effect $p = 0.0243$, interaction effect $p = 0.022$, $F_{(1, 20)} = 6.165$, with Sidak's multiple comparison test $**p = 0.0043$, Control noSD vs. SD, n.s. = not significant, *Cyfp^{85.1/+}* noSD vs. SD. n = pool of 23 fly heads. Mean \pm S.E.M.

2) Using a series of complementary and elegant experiments, the authors convincingly show that the master regulator of lipogenesis SREBP is unregulated in *Cyfp* mutants. What about the defects in sleep homeostasis shown in Extended Data Figure 2g and Figure 1e? Can they be rescued upon decreasing the expression of SREBP?

We apologize if the suggested experiments were not well-described in our initial submission. In the former Extended Data Fig. 2g (now new Fig. 4g) we sleep-deprived flies for 6 hrs and measured the sleep rebound (i.e., sleep after deprivation – baseline sleep) in control, *Cyfp^{85.1/+}* and *Cyfp^{85.1/Srebp¹⁸⁹}* flies (rescue line). We observed rebound deficits in *Cyfp^{85.1/+}* flies rescued in *Cyfp^{85.1/Srebp¹⁸⁹}* flies.

Moreover, does neuronal SREBP expression block synaptic homeostasis? This should be addressed in the paper.

We thank the reviewer for this question. We assessed sleep behaviour using a constitutive active form of *Srebp*, that enters the nucleus activating transcription, using one of the sleep deprivation paradigms tested in our paper (6 hrs). The *Ras2Gal4;Gal80^{ts} > UAS-Srebp^{c.del}* flies show indeed a deficit on night sleep homeostasis as shown by the last peak of sleep rebound (orange color) that is lower than the respective controls (Fig. 7 for the reviewer only). We therefore can conclude that SREBP levels have an affects the sleep homeostasis.

Fig. 7 for the reviewer, Sleep profiles of the *Ras2Gal4;Gal80^{ts} > UAS-Srebp^{c.del}* flies and controls. After baseline sleep, flies were sleep deprived for 6 hrs. n = 20-32 flies.

The images shown in Figure 4d are very difficult to quantify and to interpret due to the low signal-to-noise ratio. Looking at the pictures, it seems that the fluorescence signal is higher in WT than mutant 'ring neurons' for example (it is visible from the pictures that SREBP is active in a subpopulation of ring neurons). Yet, the 'background' (or neuropil or SREBP?) signal is much higher in mutant than WT brains and quantifying this signal without correcting for background or without attributing it to neuronal expression might lead to wrong conclusions. One way to circumvent this problem would simply be to stain for GFP (instead of doing live imaging) and additionally to counterstain the neuropil with anti-Bruchpilot for example. This would not only amplify the signal and improve the signal-to-noise ratio but it would also allow to do ratiometric imaging and to correct, e.g., for background noise or antibody penetration. Without these changes, the results shown in Figure 4d are not conclusive. We thank the reviewer for raising this concern that was also raised by the other reviewers. We indeed discovered that the stock we were using had a P{ActGFP}JMR1 transgene in both wild-type and *Cyfp* mutants, generating GFP expression in ellipsoid bodies (Fig.6A, for the reviewer). Therefore, we removed the P{ActGFP}JMR1 transgene that was present on the balancer (Fig. 6B, for the reviewer). Importantly, the new experiment confirmed the increase in GFP expression from the SREBP-responsive reporter in the context of *Cyfp*^{85.1/+} confirming the *Cyfp*-Sreb axis. We have replaced the figure (new Fig. 4d).

We understand the concerns about the quantification, and we have therefore performed additional stainings improving the signal-to-noise ratio detecting GFP with an antibody (compared to the previous images that were acquired at the confocal detecting the endogenous GFP with no signal amplification). The quantification is based on a ROI (same size) on 3 brain areas with GFP expression (superior lateral protocerebrum; suboesophageal ganglion and the ventrolateral protocerebrum). To account for background intensity, we subtracted an ROI of the same dimension outside the brain because *Sreb* seems to be expressed in all brain areas.

We did not use *Brp* as normalizer because in the past we had generated data suggesting that *Brp* expression is different between control and *Cyfp*^{85.1/+} fly brain (Mariano and Bagni, unpublished). In addition, *Brp* expression is modulated by sleep pressure (Gilestro et al., 2005, PMID: 19342593).

Fig. 6 for the reviewer only, A-B, Confocal fly brain sections encompassing the ellipsoid body from *CyO*, P{ActGFP}JMR1, SREBP reporter flies (A), and SREBP reporter/+ (B). The P{ActGFP}JMR1 transgene has a marked expression of GFP in the ellipsoid body.

Fig. 4. e, Schematic representation of the *Gal4-SREBP::GFP* activation reporter and its activation. The transcription factor domain-encoding sequence was replaced by a *Gal4*-encoding sequence, rendering it specific for the UAS promoter sequence. f, Quantification of the relative GFP labelling intensity at ZT12, in control (n = 12 brains) and *Cyfp*^{85.1/+} (n = 23 brains) flies. **p =

0,0017 by two-tailed unpaired Student's t-test. Mean \pm S.E.M. Scale bar = 50 μ m. In the representative figure, the max projections of the Z-stack and magnified view of the region marked by white square are presented.

4) In Kanellopoulos et al, the same lab found a higher mitochondrial activity in the *Cyfp* mutant brain. This led to a redistribution of the GABA pool from synaptic vesicles into the mitochondria, thereby dampening GABAergic synaptic transmission, which, in turn, had severe effects on the social behaviour and locomotor activity of mutant flies. In the same paper, the authors also showed that the social deficits but not the locomotor activity could be rescued by decreasing mitochondrial activity or increasing GABA levels. I am surprised that the authors did not follow up on this finding in the context of sleep. It would be interesting to know whether similar manipulations as used in Kanellopoulos et al can also rescue the effects on nighttime sleep and sleep homeostasis observed in *Cyfp* mutants. If not, it might be very intriguing that the same protein might affect different types of behaviour (social behaviour and sleep) via entirely different metabolic mechanisms and possibly different neuronal substrates.

We thank the reviewer for this suggestion and indeed our curiosity at the time of Kanellopoulos et al lead us to investigate if sleep as well is under the control of mitochondria metabolism. We tested sleep/wake behaviour in *Aralar*^{MI07552/+} and *Cyfp*^{85.1}/*Aralar*^{MI07552} flies and did not observe an amelioration of night-time sleep in *Cyfp*^{85.1} flies, suggesting that *Aralar* activity has a specific effect on social behaviour while sleep homeostasis is under the control of other factors such *Srebp-Men*. We have now included this dataset in this revised manuscript (new Supplementary Fig. 3b).

b

Supplementary Fig. 3b, Night sleep amount in control (n = 23), *Cyfp*^{85.1/+} (n = 15), *Cyfp*^{85.1}/*Aralar*^{MI07552} (n = 30) and *Aralar*^{MI07552/+} flies (n = 27). One-way ANOVA followed by Sidak's multiple comparison test. **p = 0.0062, ***p = 0.0003 ****p < 0.0001, n.s. = not significant. Mean \pm S.E.M.

5) In Kanellopoulos et al, the same lab mapped the effect of *Cyfp* mutant protein onto specific GABAergic neurons. It would be interesting to know whether, e.g., knocking down *Cyfp* in these subsets of GABAergic neurons has an effect on sleep?

The analysis of night-time sleep behaviour of fly lines expressing *UAS-Cyfp-IR*^{1/2} in GABAergic circuitries, where *Cyfp* haploinsufficiency had an effect on social behaviour (Kanellopoulos et al., 2020, PMID: 32200800), did not reveal differences suggesting that these circuitries are not involved in the night-time sleep phenotype. We added this dataset in new Supplementary Fig. 3a.

Supplementary Fig. 3a Night sleep amount in *APLGal4/+* (n = 36), *UAS-Cyfp-IR¹* (n = 38), *UAS-Cyfp-IR²* (n = 20), *Gad1Gal4> UAS-Cyfp-IR¹* (n = 40), *Gad1Gal4> UAS-Cyfp-IR²* (n = 20); *Gad1Gal4/+* (n = 40), *APLGal4> UAS-Cyfp-IR¹* (n = 40), *APLGal4> UAS-Cyfp-IR²* (n = 20). One-way ANOVA followed by Sidak's multiple comparisons test. **p* = 0.0403, ****p* = 0.0003, n.s. = not significant, mean ± S.E.M.

6) Page 8, line 12 & page 15, line 6: based on the sleep deprivation experiments and the NADP⁺/NADPH results, the authors propose that the mutants show a lower need for sleep. Yet, how do they reconcile this claim with the fact that the mutants show defects in associative learning (Woo et al (2019, Biol Psychiatry), a hallmark of cognitive defects due to sleep deprivation? Couldn't it also be that the sleep homeostasis machinery is defect, resulting in a reduction of homeostatic sleep?

We apologize that this point of the discussion was misleading. We believe that *Cyfp* heterozygous flies have deficits in sleep homeostasis, and that the dysregulation of the NADP⁺/NADPH ratio is the driving force: in Fig. 1e and Supplementary 3i, after 12hrs or 6 hrs sleep deprivation, *Cyfp* mutants exhibit reduced sleep rebound. In new Fig. 2b we showed that *Cyfp^{85.1/+}* flies after sleep deprivation do not show increase in NADP ratio, as it was observed in control flies sleep deprived, demonstrating that they have impairments in homeostatic sleep pressure. Further, the NADP⁺/NADPH ratio over 24 hrs mirrors deficits at the night-time sleep onset in *Cyfp* mutants.

Therefore, it is quite possible that the defects in associative learning (Woo et al, 2019) are a consequence of sleep deprivation, similar to the *cv-c* mutants characterised in Donlea et al., 2014 (PMID: 24559676). This mutant, encoding a Rho-GTPase-activating protein, exhibits decreased sleep time and sleep rebound, memory deficits comparable to those observed after sleep loss. We added this notion to the discussion as follows:

"In addition, Cyfp heterozygous flies exhibit impaired associative learning³⁰, a hallmark of sleep loss,^{96, 97, 98, 99} suggesting that sleep disorders and/or chronic sleep deprivation might contribute and exacerbate the cognitive defects reported in Cyfp^{85.1/+} flies."

Minor points:

1) Typo in Extended Data Figure 2e: third group from the left should be 'R23E10' instead of R23E18'. Thank you for highlighting this typo that has been corrected in the revised version.

2) In Extended Data Figure 2f, some, e.g., 'Mai179Gal4', but not all of the Gal4 names are written in italic characters. The same applies to other figures throughout the nomenclature. Could the authors homogenize the nomenclature?

We thank the reviewer for this remark, the nomenclature is now in italic.

3) Page 12, Line 24/25: does this imply that the malic enzyme is only over-active at ZT16 or regulated in a circadian-dependent manner? Based on the previously published proteomics results it doesn't seem to be the case. It would be informative if the authors would comment on that.

Following the reviewers' comment we performed RT-qPCR to detect *Men* mRNA in wild-type during undisturbed sleep (noSD) or sleep-deprived (SD) as in Cirelli et al., 2005 PMID: 16001966. Our findings are also consistent with published literature showing that *Men* expression is associated with wakefulness and upregulated upon sleep deprivation in wild-type flies (Cirelli et al., 2005, PMID: 16001966) (new Fig. Suppl. 5d) and fly models of insomnia (Seugnet et al., 2009, PMID: 19494137). *Cyfp* and *Srebp* mRNA levels did not change between noSD and SD conditions in wild-type flies (Fig. 3 for the reviewer only).

In addition, investigating the database for circadian rhythm (<http://cgdb.biocuckoo.org/download.php>, last update July 2022), we could not find *Men* protein to be under circadian rhythm regulation in brain.

Fig. 3 for the reviewer only, RT-qPCR to assess *Men* (A), *Cyfp* (B) and *Srebp* (C) gene expression level in control flies during undisturbed sleep (noSD, n = 4) or sleep-deprived (SD, n = 4). n = pool of 15 heads. Two-tailed Student's t-test, *p<0.05. Mean ± S.E.M.

4) Page 14, line 20: 'Thus, the NADP+/NADPH ratio reflects neurons' sleep pressure.' This was only shown for dFB neurons in Kempf et al (2019). The authors should specify that.

We thank the reviewer for this remark, this aspect has been specified in the discussion.

5) Page 13, line 21/22: 'supporting the idea that lipid synthesis and remodeling is important during sleep, ...'. If so, then why do *Cyfp* mutants sleep less despite an upregulation of genes involved in lipid synthesis? What do the authors speculate?

We agree with the reviewer's point which indeed merits a discussion. Sleep restriction and sleep deprivation have been shown to affect lipid profiles at the level of transcriptome and metabolome in mice and humans. These findings, supporting the idea that lipid synthesis and remodeling is relevant during sleep to restore the metabolic pool used for brain activity during wakefulness. Our hypothesis is that increased expression of genes involved in energy metabolism and lipid synthesis during the day affects the homeostatic sleep regulation in *Cyfp* mutant flies presumably reducing the homeostatic pressure at the night onset.

Reviewer #4 (Remarks to the Author):

In this manuscript, Mariano et al provide an interesting and thorough description of a sleep phenotype associated with the *Cyfp* gene and its epistatic pathway. The authors show that *cyfp* mutant heterozygous flies have a short-sleep nocturnal phenotype and possible homeostatic misregulation and they propose that upregulation of SREBP and *Men* are responsible for this phenotype.

The work features an impressive amount of work, generally well-controlled and well presented. I personally do not like the overly translational approach in the manuscript, insisting over and over again on possible links with human disease that may or may not exist; I think the manuscript ultimately suffers from adopting this approach and the findings end up being secondary but I recognise this may be the authors' style and they are entitled to use whichever style they prefer.

Overall, I think the work is solid and self-standing and I do not have any requests for further experiments. I only have some minor suggestions/requests mostly regarding data presentation and wording of the conclusions:

We appreciate the reviewer's positive comments on our work. In the revised version we have followed his/her suggestions and performed the requested experiments.

1) The term haploinsufficient refers to genes, not to animals. The use of "haploinsufficient flies" (e.g. line 3 pg5) is technically inappropriate. The flies are heterozygous for the gene mutation whilst the gene is haploinsufficient.

We thank the reviewer for this important remark. We have now replaced the term haploinsufficient with heterozygous in the revised manuscript.

2) An EEG/MRI abnormality is not a "sleep disturbance" (line 12 pg 4). It may in some cases correlate with one but a change in EEG pattern is not a disturbance per se.

We thank this important comment and have now removed the citations referring to EEG/MRI abnormalities that did not describe sleep deficits and left the references that describe insomnia and sleep disorders in these patients.

3) In the figures, please add more information regarding the number of animals used. Figure 1a-c, for instance, says the n is 75 without specifying the n for each condition. Is it 75 per condition or overall?

We have now added in the figure legend the n for each genotype and summarised this information in a new Supplementary Data 6.

4) Replace all bar plots (such as the ones in fig 1bc,d) with something statistically more appropriate showing the actual distribution of data (as you do for instance in 1e).

5) Some plots have dots as values but no indication of what they represent. For instance, I assume the dots in the rightmost panel of 1e or in 4e represent outliers while the dots in 2a or 4c are actual data points? It would be good to stick to more consistent data visualisation.

We have modified the graphs showing each individual data point, all the figures are now consistent, many thanks for this remark.

6) Do you have any information regarding the expression pattern of *Cyfp* in the adult brain? Schenck et al characterised its expression during development but what about adulthood? The results in Fig. 4d suggest its expression (or at least its action on SREBP) is limited to the ellipsoid bodies and that is confirmed by the 23E10 KD experiment but do you know if the gene is expressed anywhere else? I am asking because of the results in fig.2 and fig 5b which do not seem really compatible with a very localised expression of the gene. If you do have data about *cyfp* expression please show it. If you do not have those data and do not want to obtain them please add some caveats/discussion regarding the results in fig. 2 and 5. I am just quite surprised that a change in enzymatic activities in only a few neurons can be picked up doing biochemical assays in the whole brain.

We understand the reasoning of having the expression pattern of *Cyfp* in the entire brain because - as of today - it has not been shown in the adult brain. Unfortunately, there are no good antibodies to detect the endogenous CYFIP protein. We have tried to have them produced by 21st Century and the result is not compelling. We therefore opted for the generation of transgenic fly with a tagged version of CYFIP using the CRISPR-Cas9 technology in which the endogenous CYFIP has a HA-tag at the N terminus. Immunostaining for HA (in green) and for nc82 (anti-Brp, pre-synaptic marker) (in magenta) revealed that CYFIP^{N-HA} is expressed in most of the brain regions and at synapses (new Suppl. Fig. 1I).

Supplementary Fig. 1, Representative pictures of whole brain immunohistochemistry of CYFIP^{N-HA} flies for anti-HA (green), anti-BRP (nc82) (purple) and merge. Scale bar = 50 μ m.

7) Please provide more details regarding the statistical analysis used to study the overlap of genes shown in fig. 3 and discussed on pg 9. A hypergeometric distribution is mentioned in the figure legend without further details.

We thank the reviewer for these comments. We have now added in the methods section a sentence stating that the hypergeometric distribution analysis in the RNA-seq datasets was performed using R (<https://stat.ethz.ch/R-manual/R-devel/library/stats/html/Hypergeometric.html>) and the exact p-value in the result section.

Was the same analysis done for the data in figure 4a? with what significance?

The same hypergeometric distribution analysis was indeed also performed for Fig. 4a. In this context, we obtained a p-value of 0.98 because even if the overlap is for 129 genes, it is less than 10% (129 genes from 1303 and 1366) and not significant. Next, we performed a new analysis reasoning on which genes of the SREBP-overexpression datasets are relevant to wakefulness (as we did in Fig. 3e for the *Cytip* mut dataset). Here we identified a subset of SREBP-regulated genes associated to wakefulness (21 genes, new Supplementary Data 5). When comparing the wakefulness associated genes dysregulated in *Cytip*^{85.1/+} and in the SREBP-OE we identified 10 genes (new Supplementary Table 1). The hypergeometric distribution analysis on the overlap of these 2 datasets reveals a significant p-value ($p = 3.467^{-21}$, new Fig. 4b), consistent with all the biological observations in this work.

Fig. 4a, Venn diagram showing the overlap between the *Cytip* mutants RNA-Seq dataset at ZT16 and the dataset of dysregulated mRNAs upon SREBP-OE^{70, 71, 72} genes $p = 0.98$, assessed by hypergeometric analysis. **b**, Venn diagram showing the overlap between *Cytip* wakefulness associated genes (WAG) and the

WAGs dysregulated upon SREBP-OE. $p = 3.467^{-21}$ assessed by hypergeometric analysis.

8) The epistatic experiments shown in fig. 5 are very nice but the only evidence that Men is involved with this process is coming from the one experiment shown in fig. 5c and 5b. While suggestive, I am not convinced this is enough evidence for such a definitive claim. I recommend using a more cautious approach in your conclusions.

We agree with reviewer's comment that is important to further validate the link between NADP⁺/NADPH and Men. We performed targeted mass spectrometry to detect with higher sensitivity the NADP ratio in Men mutant flies. Importantly, Men mutants show an opposite phenotype of *Cytip* het flies, with increase NADP⁺/NADPH (new Fig. 5a), and increased night-time sleep (new Fig. 5c). In addition, abrogation of Men in the *Cytip*^{85.1/+} background can fully rescue the NADP ratio. These data confirm that Men modulation affects NADP ratio and night-time sleep according to our model. We decided to replace the former figure 5a obtained using the kit Colorimetric NADP⁺/NADPH kit from Abcam with the new dataset generated with the HILIC-MS/MS.

Fig. 5a, NADP⁺/NADPH ratio at ZT12 by HILIC-MS/MS in control (n = 10), *Cyfip*^{85.1/+} (n = 10), *Cyfip*^{85.1}/*Srebp*¹⁸⁹ (n = 5), *Srebp*^{189/+} (n = 5), *Cyfip*^{85.1}/*Men*^{BG02790} (n = 5), *Men*^{BG02790/+} (n = 5). One-way ANOVA test followed by Sidak's multiple comparisons test, **p* = 0.0359, control vs. *Cyfip*^{85.1/+}; **p* = 0.0188, *Cyfip*^{85.1/+} vs. *Cyfip*^{85.1}/*Men*^{BG02790}; ***p* = 0.0075, *Cyfip*^{85.1}/*Men*^{BG02790} vs. *Men*^{BG02790/+}; ****p* = 0.0001, control vs. *Men*^{BG02790/+}. n = group of 45 fly heads/sample. Mean ± S.E.M.

Fig. 5c, Quantification of night-time sleep amount in control (n = 71), *Cyfip*^{85.1/+} (n = 51), *Cyfip*^{85.1}/*Men*^{BG02790} (n = 60) and *Men*^{BG02790/+} (n = 35) flies. One-way ANOVA followed by Sidak's multiple comparisons test, **p* = 0.034, control vs. *Men*^{BG02790/+}; **p* = 0.041, *Cyfip*^{85.1/+} vs. *Cyfip*^{85.1}/*Men*^{BG02790}; ***p* = 0.0046, control vs. *Cyfip*^{85.1}/*Men*^{BG02790}; *****p* < 0.0001, control vs. *Cyfip*^{85.1/+}. Mean ± S.E.M.

9) On the same line: please amend the title to be more accurate in its description. "drives" and "control" are two very strong words for something that affects only a relatively small part of the phenotype.

We thank the reviewer for this comment, we modified the title of the manuscript as: "SREBP modulates the NADP⁺/NADPH cycle to control night-time sleep"

REVIEWER COMMENTS

Reviewer #1 (Remarks to the Author):

The authors have done an excellent job addressing all of the reviewer comments. They provide a substantial amount of new data and reanalysis, which has now made their work more convincing. It was also good to see consistency among the different reviews, and that the authors went out of their way to address every single point thoroughly.

I have only two remaining suggestions, that the authors and editors can decide how to handle.

1. My suggestion for a schema up front in Figure 1 was not in relation to Trikinetics hardware implementation, which is trivial. Rather, it related to placing the molecule in question in some cellular pathway already, schematically. It is often hard for the average reader to jump straight into sleep profile data involving mutant acronyms, and it does help to at least have some idea of the biology at stake, visually. This might be a membrane schema with the multiple players involved, including the first proteins of interest, and how they interact. Why should we be interested in this, and can we already imagine a consequence (on sleep) of mutating these proteins? Try to provide this visually, up front.

2. The authors now provide most of the daytime sleep data as requested, to compare with nighttime sleep. Thank you. It would however be a missed opportunity if the authors don't provide a little more discussion as to why there might be such a dichotomy, beyond the referenced observation that arousal thresholds are different. Why might nighttime sleep be qualitatively different? Are different functions being served? How might this force us to reconsider how the field studies sleep in this model? Are metabolic processes not relevant for daytime sleep? I recommend the authors provide some interesting discussion here, beyond just stating that they are different.

Reviewer #2 (Remarks to the Author):

In this revised manuscript, the authors perform a significant number of new experiments, adding new data and replacing some of the original data. Although their model connecting Cyfip -> lipid metabolism -> SREBP/MEN -> NADP/NADPH -> sleep pressure continues to be interesting, there remain concerns regarding whether the experimental data adequately support this model.

1. In new Fig. 2a, the authors use a new approach (HILIC-MS/MS) to quantify the NADP/NADPH ratio across time. However, the curves shown are not clearly linked to sleep pressure. For example, in control flies, the NADP/NADPH ratio decreases across the day (except for the small peak at ZT12) and increases across the night when flies should be sleeping. The authors claim that reduced sleep pressure around ZT12 is due to the small NADP/NADPH peak at ZT12 being lost in Cyfip heterozygotes. However, there is a greater difference at ZT0—according to their argument, wouldn't there also be a reduction in sleep at ZT0 in Cyfip heterozygotes? The more likely explanation is that NADP/NADPH is a downstream marker of increased locomotor activity.

2. Re: Figure 1d, the authors argue that the differences in sleep lost following light pulses are not due to changes in arousal threshold because similar or lower numbers of Cyfip heterozygotes exhibit a >2% reduction in sleep following these stimuli. While the authors may ultimately be correct, it is still difficult to definitively rule out changes in arousal threshold without more experimentation. First, most researchers use any awakening as the criteria to define arousal threshold. Second, if the authors choose not to use that criterion, they should perform a careful analysis of different "arousal thresholds" to more clearly demonstrate that arousal threshold is not affected.

3. There remains concern about imprecision about the relevant circuits. The authors use Ras2-Gal4,

which is known to have non-neuronal expression, but claim that in adult flies Ras2-Gal4 is exclusively expressed in neurons. They cite Gouzi, 2011 to support this claim, but I didn't find any evidence in that paper that Ras2-Gal4 is only expressed in neurons in adult flies? Instead, Gouzi 2011 cites Walker, 2006, which in turn cites Salzberg, 1993, and Salzberg 1993 describes expression in the adult reproductive system. Thus, based on that observation, Ras2-Gal4 is not strictly pan-neuronal in adult flies. They are unable to reproduce their Cyfip knockdown phenotype using nsyb-Gal4 and claim that ras2-Gal4 and nsyb-Gal4 are not perfectly overlapping in their neuronal expression, but do not provide these data for the reviewers. In addition, there are other pan-neuronal drivers that could be used such as R57C10-Gal4. In light of their inability to find a difference using SREBP or MEN overexpression with R23E10-Gal4, conclusively demonstrating that Cyfip acts in neurons would seem to be essential.

4. In Fig. 5a, the authors show genetic interactions between Cyfip heterozygotes and SREBP and MEN alleles and use these data to argue that the NADP/NADPH ratio is "rescued." However, there is concern that the interactions shown are simply additive, and thus lack specificity. Cyfip heterozygotes exhibit a reduced NADP/NADPH ratio while reduction of SREBP or MEN on their own results in an increase in NADP/NADPH ratio; when combined, the transheterozygotes exhibit an intermediate phenotype. In other words, if one were to take any random genetic manipulation that led to an increase of NADP/NADPH ratio, it is possible that they would also "rescue" the reduced NADP/NADPH ratio seen in Cyfip heterozygotes.

5. In Figure S5d, they show that Men transcript levels rise with sleep deprivation. However, according to their model, if Men acts upstream as part of the signaling mechanism for sleep pressure, then one would expect that Men levels would be reduced with sleep deprivation. Instead, the increased Men levels (which should reduce NADP/NADPH ratio and lower sleep pressure) appear to be a downstream compensatory response, consistent with their NADP/NADPH timecourse data.

Minor comments:

1. There is awkward language and minor errors in the manuscript, and the text would benefit from careful editing. For example, line 139, "5 min of light pulse." Line 141 "loose". Line 313, "The malic enzyme pairs SREBP to sleep behavior." Line 339, "3 key aspects" but then they list 4 aspects. Line 119, unnecessary comma after "SREBP activity."

2. "Fig 3 only for reviewers" on the top of page 14 of the rebuttal, should be provided in its entirety in the supplemental info.

3. Re: the GFP measurements in Figures 4e and f, usually quantification is more linear when using native fluorescence?

4. In multiple panels showing fly sleep behavior at 29 deg and 30 deg, the sleep behavior looks unexpectedly normal for control flies. Usually, elevated temperatures lead to obvious changes in sleep/wake behavior (reduced sleep at night for example).

5. Figures S3e and S3f, the activity traces in LD look remarkably "clean." Also, if one were to estimate sleep based on those activity traces, they would not seem to correspond to the sleep data shown in Figure 1a?

Reviewer #3 (Remarks to the Author):

I would like to thank the authors for addressing my comments and other Reviewer's comments. The revision significantly improved the manuscript and I would like to recommend the revised version for publication at Nature Communications.

Minor points:

- Typo in Supplementary Figure 3c: third group from the left should be 'R23E10' instead of R23E18'.
- Typo in Supplementary Figure 3d: third group from the left should be 'R23E10' instead of R23E18'.
- Page 15, lines 351-353, 'The arousal threshold is higher at night compared to daytime, suggesting that night-time sleep might be associated with sleep homeostasis' citing Ishimoto et al 2012 as supporting reference. I could not find any claim supporting the author's statement in the reference cited. The authors should revise this sentence.

Reviewer #4 (Remarks to the Author):

Having reviewed the responses given by the authors to my comments - and to the comments of the other 3 reviewers - I believe the manuscript is now improved and I have no further comments or suggestions.

Point by Point

Mariano et al. NCOMMS-21-46266-T

“SREBP drives the NADP⁺/NADPH cycle to control night sleep”.

We would like to thank the reviewers for their positive feedback supporting publication of our manuscript in Nature Communications. We have also taken the additional comments of the reviewers into consideration and provide additional data in support of our model. We hope that the current revised version will be suitable for publication.

REVIEWER COMMENTS

Reviewer #1 (Remarks to the Author):

The authors have done an excellent job addressing all of the reviewer comments. They provide a substantial amount of new data and reanalysis, which has now made their work more convincing. It was also good to see consistency among the different reviews, and that the authors went out of their way to address every single point thoroughly.

We really thank the reviewer for appreciating the extensive work we did to address the relevant comments we received on our manuscript.

I have only two remaining suggestions, that the authors and editors can decide how to handle.

1. My suggestion for a schema up front in Figure 1 was not in relation to Trikinetics hardware implementation, which is trivial. Rather, it related to placing the molecule in question in some cellular pathway already, schematically. It is often hard for the average reader to jump straight into sleep profile data involving mutant acronyms, and it does help to at least have some idea of the biology at stake, visually. This might be a membrane schema with the multiple players involved, including the first proteins of interest, and how they interact. Why should we be interested in this, and can we already imagine a consequence (on sleep) of mutating these proteins? Try to provide this visually, up front.

We understand the suggestion of the reviewer and thought extensively how to make a scheme. At the end we felt that the concept we developed had an important overlap with the graphical abstract so we would like to keep the graphical abstract only.

2. The authors now provide most of the daytime sleep data as requested, to compare with nighttime sleep. Thank you. It would however be a missed opportunity if the authors don't provide a little more discussion as to why there might be such a dichotomy, beyond the referenced observation that arousal thresholds are different. Why might nighttime sleep be qualitatively different? Are different functions being served? How might this force us to reconsider how the field studies sleep in this model? Are metabolic processes not relevant for daytime sleep? I recommend the authors provide some interesting discussion here, beyond just stating that they are different.

We would like to thank the reviewer for this comment. We have now expanded on a few concepts in the discussion. Specifically, on pg 15 we state:

“Notably, night-time and daytime sleep features differ in flies⁸¹. Brain activity, measured as local field potential (LFP) and the responsiveness to stimuli during night compared to daytime sleep, suggesting different levels of homeostatic pressure for these processes along the 24 hrs^{82, 83}. While these differences have been reported, the function of daytime and night-time sleep in *Drosophila* remains unclear, although some hypotheses point to memory consolidation and synaptic homeostasis^{82, 83}. Therefore, the identification of factors, as *Cyfp*, regulating differentially daytime and night-time sleep helps to shed light on the characteristics of these temporally distanced sleep. Only few genetic factors and mechanisms have been so far

identified as regulators of daytime and night-time sleep, for instance hormones, immune response, oxidative stress and lipid binding molecules⁸¹.”

Reviewer #2 (Remarks to the Author):

In this revised manuscript, the authors perform a significant number of new experiments, adding new data and replacing some of the original data. Although their model connecting Cyfip -> lipid metabolism -> SREBP/MEN -> NADP/NADPH -> sleep pressure continues to be interesting, there remain concerns regarding whether the experimental data adequately support this model.

We would like to thank the reviewer for finding the link CYFIP-NADP/NADPH interesting. We hope that with the additional experiments added in this additional revision we successfully convince him/her about our model.

1. In new Fig. 2a, the authors use a new approach (HILIC-MS/MS) to quantify the NADP/NADPH ratio across time. However, the curves shown are not clearly linked to sleep pressure. For example, in control flies, the NADP/NADPH ratio decreases across the day (except for the small peak at ZT12) and increases across the night when flies should be sleeping. The authors claim that reduced sleep pressure around ZT12 is due to the small NADP/NADPH peak at ZT12 being lost in *Cyfip* heterozygotes. However, there is a greater difference at ZT0—according to their argument, wouldn't there also be a reduction in sleep at ZT0 in *Cyfip* heterozygotes? The more likely explanation is that NADP/NADPH is a downstream marker of increased locomotor activity.

We thank the reviewer for this comment.

We believe that – in our experimental conditions - the NADP⁺/NADPH oscillation is not a merely downstream marker of increased locomotor activity. This conclusion is based on the following experiment: we quantified the locomotion amount (beam crossing quantification) in control and *Cyfip* heterozygous flies at ZT0 and ZT12, the moment where the flies show the highest activity – and we did not observe any differences between control and *Cyfip* heterozygous flies.

In addition, if NADP⁺/NADPH would be a downstream marker of increased locomotor activity we would have not observed differences in locomotion between ZT8, ZT12 and ZT16 in *Cyfip* heterozygous flies, as the NADP⁺/NADPH is not oscillating between these 3 time points in *Cyfip* heterozygous flies (Figure 1 for the reviewer).

Figure 1 for the reviewer: Beam crossing/ hour of control and *Cyfip*^{85.1/+} flies at ZT0, ZT8, ZT12. Unpaired Student's t-test. Control_ZT16 vs. *Cyfip*^{85.1/+}_ZT16, $p^{****} < 0.0001$. Control (n = 45) and *Cyfip*^{85.1/+} (n = 46) flies. Mean \pm S.E.M.

In addition, in Figure 2a we don't link the observed pattern with homeostasis sleep pressure but we speculate that the ratio NADP⁺/NADPH reflects the behavioral state of the fly, i.e., peaking at high wakefulness periods (ZT0 and ZT12) before sleep initiation and descending at ZT16 when sleep need is gradually reduced and sleep behavior increases (Fig. 2a).

The experiment that shows that NADP⁺/NADPH is linked to homeostatic sleep pressure is summarized in Fig. 2b in which, following a gold standard protocol, we show that NADP⁺/NADPH is increased at sleep deprivation.

2. Re: Figure 1d, the authors argue that the differences in sleep lost following light pulses are not due to changes in arousal threshold because similar or lower numbers of *Cyfp* heterozygotes exhibit a >2% reduction in sleep following these stimuli. While the authors may ultimately be correct, it is still difficult to definitively rule out changes in arousal threshold without more experimentation. First, most researchers use any awakening as the criteria to define arousal threshold. Second, if the authors choose not to use that criterion, they should perform a careful analysis of different "arousal thresholds" to more clearly demonstrate that arousal threshold is not affected.

We thank the reviewer for these suggestions, and we performed a new analysis of the data relative to the arousal experiments using the awakening as criteria to define the not responder and responder flies. In this new analysis flies were divided in "responders" flies with no activity 5 min before the stimulus and exhibiting beam crossings within 5 min after the light pulse" and "not responders": flies with no activity 5 min before the stimulus and not exhibiting any response withing 5 min after the light pulse (as in Kayser et al., 2014, Science PMID: 24744368) (Figure 2 for the reviewer).

Statistical analysis (Fisher's exact test, between "responders" and "not responders" at ZT16, ZT18 and ZT20 in control and *Cyfp*^{85.1/+} flies did not reveal differences between the 2 genotypes. Therefore, we conclude that the *Cyfp*^{85.1/+} are not largely hyper-aroused because after providing the light stimuli at night-time we did not observe more *Cyfp*^{85.1/+} responders compared to control flies.

Criteria:			
Awake: flies that are moving in the 5 minutes before the stimulus			
Not responding: flies that were not moving in the 5 minutes before the stimulus and that did not moved in the 5 min after the stimulus			
Responding: flies that were not moving in the 5 minutes before the stimulus and that moved within the 5 minutes after the stimulus			
Control			
	ZT16	ZT18	ZT20
Awake	2	2	8
Not responding	0	1	2
Responding	22	21	14
Cyfp^{85.1/+}			
	ZT16	ZT18	ZT20
Awake	20	15	12
Not responding	0	2	1
Responding	10	13	17

Figure 2 for the reviewer: Number of "responding" flies showing waking response after light stimulus and "not responding" flies that did not respond to light stimuli. Shown are the data for control and *Cyfp*^{85.1/+} flies at three time points when the arousal stimulus was provided (ZT16, ZT18 and ZT20). Fisher's exact test between responding and not responding flies in control and *Cyfp*^{85.1/+} at ZT16, ZT18 and ZT20, n.s.

3. There remains concern about imprecision about the relevant circuits. The authors use Ras2-Gal4, which is known to have non-neuronal expression, but claim that in adult flies Ras2-Gal4 is exclusively expressed in neurons. They cite Gouzi, 2011 to support this claim, but I didn't find any evidence in that paper that Ras2-Gal4 is only expressed in neurons in adult flies? Instead, Gouzi 2011 cites Walker, 2006, which in turn cites Salzberg, 1993, and Salzberg 1993 describes expression in the adult reproductive system. Thus, based on that observation, Ras2-Gal4 is not strictly pan-neuronal in adult flies. They are unable to reproduce their Cyfip knockdown phenotype using nsyb-Gal4 and claim that ras2-Gal4 and nsyb-Gal4 are not perfectly overlapping in their neuronal expression, but do not provide these data for the reviewers. In addition, there are other pan-neuronal drivers that could be used such as R57C10-Gal4.

We thank the reviewer for this comment that we have taken into consideration. We cited Gouzi et al., 2011 as they used the Ras2-Gal4 to drive their gene of interest in the adult CNS. We are aware that at the larval stage the Ras2 is also expressed in salivary glands, guts, central brain region and ventral nerve cord (Saltzberg et al., 1993; Walker et al., 2006). We have therefore revised the text as follows

"We used the neuronal Ras2Gal4 driver, coupled with TubGal80^{ts} to specifically avoid lethal effects of Cyfip abrogation during larval development⁵⁶, that is largely expressed in the overall neuronal population of adult fly CNS⁶², although not exclusively during larval development^{63, 64}." mentioning for the reader that the Ras2-Gal4 is not exclusively pan-neuronal in larval flies, since it targets neurons but also non-neuronal cells in the body of the fly.

In addition, we are now providing confocal images of adult brain from the nSyb-Gal4> UAS-mCD8::GFP and Ras2-Gal4> UAS-mCD8::GFP flies stained with the neuronal marker Elav.

We now show that the expression pattern of nSyb-Gal4 and the Ras2-Gal4 drivers is not completely overlapping.

As an example, in the nSyb-Gal4 flies we have observed cells that are Elav+ and GFP- in the region around the antennal lobes (Figure 3A for the reviewer) marked by the white arrows (2 independent representative images) suggesting that nSyb is not expressed in all neuronal cells (Figure 3B). In addition, in the nSyb-Gal4 flies we observed cells that are GFP+ and Elav- (marked by the yellow arrow) (Figure 3B).

As an additional example suggesting a non 100% overlapping pattern is shown in Figure 3C. In the Ras2-Gal4 the group of cells (around the antennal lobe) are all Elav+ and GFP+ (Figure 3C) suggesting that nSyb is expressed in all neuronal cells here analyzed.

We can conclude that Ras2-Gal4 in adult flies targets neurons with a pattern that is not 100% overlapping with the nSyb-Gal4 driver.

Figure 3 for the reviewer: **A.** Graphic of the *Drosophila* brain showing pan-neuronal labeling (red dots depict cell nuclei). The square black box correspond to the regions analyzed by confocal in B and in C (*nSyb-Gal4*> *UAS-mCD8::GFP* and *Ras2-Gal4*> *UAS-mCD8::GFP* respectively) for anti-Elav (in red) and anti-GFP (in green). AL: Antennal lobe, MB: mushroom

body. White arrows point to Elav+ and GFP- cells and the yellow arrow to a cell GFP+ and Elav- cell. Scale bar = 50um.

In light of their inability to find a difference using SREBP or MEN overexpression with R23E10-Gal4, conclusively demonstrating that *Cyfp* acts in neurons would seem to be essential.

We thank the reviewer for this suggestion. While monitoring *Men* overexpression in this circuit is a great suggestion, unfortunately a strain with mutations in *Men* is not available and the generation of additional specific fly lines will be considered for future studies.

4. In Fig. 5a, the authors show genetic interactions between *Cyfp* heterozygotes and SREBP and MEN alleles and use these data to argue that the NADP/NADPH ratio is “rescued.” However, there is concern that the interactions shown are simply additive, and thus lack specificity. *Cyfp* heterozygotes exhibit a reduced NADP/NADPH ratio while reduction of SREBP or MEN on their own results in an increase in NADP/NADPH ratio; when combined, the transheterozygotes exhibit an intermediate phenotype. In other words, if one were to take any random genetic manipulation that led to an increase of NADP/NADPH ratio, it is possible that they would also “rescue” the reduced NADP/NADPH ratio seen in *Cyfp* heterozygotes.

The involvement of *Men* was identified from the unbiased RNA-seq data and the involvement of SREBP derived by the list of dysregulated mRNAs – targets of SREBP – that we found in the RNA-seq data.

While we have identified SREBP and *Men* as key players acting on the NADP ratio to well rescue/ameliorate *Cyfp* mutants sleep deficits, we do not exclude the involvement of other players.

Of note, a network of 4 genes maintains the NADP⁺/NADPH balance and supply reducing power for lipogenesis and antioxidation (Merritt 2009): the cytosolic malic enzyme (MEN), the cytosolic isocitrate dehydrogenase (IDH), and the two oxidative enzymes of the pentose shunt, glucose-6-phosphate dehydrogenase (G6PD) and 6-phosphogluconate (6PGD). In *Cyfp* heterozygous condition MEN is the one that appears significantly dysregulated. Therefore, we focused specifically on this enzyme. Even if there are other molecules that can affect the NADP/NADPH ratio and modulate sleep, our conclusions won't not be affected.

5. In Figure S5d, they show that *Men* transcript levels rise with sleep deprivation. However, according to their model, if *Men* acts upstream as part of the signaling mechanism for sleep pressure, then one would expect that *Men* levels would be reduced with sleep deprivation. Instead, the increased *Men* levels (which should reduce NADP/NADPH ratio and lower sleep pressure) appear to be a downstream compensatory response, consistent with their NADP/NADPH timecourse data.

We thank the reviewer for raising this point. We have observed that in control flies the NADP⁺/NADPH increases upon sleep deprivation (initial submitted Figure 2b), the same occurs for *Men* mRNA transcript (initial submitted Figure S5d).

Following the reviewer suggestion, we assessed *Men* activity in control and *Cyfp* heterozygous flies during sleep (noSD) and after sleep deprivation (SD) using the same protocol to assess NADP⁺/NADPH and *Men* mRNA levels (submitted Figure 2b, S5d). In control flies, we did not observe any change in *Men* activity after sleep deprivation, while an increase in *Men* activity is observed in *Cyfp* heterozygous flies (Figure 4 for the reviewer) consistent with the deficit in the NADP⁺/NADPH ratio (submitted Figure 2b).

Since *Cyfp* is a translational repressor, we hypothesized that *Men* mRNA translation could be under translational regulation by *Cyfp*. A follow up study could explore such a possibility identifying *Cyfp* mRNA targets and in condition of sleep and sleep deprivation. We believe addressing this point is beyond what we perceive to be the scope of the current work.

A**B**
Figure 4 for the reviewer: (A) Men activity in control flies during undisturbed sleep (noSD, $n = 5$, pool of 10 fly heads) and sleep-deprived (SD, $n = 3$, pool of 10 fly heads). Two-way repeated measures ANOVA followed by Tukey's multiple comparisons test, interaction time-condition (noSD vs SD) $p = 0.9$, not significant. (B) MEN activity in *Cyfip*^{85.1/+} flies during undisturbed sleep (noSD, $n = 5$, pool of 10 fly heads) and sleep-deprived (SD, $n = 3$, pool of 10 fly heads). Two-way repeated measures ANOVA followed by Tukey's multiple comparisons test, interaction time-condition (noSD vs SD) **** $p < 0.0001$, $F_{(68,408)} = 5.266$. *Cyfip*^{85.1/+} noSD vs *Cyfip*^{85.1/+} SD, ** $p = 0.0026$. Mean \pm S.E.M.

Minor comments:

1. There is awkward language and minor errors in the manuscript, and the text would benefit from careful editing. For example, line 139, "5 min of light pulse." Line 141 "loose". Line 313, "The malic enzyme pairs SREBP to sleep behavior." Line 339, "3 key aspects" but then they list 4 aspects. Line 119, unnecessary comma after "SREBP activity."

We have carefully revised the manuscript. We hope that the awkward language has now been corrected.

2. "Fig 3 only for reviewers" on the top of page 14 of the rebuttal, should be provided in its entirety in the supplemental info.

We thank the reviewer for this suggestion. We have now added Fig. 3 for the reviewer in the manuscript as Supplementary figure 5e and 5f.

3. Re: the GFP measurements in Figures 4e and f, usually quantification is more linear when using native fluorescence?

We thank the reviewer for this remark, however even if the range was not linear, the difference between control and mutant remains valid. In addition, all samples (controls and mutants) have been processed at the same time, with the same reagents and solutions and acquired at the same time using the same confocal settings.

4. In multiple panels showing fly sleep behavior at 29 deg and 30 deg, the sleep behavior looks unexpectedly normal for control flies. Usually, elevated temperatures lead to obvious changes in sleep/wake behavior (reduced sleep at night for example).

We thank the reviewer for this comment. We agree that increased temperature induces an increase in daytime sleep and reduced sleep at night in flies, as reported by different laboratories, below an example here reporting a Supplementary figure 1 in Parisky et al., 2016 PMID: 26972320.

Figure S1. Parisky et al., 2016. PMID: 26972320. (A) Baseline sleep per 30 min comparing 25C to 29C after a temperature shift at ZT0 in wild type Canton S male flies (n = 91).

Consistent with what is reported in the literature, we observe an overall reduction of the nighttime sleep in all the fly genotype – please compare nighttime sleep at 18C (left panel) and 29C (right panel). We also acknowledge that these changes in sleep amount might not be “obvious”. In our experiments we used male flies and, as also pointed out by in Parisky et al., 2016 “increase in daytime sleep and decreased in nighttime sleep in males is less evident, even if present”.

Figure S2b (left) and Figure S2 (right) from the initial submitted manuscript.

5. Figures S3e and S3f, the activity traces in LD look remarkably “clean.” Also, if one were to estimate sleep based on those activity traces, they would not seem to correspond to the sleep data shown in Figure 1a?

We thank the reviewer for giving us the possibility to clarify this point. The software FaasX (prof. F. Rouyer’s lab – CNRF) was used to produce the actograms in Figure S3e and the period analysis in Figure S3f. Different cutoff of hash density (HD) (based on the n of movement of the fly to produce a hash mark on the actogram) can be chosen in the setting to filter the levels of activity in the actogram. Reducing the cutoff, more noise is introduced. This is evident if we compare the Figure S3e with the same actogram with a lower HD (right panels)

(Figure 5 for the reviewer). In addition, in the *Cyfp* heterozygous condition the moments of higher activity, in light-dark condition, are further extended in the dark phase (red squares), a sign that they are active for longer periods compared to controls indicating a decrease in sleeping during this period.

Figure 5 for the reviewer: Left 2 panels, initially submitted Figure S3e showing double plotted actogram of control (n = 20) and *Cyfp*^{85.1/+} (n = 18) during a period of light-dark 12:12 followed by a free running rhythm in continuous darkness (DD) (HD = 08). The start of the DD period is indicated by dashed red line. Red squares show the evening peaks at ZT12. Central and right panels, same double plotted actogram of figure S3e showing control (n = 20) and *Cyfp*^{85.1/+} (n = 18) with a lower HD = 05

Reviewer #3 (Remarks to the Author):

I would like to thank the authors for addressing my comments and other Reviewer's comments. The revision significantly improved the manuscript, and I would like to recommend the revised version for publication at Nature Communications.

We would like to thank the reviewer for his/her positive comments, and we are pleased that he/she appreciated our revised work.

Minor points:

- Typo in Supplementary Figure 3c: third group from the left should be 'R23E10' instead of R23E18'.
- Typo in Supplementary Figure 3d: third group from the left should be 'R23E10' instead of R23E18'.

Many thanks for having highlighted those typos that have been now corrected.

- Page 15, lines 351-353, 'The arousal threshold is higher at night compared to daytime, suggesting that night-time sleep might be associated with sleep homeostasis' citing Ishimoto et al 2012 as supporting reference. I could not find any claim supporting the author's statement in the reference cited. The authors should revise this sentence.

We thank the reviewer for this suggestion. That sentence has been rephrased to avoid confusion. The reference that mentions that arousal threshold differs at nighttime compared to daytime is Van Alphen et al., 2013 PMID: 23595750 included in our manuscript. In addition, we added the reference from the same group Faville et al., 2015 PMID: 25677943 showing that the fly responsiveness to stimuli is specifically suppressed following sleep deprivation at nighttime, and not during daytime. The authors, hypothesizing different levels of homeostatic pressure during daytime and nighttime sleep.

Reviewer #4 (Remarks to the Author):

Having reviewed the responses given by the authors to my comments - and to the comments of the other 3 reviewers - I believe the manuscript is now improved and I have no further comments or suggestions.

We would like to thank the reviewer for his/her positive comments and we are pleased that he/she appreciates our revised work.

REVIEWERS' COMMENTS

Reviewer #2 (Remarks to the Author):

I would first like to emphasize that I am sympathetic to the substantial efforts the authors have made during the revision process. Moreover, I am not in favor of a manuscript having to go through multiple rounds of revision. Nonetheless, there remain substantive issues to address, which given the points above, I suggest should be resolved by including text that acknowledges caveats and limitations to the study.

Comment 1:

The authors rebut the concern that changes in locomotor activity drive the changes in NADP/NADPH ratio by arguing that there is no difference between controls and Cyfip hets in terms of locomotor activity (despite differences in NADP/NADPH ratio) and further argue that they don't see changes in NADP/NADPH ratio at ZT8, ZT12, and ZT16 in Cyfip hets despite significant differences in locomotion. The problem with this argument is that Cyfip itself affects NADP/NADPH ratios, so one cannot clearly examine the relationship between locomotion and NADP/NADPH ratios in these animals. Instead, if one examines control animals, the NADP/NADPH ratio follows the locomotor activity pattern reasonably well. The authors next argue that they see an increase in NADP/NADPH ratio with nighttime sleep deprivation for 8 hrs and claim this demonstrates the NADP/NADPH ratios are changing with sleep need. The potential issue with this claim is that their sleep deprivation is driving intense locomotor activity. To disentangle the effect of sleep loss vs locomotor activity in this experiment, the authors could have performed sleep deprivation during the day, which would have resulted in a similar increase in locomotor activity and affect sleep to a much lower degree. In the absence of performing additional experiments, the authors should clearly state in the text that NADP/NADPH ratios do not follow sleep need under baseline conditions (where it is instead associated with arousal or locomotor activity) and acknowledge, despite their SD experiment, that the NADP/NADPH changes might be related to increased locomotion.

Comment 2:

Using a more traditional measure of arousal threshold, the authors now show that nearly all flies in control and mutant groups respond to the stimulus. Based on these data, the authors cannot claim that there are no differences in arousal threshold between controls and Cyfip hets. In other words, if one uses a very strong stimulus that arouses essentially all flies, one lacks any sensitivity to detect differences in arousal threshold. The correct way to perform these experiments is to use sub-maximal stimuli, where a minority or subsets of animals respond and show no differences between control and mutant. The authors should be careful to tone down any claims in the text of a lack of differences in arousal threshold.

Comment 3:

The authors provide some staining images to argue that nsyb-Gal4 does not cover all neurons and likely differs in its expression pattern from ras2-Gal4. These data are not very persuasive—are the authors claiming that the small number of neurons not covered by nsyb-Gal4 but are present in Ras2-Gal4 are responsible for their Cyfip-related phenotype? Ras2-Gal4 is not a commonly used driver to express broadly in neurons. It would be much more convincing to provide data using another pan-neuronal driver such as R57C10-Gal4. In the absence of this, the authors should at least acknowledge the possibility in the text that Cyfip may be required in non-neuronal tissues for their phenotype.

Comment 4:

The authors argue that they identified Men and SREBP in an unbiased manner and that they are not claiming that other factors are not involved in NADP/NADPH ratio. While this is true, it does not resolve the issue that their "rescue" of NADP/NADPH ratios using Cyfip heterozygotes and SREBP and MEN alleles may be non-specific. The authors should remove the word "rescue" throughout the text, which implies that these molecules are acting in the same genetic pathway, and replace it with words

like NADP/NADPH ratios can be "increased" or "enhanced."

Comment 5:

The authors now provide MEN activity data in the rebuttal, but not the manuscript itself, showing that in controls, MEN activity does not change with SD, whereas MEN activity is increased with SD in Cyfip heterozygotes. These new data argue that MEN is NOT normally involved in homeostatic regulation of sleep, but instead is pathologically upregulated in Cyfip heterozygotes which could explain downstream effects and phenotypes. The authors should put the MEN activity in the manuscript, as these data are more relevant than showing transcript level changes, which could have no functional consequences (and indeed, the authors appear to show that this is the case).

Minor Comment 5:

Please make sure to provide information in the methods that would allow readers to understand why the activity traces look that way.

Reviewer #3 (Remarks to the Author):

I would like to thank the authors for addressing my comments and other Reviewer's comments. The revision significantly improved the manuscript and I would like to recommend the revised version for publication provided they address the minor typos stated below:

- Abstract, line 39: '..sleep disorders are a frequent comorbidity of neuropsychiatric disorders'. —> change to '...sleep disturbances are a frequent comorbidity of neuropsychiatric disorders...' or similar
- Introduction, line 65: 'The sleep is regulated by ...' —> change to 'Sleep is regulated by ...'
- Introduction, line 69: '..that promotes sleep...' —> change to '..that promote sleep...'
- Results section, line 173-174: '.. in the dorsal fan-shaped bodies (dFBs/ExFI2)...' —> change to '.. in the dorsal fan-shaped body (dFB/ExFI2)...'
- Results section, line 182: '.. in ellipsoid bodies ...' —> change to '.. in ellipsoid body)...
- Results section, line 196: '.. the sleep-initiating circuitries of the fly brain are sensitive to the balance of...' —> change to '.. sleep-promoting dorsal fan-shaped body neurons are sensitive to the balance of...'. To my knowledge, it was only shown in dorsal fan-shaped body neurons that there is such a mechanism and not in multiple 'sleep circuitries'.
- Title of Supplementary Fig. 1: 'Cyfip regulates the quality of night-time sleep'. Given that it also regulates the quantity of night-time sleep I would modify the title of the legend. Please also indicate in the figure legend whether the quantification of sleep bout length, sleep bout number, sleep latency and WASO was done using daytime or nighttime sleep data or both.
- Legend Supplementary Fig. 1, line 24: 'daytime sleep' instead of 'sleep day amount'.
- Typo in figure legend of Supplementary Fig. 1, line 27: *p=0.0977. I assume the authors meant a value of p<0.05 as stated in the methods section (page 32, line 758-759)?
- I agree with Reviewer 1 that the quantification of the arousal threshold in Supplementary Figure 1g is misleading. In addition to 'Responders/total' the authors should also plot 'Responders/(Responding + Not responding)', thereby only taking into account awakenings from sleep, especially given the fact that the mutants sleep less during nighttime.
- Legend title of Supplementary Fig. 4 does not fit with content of figure. —> Change to, e.g., SREBP modulates sleep?
- Typo in Supplementary Table 1: 2x 'wakefulness' instead of 'wakefulness' in title of Supplementary Table 1.

Point by Point

Mariano et al. NCOMMS-21-46266-T

Reviewer #2 (Remarks to the Author):

I would first like to emphasize that I am sympathetic to the substantial efforts the authors have made during the revision process. Moreover, I am not in favor of a manuscript having to go through multiple rounds of revision. Nonetheless, there remain substantive issues to address, which given the points above, I suggest should be resolved by including text that acknowledges caveats and limitations to the study.

We would like to thank the reviewer for appreciating the extensive work we did to address the relevant comments we received on our manuscript. We have taken the additional comments of the reviewers into consideration and addressed the issues in the text presenting caveats and limitations of the study.

Comment 1:

The authors rebut the concern that changes in locomotor activity drive the changes in NADP/NADPH ratio by arguing that there is no difference between controls and Cyfip hets in terms of locomotor activity (despite differences in NADP/NADPH ratio) and further argue that they don't see changes in NADP/NADPH ratio at ZT8, ZT12, and ZT16 in Cyfip hets despite significant differences in locomotion. The problem with this argument is that Cyfip itself affects NADP/NADPH ratios, so one cannot clearly examine the relationship between locomotion and NADP/NADPH ratios in these animals. Instead, if one examines control animals, the NADP/NADPH ratio follows the locomotor activity pattern reasonably well. The authors next argue that they see an increase in NADP/NADPH ratio with nighttime sleep deprivation for 8 hrs and claim this demonstrates the NADP/NADPH ratios are changing with sleep need. The potential issue with this claim is that their sleep deprivation is driving intense locomotor activity. To disentangle the effect of sleep loss vs locomotor activity in this experiment, the authors could have performed sleep deprivation during the day, which would have resulted in a similar increase in locomotor activity and affect sleep to a much lower degree. In the absence of performing additional experiments, the authors should clearly state in the text that NADP/NADPH ratios do not follow sleep need under baseline conditions (where it is instead associated with arousal or locomotor activity) and acknowledge, despite their SD experiment, that the NADP/NADPH changes might be related to increased locomotion.

We thank the reviewer for the comment and suggestions. We revised the text and the discussion accordingly as follows: Importantly, we observed that in control flies the NADP+/NADPH ratio follows the behavioural state of the fly, i.e., rising at periods when the flies are awake and show increased locomotion activity (ZT0 and ZT12) and descending at ZT16 when sleep need is gradually reduced (Fig. 2a). In addition, we have added the following sentence in the discussion: The accumulation of the NADP+ over NADPH during sleep deprivation in wild-type flies, suggests its involvement in the homeostatic sleep regulation (Fig. 2). We cannot exclude that NADP/NADPH changes might be related to increased locomotion activity: period of motor/wakefulness increase sleep need. In support of our model, a mutation that impacts the NADPH+/NADPH ratio alters sleep behaviour (Fig. 5).

Comment 2:

Using a more traditional measure of arousal threshold, the authors now show that nearly all flies in control and mutant groups respond to the stimulus. Based on these data, the authors cannot claim that there are no differences in arousal threshold between controls and Cyfip hets. In other words, if one uses a very strong stimulus that arouses essentially all flies, one lacks any sensitivity to detect differences in arousal threshold. The correct way to perform these experiments is to use sub-maximal stimuli, where a minority or subsets of animals respond and show no differences between control and mutant. The authors should be careful to tone down any claims in the text of a lack of differences in arousal threshold.

We thank the reviewer for his comments. We have revised our conclusions as follows acknowledging that the Cyfip heterozygous flies might show increased arousal as follows: The number of flies that were awakened by the light stimulus did not differ between Cyfip85.1/+ and control conditions, with a stimulus sufficiently strong to disturb efficiently the overall population (Supplementary Fig. 1g). Instead, Cyfip

mutants showed a reduction by approximately 50% of baseline sleep over the three time points, thus remaining awake for a longer time (Fig. 1d), suggesting a state of increased arousal.

Comment 3:

The authors provide some staining images to argue that *nsyb-Gal4* does not cover all neurons and likely differs in its expression pattern from *ras2-Gal4*. These data are not very persuasive—are the authors claiming that the small number of neurons not covered by *nsyb-Gal4* but are present in *Ras2-Gal4* are responsible for their *Cyfp*-related phenotype? *Ras2-Gal4* is not a commonly used driver to express broadly in neurons. It would be much more convincing to provide data using another pan-neuronal driver such as *R57C10-Gal4*. In the absence of this, the authors should at least acknowledge the possibility in the text that *Cyfp* may be required in non-neuronal tissues for their phenotype.

We thank the reviewer for his comments. We have revised the text and acknowledged the possibility that *Cyfp* might be required in also in non-neuronal tissues for its phenotype as follows: Since the *Ras2Gal4* driver is not exclusively neuronal^{63, 64}, *CYFIP* might be required also in non-neuronal tissue to regulate night-time sleep behaviour.

Comment 4:

The authors argue that they identified *Men* and *SREBP* in an unbiased manner and that they are not claiming that other factors are not involved in *NADP/NADPH* ratio. While this is true, it does not resolve the issue that their “rescue” of *NADP/NADPH* ratios using *Cyfp* heterozygotes and *SREBP* and *MEN* alleles may be non-specific. The authors should remove the word “rescue” throughout the text, which implies that these molecules are acting in the same genetic pathway and replace it with words like *NADP/NADPH* ratios can be “increased” or “enhanced.”

We have now revised the text according to the reviewer's suggestions.

Comment 5:

The authors now provide *MEN* activity data in the rebuttal, but not the manuscript itself, showing that in controls, *MEN* activity does not change with SD, whereas *MEN* activity is increased with SD in *Cyfp* heterozygotes. These new data argue that *MEN* is NOT normally involved in homeostatic regulation of sleep, but instead is pathologically upregulated in *Cyfp* heterozygotes which could explain downstream effects and phenotypes. The authors should put the *MEN* activity in the manuscript, as these data are more relevant than showing transcript level changes, which could have no functional consequences (and indeed, the authors appear to show that this is the case).

We thank the reviewer for this suggestion, we have now included this experiment in the manuscript as Supplementary Figure 5h and i. The text now reads as follows: Notably, despite the increase of *Men* transcript upon sleep deprivation in control flies, we did not observed changes in *Men* activity (Supplementary Fig. 5h). On the contrary *Men* activity results increased, upon sleep deprivation, in *Cyfp* heterozygous flies (Supplementary Fig. 5i). This is in line with the previously observed deficits in the *NADP+/NADPH* ratio after sleep deprivation in *Cyfp85.1/+* flies (Fig. 2b) and further underlines that the pathological upregulation of *Men* drives changes in the *NADP+/NADPH* ratio and sleep deficits of *Cyfp* heterozygous flies.

On addition, in the discussion we state: Notably, a key player in regulating the *NADP+/NADPH* balance is the malic enzyme, which catalyses the conversion of cytosolic malate in pyruvate, reducing *NADP+* at *NADPH*. *Men* expression and activity is pathologically upregulated in *Cyfp85.1/+* flies, resulting in the impaired *NADP+/NADPH* and sleep. Similar to *Men*, dysregulation of other enzymes involved in *NADP+/NADPH* homeostasis might contribute to sleep disorders.

Minor Comment 5:

Please make sure to provide information in the methods that would allow readers to understand why the activity traces look that way.

We revised the text according to the reviewer's suggestions and explained in the Methods section how the activity traces were analyzed: "Actograms are presented as double-plotted group average activity and were generated using the following settings: Light Cycle: LD_2400; Filter: Off; Hash Density: 08: Light on: ZT0 (08:30), Light off ZT12 (20:30); Fly survival: through the entire experiment."

Reviewer #3 (Remarks to the Author):

I would like to thank the authors for addressing my comments and other Reviewer's comments. The revision significantly improved the manuscript and I would like to recommend the revised version for publication provided they address the minor typos stated below:

We would like to thank the reviewer for his/her positive comments, and we are pleased to read that he/she appreciated our revised work.

- Abstract, line 39: '..sleep disorders are a frequent comorbidity of neuropsychiatric disorders'. —> change to '...sleep disturbances are a frequent comorbidity of neuropsychiatric disorders...' or similar
- Introduction, line 65: 'The sleep is regulated by ...' —> change to 'Sleep is regulated by ...'
- Introduction, line 69: '..that promotes sleep...' —> change to '...that promote sleep...'
- Results section, line 173-174: '.. in the dorsal fan-shaped bodies (dFBs/ExFI2)...' —> change to '.. in the dorsal fan-shaped body (dFB/ExFI2)...'
- Results section, line 182: '.. in ellipsoid bodies ...' —> change to '.. in ellipsoid body)...
- Results section, line 196: '.. the sleep-initiating circuitries of the fly brain are sensitive to the balance of...' —> change to '.. sleep-promoting dorsal fan-shaped body neurons are sensitive to the balance of...'. To my knowledge, it was only shown in dorsal fan-shaped body neurons that there is such a mechanism and not in multiple 'sleep circuitries'.
- Title of Supplementary Fig. 1: 'Cyfip regulates the quality of night-time sleep'. Given that it also regulates the quantity of night-time sleep I would modify the title of the legend. Please also indicate in the figure legend whether the quantification of sleep bout length, sleep bout number, sleep latency and WASO was done using daytime or nighttime sleep data or both.
- Legend Supplementary Fig. 1, line 24: 'daytime sleep' instead of 'sleep day amount'.
- Typo in figure legend of Supplementary Fig. 1, line 27: *p=0.0977. I assume the authors meant a value of p<0.05 as stated in the methods section (page 32, line 758-759)?

We have revised the text accordingly.

- I agree with Reviewer 1 that the quantification of the arousal threshold in Supplementary Figure 1g is misleading. In addition to 'Responders/total' the authors should also plot 'Responders/(Responding + Not responding)', thereby only taking into account awakenings from sleep, especially given the fact that the mutants sleep less during nighttime.

We have revised the Supplementary figure 1g to make it clear and inserted the percentage of flies Responders/(responding + not responding) taking into accounts as "responders" only flies that awaken from sleep after the light stimulus.

- Legend title of Supplementary Fig. 4 does not fit with content of figure. —> Change to, e.g., SREBP modulates sleep?
- Typo in Supplementary Table 1: 2x 'wakefulness' instead of 'wakefulness' in title of Supplementary Table 1.

We thank the reviewer for his/her comments, and we revised the text accordingly.